# Minimax Optimal Rate for Parameter Estimation in Multivariate Deviated Models

**Dat Do\***
Department of Statistics
University of Michigan at Ann Arbor
Ann Arbor, MI 48109
dodat@umich.edu

**Huy Nguyen\***
Department of Statistics and Data Sciences
The University of Texas at Austin
Austin, TX 78712
huynm@utexas.edu

**Khai Nguyen**
Department of Statistics and Data Sciences
The University of Texas at Austin
Austin, TX 78712
khainb@utexas.edu

**Nhat Ho**
Department of Statistics and Data Sciences
The University of Texas at Austin
Austin, TX 78712
minhnhat@utexas.edu

## Abstract

We study the maximum likelihood estimation (MLE) in the multivariate deviated model where the data are generated from the density function $(1 - \lambda^*)h_0(x) + \lambda^* f(x|\mu^*, \Sigma^*)$ in which $h_0$ is a known function, $\lambda^* \in [0, 1]$ and $(\mu^*, \Sigma^*)$ are unknown parameters to estimate. The main challenges in deriving the convergence rate of the MLE mainly come from two issues: (1) The interaction between the function $h_0$ and the density function $f$; (2) The deviated proportion $\lambda^*$ can go to the extreme points of $[0, 1]$ as the sample size tends to infinity. To address these challenges, we develop the *distinguishability condition* to capture the linear independent relation between the function $h_0$ and the density function $f$. We then provide comprehensive convergence rates of the MLE via the vanishing rate of $\lambda^*$ to zero as well as the distinguishability of two functions $h_0$ and $f$.

## 1 Introduction

The goodness-of-fit test [11] is one of the foundational tools in statistics with several applications in data-driven scientific fields, namely kernel Stein discrepancy [27, 31], point processes [37] and Bayesian statistics [32], etc. Given a sample set of data and a pre-specified distribution with density function $h_0$, the test indicates whether the samples are reasonably distributed according to $h_0$ (*null hypothesis*) or to another family of distributions $\{p(\cdot|\theta) : \theta \in \Theta\}$ (*alternative hypothesis*). It is worth noting that knowledge about the null hypothesis distribution can come from prior knowledge of scientists. A key to understanding the statistical efficiency of testing is via the likelihood ratio and the maximum likelihood estimation (MLE) methods. [6].

While traditional testing problems often assume the null distribution $h_0 = p(\cdot|\theta_0)$ and the alternative one $p(\cdot|\theta)$ are from a single simple family of distributions such as exponential families, there are also many problems in science require to test $h_0$ against the alternative $f(\cdot|\theta)$ that can be *deviated* from $h_0$ by a distribution from a potentially different family. Specifically, in this paper, we consider the family of distributions named *multivariate deviated model* with density functions defined as follows:

$$p_G(x) := (1 - \lambda)h_0(x) + \lambda f(x|\mu, \Sigma), \tag{1}$$

---

\* Equal contribution.

37th Conference on Neural Information Processing Systems (NeurIPS 2023).

where $x \in \mathbb{R}^d$, $G := (\lambda, \mu, \Sigma)$ are the model's parameters with $\lambda \in [0,1]$ being the *deviated proportion* (from $h_0$) and $(\mu, \Sigma) \in \Theta \times \Omega$ are parameters of a vector-matrix family of distributions $f$, where $\Theta \subset \mathbb{R}^d$ and $\Omega \subset \mathbb{R}^{d \times d}$ being compact. When $\lambda = 0$, this recovers the null hypothesis distribution $h_0$.

The deviated model can be motivated by many applications in science. For instance, in microarray data analysis, it can be used to detect differentially expressed genes under two or more conditions [1, 2], where $h_0$ is the uniform distribution and $f(\cdot|\mu, \Sigma)$ is required to estimate. Many other applications can be seen in many contamination problems in astronomy and biology [29]. Besides, the deviated model can also be viewed as a low-rank adaptation model in the domain adaptation problem [23], where $h_0$ is a pre-trained model on large data, and $f$ is a simpler component to be estimated from the smaller data domain. Our goal in this paper is to study the parameter estimation rate of the deviated model.

**Problem setup.** Suppose that we observe $n$ i.i.d. samples $X_1, \ldots, X_n$ from the true multivariate deviated model:

$$p_{G_*}(x) := (1 - \lambda^*)h_0(x) + \lambda^* f(x|\mu^*, \Sigma^*), \tag{2}$$

where $G_* := (\lambda^*, \mu^*, \Sigma^*)$ are true but unknown parameters with $\lambda^* \neq 0$. Throughout the paper, we allow $G_*$ to change with the sample size $n$ (see Appendix F.1 for a discussion). To facilitate our presentation, we suppress the dependence of $G_*$ on $n$, and then estimate $G_*$ from the data. The main focus of this paper is to establish both a uniform convergence rate and minimax rate for parameter estimation via the MLE approach, which is given by:

$$\widehat{G}_n \in \arg\max_{G \in \Xi} \sum_{i=1}^{n} \log p_G(X_i), \tag{3}$$

where $\widehat{G}_n := (\widehat{\lambda}_n, \widehat{\mu}_n, \widehat{\Sigma}_n)$ and $\Xi := [0, 1] \times \Theta \times \Omega$.

**Contribution.** There are two main challenges in studying the convergence rate of the MLE $\widehat{G}_n$: (1) The interaction between the function $h_0$ and the density function $f$, e.g., $h_0$ belongs to the family of $f$ and $(\mu^*, \Sigma^*)$ approaches $h_0$ as the sample size $n$ goes to infinity; (2) The deviated proportion $\lambda^*$ can go to the extreme points of $[0, 1]$ as the sample size goes to infinity and make the estimation become more challenging, because when $\lambda^* = 0$, all the parameters $(\mu^*, \Sigma^*)$ yield the same model. To address these *singularity* and *identifiability* issues, we first develop the *distinguishability condition* to capture the linear independent relation between the function $h_0$ and the density function $f$. We then study the optimal convergence rate of parameters under both distinguishable and non-distinguishable settings of the multivariate deviated model. Our theoretical results can be summarized as follows:

**1. Distinguishable settings:** We demonstrate that as long as the function $h_0$ and the density function $f$ are distinguishable, the convergence rate of $\widehat{\lambda}_n$ to $\lambda^*$ is $\mathcal{O}(n^{-1/2})$ while $(\widehat{\mu}_n, \widehat{\Sigma}_n)$ converges to $(\mu^*, \Sigma^*)$ at a rate determined by the vanishing rate of $\lambda^*$ as follows:

$$\lambda^* \|(\widehat{\mu}_n, \widehat{\Sigma}_n) - (\mu^*, \Sigma^*)\| = \mathcal{O}(n^{-1/2}).$$

It indicates that if $\lambda^*$ goes to 0, the convergence rate of estimating $(\mu^*, \Sigma^*)$ is slower than the parametric rate.

**2. Non-distinguishable settings:** When $h_0$ and $f$ are not distinguishable, it becomes complicated to capture the convergence rate of the MLE. To shed light on the behaviors of the MLE under the non-distinguishable settings of multivariate deviated model, we specifically study the settings when $h_0$ belongs to the same family as $f$, namely, $h_0(.) = f(.|\mu_0, \Sigma_0)$ for some $(\mu_0, \Sigma_0)$. To precisely characterize the rates of the MLE under this setting, we consider the second-order strong identifiability of $f$, which requires the linear independence up to second-order derivatives of $f$ with respect to its parameters. The second-order identifiability had also been considered in the literature to investigate the convergence rate of parameter estimation in finite mixtures [9, 28, 22, 21, 20, 19].

**2.1. Strongly identifiable and non-distinguishable settings:** When $f$ is strongly identifiable in the second order, we demonstrate that $\|(\Delta\mu^*, \Delta\Sigma^*)\|^2 |\widehat{\lambda}_n - \lambda^*| = \mathcal{O}(n^{-1/2})$ and

$$\lambda^* (\|(\Delta\mu^*, \Delta\Sigma^*)\| + \|(\Delta\widehat{\mu}_n, \Delta\widehat{\Sigma}_n\|)\|(\widehat{\mu}_n, \widehat{\Sigma}_n) - (\mu^*, \Sigma^*)\| = \mathcal{O}(n^{-1/2}),$$

where $\Delta\mu := \mu - \mu_0$ and $\Delta\Sigma := \Sigma - \Sigma_0$. It indicates that the convergence rate of $\widehat{\lambda}_n$ to $\lambda^*$ depends on that of $(\mu^*, \Sigma^*)$ to $(\mu_0, \Sigma_0)$ while the convergence rate of $(\widehat{\mu}_n, \widehat{\Sigma}_n)$ to $(\mu^*, \Sigma^*)$ depends on both the rate of $\lambda^*$ to 0 and the rate of $(\mu^*, \Sigma^*)$ to $(\mu_0, \Sigma_0)$. These results are strictly different from those in the distinguishable settings, which is mainly due to the non-distinguishability between $h_0$ and $f$.

**2.2. Weakly identifiable and non-distinguishable settings:** When $f$ is weakly identifiable, i.e., it is not strongly identifiable in the second order, we specifically consider the popular setting when $f$ is the density of a multivariate Gaussian distribution. The loss of the strong identifiability of the Gaussian distribution is due to the following partial differential equation (PDE) between the location and scale parameters (the heat equation):

$$\frac{\partial^2 f}{\partial\mu\partial\mu^\top}(x|\mu, \Sigma) = 2\frac{\partial f}{\partial\Sigma}(x|\mu, \Sigma).$$

Due to the above PDE, the convergence rate of the MLE under this setting exhibits very different behaviors from those under the strongly identifiable setting. In particular, we prove that $\left[\|\Delta\mu^*\|^4 + \|\Delta\Sigma^*\|^2\right]|\widehat{\lambda}_n - \lambda^*| = \mathcal{O}(n^{-1/2})$ and

$$\lambda^*(\|\Delta\mu^*\|^2 + \|\Delta\widehat{\mu}_n\|^2 + \|\Delta\Sigma^*\| + \|\Delta\widehat{\Sigma}_n\|)(\|\widehat{\mu}_n - \mu^*\|^2 + \|\widehat{\Sigma}_n - \Sigma^*\|) = \mathcal{O}(n^{-1/2}).$$

Notably, there is a mismatch in the orders of convergence rates of the location vector and covariance matrix. Furthermore, the rate of the deviated mixing proportion also depends on different orders of $\mu^*$ to $\mu_0$ and $\Sigma^*$ to $\Sigma_0$. Such rich behaviors of the MLE are mainly due to the PDE between the location and scale parameters.

**Comparing to moment methods.** We would like to remark that the results for the MLE under the non-distinguishable settings in the paper are (much) tighter than those obtained from moment methods for a general mixture of two components in the literature. In particular, when $f$ is multivariate Gaussian distribution with fixed covariance matrix, i.e., $f$ is strongly identifiable in the second order and we do not estimate $\Sigma^*$, an application of the results with moment methods from [36] to the deviated models leads to $\|\Delta\mu^*\|^3|\lambda_n^{\text{moment}} - \lambda^*| = \mathcal{O}(n^{-1/2})$ and $\lambda^*\|\mu_n^{\text{moment}} - \mu^*\|^3 = \mathcal{O}(n^{-1/2})$, which are much slower compared to the results for the MLE in the strongly identifiable and non-distinguishable settings, where $(\lambda_n^{\text{moment}}, \mu_n^{\text{moment}})$ denote moment estimators of $\lambda^*$ and $\mu^*$.

When $f$ is a multivariate Gaussian density and we estimate both the location vector and covariance matrix, i.e., $f$ is weakly identifiable, an adaptation of the moment estimators from the seminal work [18] to the multivariate deviated models shows that $\lambda^*\|\|\tilde{\mu}_n^{\text{moment}} - \mu^*\|^6 = \mathcal{O}(n^{-1/2})$, $\lambda^*\|\|\tilde{\Sigma}_n^{\text{moment}} - \Sigma^*\|^3 = \mathcal{O}(n^{-1/2})$ and $(\|\Delta\mu^*\|^6 + \|\Delta\Sigma^*\|^3)|\tilde{\lambda}_n^{\text{moment}} - \lambda^*| = \mathcal{O}(n^{-1/2})$, where $(\tilde{\lambda}_n^{\text{moment}}, \tilde{\mu}_n^{\text{moment}}, \tilde{\Sigma}_n^{\text{moment}})$ are moment estimators of $(\lambda^*, \mu^*, \Sigma^*)$. These results are also much slower than those of the MLE in weakly identifiable settings.

**Other related work.** The hypothesis testing and MLE problem related to the multivariate deviated model had been considered in previous work, including the problem of detecting sparse homogeneous and heteroscedastic mixtures [14, 15, 4, 3, 5, 34], the problem of determining the number of components [8, 26, 10, 24, 25], and the problem of multiple testing [30, 12]. In particular, [4] considers testing problem for the deviated model with $h_0 = N(0, 1)$ and $f = N(\mu^*, 1)$ being one-dimensional Gaussian distributions. They show that no test can reliably detect $\lambda^* = 0$ against $\lambda^* > 0$ if $\lambda^*\mu^* = o(n^{-1/2})$, while the Likelihood Ratio test can consistently do it when $\lambda^*\mu^* \gtrsim n^{-1/2+\epsilon}$ for any $\epsilon > 0$. However, no guarantee for estimation of $\lambda^*$ and $\mu^*$ is provided. In the same setting where $f$ is the density of a location Gaussian distribution, the convergence rate of parameter estimation in the deviated model had been studied in the work of [16]. Since the location Gaussian distribution is a special case of the strongly identifiable distribution, our result in the strongly identifiable and non-distinguishable settings is a generalization of the results in [16], but with a different proof technique as their proof technique relies strictly on the properties of the location Gaussian distribution.

**Organization.** The paper is organized as follows. In Section 2, we provide background on the identifiability and density estimation rate of the multivariate deviated model. Then, we establish the lower bounds of the Total Variation distance between two densities in terms of loss functions among parameters under both the distinguishable and non-distinguishable settings in Section 3. Next, we characterize the convergence rates of parameter estimation as well as derive the corresponding minimax lower bounds in Section 4. In Section 5, we carry out a simulation study to empirically verify our theoretical results before concluding the paper in Section 6. Rigorous proofs and additional results are deferred to the supplementary material.

**Notations.** For any $a, b \in \mathbb{R}$, we denote $a \vee b := \max\{a, b\}$ and $a \wedge b := \min\{a, b\}$. Next, we say that $h_0$ is identical to $f$ if $h_0(x) = f(x|\mu_0, \Sigma_0)$ for some $(\mu_0, \Sigma_0) \in \Theta \times \Sigma$. For each parameter $G \in \Xi$, let $\mathbb{E}_{p_G}$ be the expectation taken with respect to product measure with density $p_G$. Lastly, for any two density functions $p$ and $q$ (with respect to the Lebesgue measure $m$), the Total Variation distance between them is given by $V(p, q) := \frac{1}{2} \int |p(x) - q(x)| dm(x)$, while we define their squared Hellinger distance as $h^2(p, q) := \frac{1}{2} \int [\sqrt{p(x)} - \sqrt{q(x)}]^2 dm(x)$.

## 2 Preliminaries

### 2.1 Identifiability Condition

Our principal goal in this paper is to assess the statistical efficiency of parameter estimation from the MLE method. To do that, we should be able to guarantee the parameter identifiability of the deviated model (2), i.e., if $p_G(x) = p_{G_*}(x)$ for almost surely $x \in \mathcal{X}$ where $G = (\lambda, \mu, \Sigma)$, then $G \equiv G_*$. That identifiability condition leads to the following notion of distinguishability between the density function $h_0(\cdot)$ and the family of density functions $\{f(\cdot|\mu, \Sigma) : (\mu, \Sigma) \in \Theta \times \Omega\}$.

**Definition 2.1 (Distinguishability).** We say that the family of density functions $\{f(\cdot|\mu, \Sigma), (\mu, \Sigma) \in \Theta \times \Omega\}$ (or in short, $f$) is *distinguishable* from $h_0$ if the following holds:

A1. For any two distinct components $(\mu_1, \Sigma_1)$ and $(\mu_2, \Sigma_2)$, if we have real coefficients $\eta_i$ for $1 \leq i \leq 3$ such that $\eta_1 \eta_2 \leq 0$ and $\eta_1 f(x|\mu_1, \Sigma_1) + \eta_2 f(x|\mu_2, \Sigma_2) + \eta_3 h_0(x) = 0$, for almost surely $x \in \mathbb{R}^d$, then $\eta_1 = \eta_2 = \eta_3 = 0$.

We can verify that as long as $f$ is distinguishable from $h_0$, the parameter identifiability of our multivariate deviated model follows. In particular, assume that there exists $G = (\lambda, \mu, \Sigma)$ such that

$$(1 - \lambda^*)h_0(x) + \lambda^* f(x|\mu^*, \Sigma^*) = (1 - \lambda)h_0(x) + \lambda f(x|\mu, \Sigma), \tag{4}$$

for almost surely $x \in \mathcal{X}$. The above equation is equivalent to $(\lambda - \lambda^*)h_0(x) + \lambda^* f(x|\mu^*, \Sigma^*) - \lambda f(x|\mu, \Sigma) = 0$. Assume that $f$ is distinguishable from $h_0$, then equation (4) indicates that if $(\mu, \Sigma) \neq (\mu^*, \Sigma^*)$, we have $\lambda = \lambda^* = 0$. Since $\lambda^* \neq 0$ from our assumption, we obtain that $(\mu, \Sigma) = (\mu^*, \Sigma^*)$. As a result, equation (4) becomes $(\lambda - \lambda^*)h_0(x) + (\lambda^* - \lambda)f(x|\mu, \Sigma) = 0$. By applying the distinguishability condition again, we get $\lambda = \lambda^*$. Therefore, the multivariate deviated model (2) is identifiable.

In the following example, we will verify the distinguishability condition in Definition 2.1 given some specific choices of function $h_0$ and density $f$.

**Example 2.2.** (a) Assume that $f$ belongs to a location family of density functions, i.e., $f(x|\mu, \Sigma) = f_\Sigma(x - \mu)$ for all $x$ where $\Sigma$ is a fixed covariance matrix. If $h_0(x) \neq f(x)$ for almost surely $x \in \mathcal{X}$, then $f$ is distinguishable from $h_0$.
(b) When $h_0$ is a finite mixture of multivariate Gaussian densities and $f$ belongs to a class of multivariate Student's density functions with any fixed odd degree of freedom $\nu > 1$, we get that $f$ is distinguishable from $h_0$.
(c) When $f$ is identical to $h_0$, then $f$ is not distinguishable from $h_0$.

### 2.2 Convergence Rate of Density Estimation

Our strategy to obtain the convergence rate of the MLE $\widehat{G}_n$ is by first establishing the convergence rate of density $p_{\widehat{G}_n}$ and then studying the geometric inequalities between the parameter space and density space. For the former, the standard method is to use the empirical process theory [17, 33], while for the latter step, we investigate those inequalities under various settings of distinguishability in Section 3. Due to space constraints and the popularity of empirical process theory, we choose to informally present a main result for yielding the parametric convergence rate for density estimation in this section. For full explanation and definition, readers are referred to Appendix B. The convergence rate for density estimation can be characterized by bounding the complexity of the parameter space $\Xi$ via a function called *bracketing entropy integral* $\mathcal{J}_B(\epsilon, \overline{\mathcal{P}}^{1/2}(\Xi, \epsilon))$ (cf. equation (8)).

**Theorem 2.3.** *Assume the following assumption holds:*

*A2. Given a universal constant $J > 0$, there exists $N > 0$, possibly depending on $\Xi$, such that for all $n \geq N$ and all $\epsilon > (\log(n)/n)^{1/2}$, we have $\mathcal{J}_B(\epsilon, \overline{\mathcal{P}}^{1/2}(\Xi, \epsilon)) \leq J\sqrt{n}\epsilon^2$.*

*Then, there exists a constant $C > 0$ depending only on $\Xi$ such that for all $n \geq 1$,*

$$\sup_{G_* \in \Xi} \mathbb{E}_{p_{G_*}} h(p_{\widehat{G}_n}, p_{G_*}) \leq C\sqrt{\log n/n}.$$

Therefore, in order to get the convergence rate for density estimators based on the MLE method, we only need to check Assumption A2, which holds true for several parametric models [33]. For our model, we give an example that it holds for a general class of $f$ and $h_0$.

**Proposition 2.4.** *Suppose that both $\Theta$ and $\Omega$ are compact, and $\{f(x|\mu, \Sigma) : \mu \in \Theta, \Sigma \in \Omega\}$ is a vector-matrix family of densities being uniformly bounded, Lipschitz, and light tail, i.e. there exists constants $M, L, B, b_1, b_2, b_3 > 0$ such that $|f(x|\mu, \Sigma)| \leq M, |f(x|\mu, \Sigma) - f(x|\mu', \Sigma')| \leq L(\|\mu - \mu'\| + \|\Sigma - \Sigma'\|)$ for all $x \in \mathbb{R}^d$, and*

$$|f(x|\mu, \Sigma)| \leq b_1 \exp(-b_2 \|x\|^{b_3}) \quad \forall \|x\| > B,$$

*for all $(\mu, \Sigma) \in \Theta \times \Omega$. Additionally, if the density $h_0$ is bounded, then the corresponding multivariate deviated model defined in equation (1) satisfies assumption A2.*

**Example 2.5.** We can check that the location-scale Gaussian density $f(x|\mu, \Sigma)$ with $\Sigma \in \Omega$ having eigenvalues bounded below by a positive constant satisfies the condition of Proposition 2.4. This condition for $h_0$ is mild and is satisfied by most distributions such as Gaussian and t-distribution.

# 3 From the Convergence Rate of Densities to Rate of Parameters

The objective of this section is to develop a general theory according to which a small distance between $p_G$ and $p_{G_*}$ under the Hellinger distance (or Total Variation distance) would imply that $G$ and $G_*$ are also close under appropriate distance where $G = (\lambda, \mu, \Sigma)$ and $G_* = (\lambda^*, \mu^*, \Sigma^*)$. By combining those results with Theorem 2.3, we can obtain the convergence rate for parameter estimation (cf. Section 4). The distinguishability condition between $h_0$ and $f$ implicitly requires that $p_G = p_{G_*}$ would entail $G = G_*$; however, to obtain quantitative bounds for their Total Variation distance, we need stronger notions of both distinguishability and classical parameter identifiability, ones which involve higher order derivatives of the densities $h_0$ and $f$, taken with respect to mixture model parameters. Throughout the rest of this section, we denote $G = (\lambda, \mu, \Sigma)$ and $G_* = (\lambda^*, \mu^*, \Sigma^*)$.

## 3.1 Distinguishable Settings

**Definition 3.1 (First-order Distinguishability).** We say that $f$ is distinguishable from $h_0$ up to the first order if $f$ is differentiable in $(\mu, \Sigma)$, and the following holds:

D1. For any component $(\mu', \Sigma') \in \Theta \times \Omega$, if we have real coefficients $\eta, \tau_\alpha$ for all $\alpha = (\alpha_1, \alpha_2) \in \mathbb{N}^{d_1} \times \mathbb{N}^{d_2 \times d_2}, |\alpha| = |\alpha_1| + |\alpha_2| \leq 1$ such that

$$\eta h_0(x) + \sum_{|\alpha| \leq 1} \tau_\alpha \frac{\partial^{|\alpha|} f}{\partial \mu^{\alpha_1} \partial \Sigma^{\alpha_2}}(x|\mu', \Sigma') = 0$$

for all $x \in \mathcal{X}$, then $\eta = \tau_\alpha = 0$ for all $|\alpha| \leq 1$.

We can verify that the examples from part (a) and part (b) of Example 2.2 satisfy the first-order distinguishability condition. Next, we introduce a notion of uniform Lipschitz condition in the following definition.

**Definition 3.2 (Uniform Lipschitz).** We say that $f$ admits uniform Lipschitz condition up to the first order if the following holds: there are positive constants $\delta_1, \delta_2$ such that for any $R_1, R_2, R_3 > 0, \gamma_1 \in \mathbb{R}^{d_1}, \gamma_2 \in \mathbb{R}^{d_2 \times d_2}, R_1 \leq \lambda_{\min}^{1/2}(\Sigma_1) \leq \lambda_{\max}^{1/2}(\Sigma_2) \leq R_2, \|\mu_1\|, \|\mu_2\| \leq R_3, \mu_1, \mu_2 \in \Theta, \Sigma_1, \Sigma_2 \in \Omega$, we can find positive constants $C(R_1, R_2)$ and $C(R_3)$ such that for all $x \in \mathcal{X}$,

$$\left| \gamma_1^\top \left( \frac{\partial f}{\partial \mu}(x|\mu_1, \Sigma) - \frac{\partial f}{\partial \mu}(x|\mu_2, \Sigma) \right) \right| \leq C(R_1, R_2)\|\mu_1 - \mu_2\|^{\delta_1}\|\gamma_1\|,$$

$$\left| \text{tr}\left( \left( \frac{\partial f}{\partial \Sigma}(x|\mu, \Sigma_1) - \frac{\partial f}{\partial \Sigma}(x|\mu, \Sigma_2) \right)^\top \gamma_2 \right) \right| \leq C(R_3)\|\Sigma_1 - \Sigma_2\|^{\delta_2}\|\gamma_2\|.$$

Now, we have the following results characterizing the behavior of $V(p_G, p_{G_*})$ regarding the variation of $G$ and $G_*$.

**Theorem 3.3.** *Assume that $f$ is distinguishable from $h_0$ up to the first order. Furthermore, $f$ admits uniform Lipschitz condition up to the first order. For any $G$ and $G_*$, we define*

$$\mathcal{K}(G, G_*) := |\lambda - \lambda^*| + (\lambda + \lambda^*)\|(\mu, \Sigma) - (\mu^*, \Sigma^*)\|.$$

*Then, the following holds:*

$$C.\mathcal{K}(G, G_*) \leq V(p_G, p_{G_*}) \leq C_1.\mathcal{K}(G, G_*),$$

*for all $G$ and $G_*$, where $C$ and $C_1$ are two positive constants depending only on $\Theta$, $\Omega$, and $h_0$.*

See Appendix C.1 for the proof of Theorem 3.3. Since the MLE approach yields the convergence rate $n^{-1/2}$ up to some logarithmic factor for $p_{G_*}$ under the first order uniform Lipschitz condition of $f$, the result of Theorem 3.3 directly yields the convergence rate $n^{-1/2}$ up to some logarithmic factor for $G_*$ under metric $\mathcal{K}$. This entails that the estimation of weight $\lambda_*$ converges at rate $n^{-1/2}$ up to some logarithmic factor while the convergence rate of estimating $(\mu^*, \Sigma^*)$ is typically much slower than $n^{-1/2}$ as it depends on the rate of convergence of $\lambda^*$ to 0 (cf. Theorem 4.1).

### 3.2 Non-distinguishable Settings

When $f$ is not distinguishable to $h_0$ up to the first order, the bound in Theorem 3.3 may not hold in general. In this section, we investigate the inverse bounds under the specific settings of non-distinguishable in the first-order models when $h_0$ belongs to the family $f(\cdot|\mu, \Sigma)$, i.e., $h_0(x) = f(x|\mu_0, \Sigma_0)$ for some $(\mu_0, \Sigma_0) \in \Theta \times \Sigma$. Our studies are divided into two separate regimes of $f$: the first setting is when $f$ is strongly identifiable in the second order (cf. Definition 3.4), while the second setting is when it is not. For the simplicity of the presentation in the paper, we define $(\Delta\mu, \Delta\Sigma) = (\mu - \mu_0, \Sigma - \Sigma_0)$ for any element $(\mu, \Sigma) \in \Theta \times \Omega$.

**Definition 3.4** (**Strong Identifiability**). We say that $f$ is strongly identifiable in the second order if $f$ is twice differentiable in $(\mu, \Sigma)$ and the following holds:

D2. For any positive integer $k$, given $k$ distinct pairs $(\mu_1, \Sigma_1), \ldots, (\mu_k, \Sigma_k)$, if we have $\alpha_\eta^{(i)}$ such that

$$\sum_{\ell=0}^{2} \sum_{|\eta|=\ell} \sum_{i=1}^{k} \alpha_\eta^{(i)} \frac{\partial^{|\eta|} f}{\partial \mu^{\eta_1} \partial \Sigma^{\eta_2}}(x|\mu_i, \Sigma_i) = 0,$$

for almost all $x \in \mathcal{X}$, then $\alpha_\eta^{(i)} = 0$ for all $i \in [k]$ and $|\eta| \leq 2$.

#### 3.2.1 Strongly Identifiable Settings

Now, we have the following result regarding the lower bound of $V(p_G, p_{G_*})$ under the strongly identifiable settings of $f$.

**Theorem 3.5.** *Assume that $h_0(x) = f(x|\mu_0, \Sigma_0)$ for some $(\mu_0, \Sigma_0) \in \Theta \times \Sigma$ and $f$ is strongly identifiable in the second order and admits uniform Lipschitz condition up to the second order. Furthermore, we denote*

$$\mathcal{D}(G, G_*) := \lambda\|(\Delta\mu, \Delta\Sigma)\|^2 + \lambda^*\|(\Delta\mu^*, \Delta\Sigma^*)\|^2 - \min\{\lambda, \lambda^*\}\left(\|(\Delta\mu, \Delta\Sigma)\|^2\|(\Delta\mu^*, \Delta\Sigma^*)\|^2\right)$$
$$+ \left(\lambda\|(\Delta\mu, \Delta\Sigma)\| + \lambda^*\|(\Delta\mu^*, \Delta\Sigma^*)\|\right)\|(\mu, \Sigma) - (\mu^*, \Sigma^*)\|$$

*for any $G$ and $G_*$. Then, there exists a positive constant $C$ depending only on $\Theta$, $\Omega$, and $(\mu_0, \Sigma_0)$ such that $V(p_G, p_{G_*}) \geq C.\mathcal{D}(G, G_*)$, for all $G$ and $G_*$.*

The proof of Theorem 3.5 and the second-order uniform Lipschitz condition are deferred to Appendix C.2. Several remarks regarding Theorem 3.5 are in order:

**(i)** For any $G$ and $G_*$, by defining

$$\overline{\mathcal{D}}(G, G_*) := |\lambda^* - \lambda|\|(\Delta\mu, \Delta\Sigma)\|\|(\Delta\mu^*, \Delta\Sigma^*)\|$$
$$+ \|(\mu, \Sigma) - (\mu^*, \Sigma^*)\|\left(\lambda\|(\Delta\mu, \Delta\Sigma)\| + \lambda^*\|(\Delta\mu^*, \Delta\Sigma^*)\|\right)$$

we can verify that $1/2 \leq \mathcal{D}(G, G_*)/\overline{\mathcal{D}}(G, G_*) \leq 2$, i.e., $\mathcal{D}(G, G_*) \asymp \overline{\mathcal{D}}(G, G_*)$. The reason that we prefer to use the formation of $\mathcal{D}(G, G_*)$ over that of $\overline{\mathcal{D}}(G, G_*)$ is not only due to the convenience of the proof argument of Theorem 3.5 later in Appendix C but also due to its partial connection with Wasserstein metric that we are going to discuss in the next remark.

**(ii)** When $f$ is a multivariate location family and is identical to $h_0$, i.e., $\mu_0 = \mathbf{0}$, it was demonstrated recently in [16] that

$$V(p_G, p_{G_*}) \gtrsim |\lambda - \lambda^*| \|\mu\| \|\mu^*\| + (\lambda^* \|\mu^*\| + \lambda \|\mu\|) \|\mu - \mu^*\|, \tag{5}$$

which is also the key result for establishing the convergence rates of parameter estimation in their work. However, their proof technique only works for the location family and it is unclear what is the sufficient condition for the family of density functions beyond the location family such that the inequality (5) will hold. As the location family is strongly identifiable in the second order, we can verify that the lower bound in Theorem 3.5 and inequality (5) are in fact similar. Therefore, the result in Theorem 3.5 gives a generalization of inequality (5) in [16] under the strongly identifiable in the second order setting of $f$.

**(iii)** As indicated in [16], we can further lower bound the right-hand side of inequality (5) in terms of the second order Wasserstein metric $W_2$ [35] between $G$ and $G_*$ when we present $G$ and $G_*$ as two discrete probability measures with two components. In particular, with an abuse of the notations we denote that $G = (1 - \lambda)\delta_{(\mu_0, \Sigma_0)} + \lambda \delta_{(\mu, \Sigma)}$ and $G^* = (1 - \lambda)\delta_{(\mu_0, \Sigma_0)} + \lambda \delta_{(\mu^*, \Sigma^*)}$, i.e., we think of $G$ and $G_*$ as two mixing measures with one fixed atom to be $(\mu_0, \Sigma_0)$. In light of Lemma E.1 in Appendix E, we have

$$W_2^2(G, G_*) \asymp \lambda \|(\Delta\mu, \Delta\Sigma)\|^2 + \lambda^* \|(\Delta\mu^*, \Delta\Sigma^*)\|^2$$
$$- \min\{\lambda, \lambda^*\} \left( \|(\Delta\mu, \Delta\Sigma)\|^2 + \|(\Delta\mu^*, \Delta\Sigma^*)\|^2 \right) + \min\{\lambda, \lambda^*\} \|\|(\mu, \Sigma) - (\mu^*, \Sigma^*)\|^2.$$

Therefore, $\mathcal{D}(G, G_*)$ and $W_2^2(G, G_*)$ share the similar term $\lambda \|(\Delta\mu, \Delta\Sigma)\|^2 + \lambda^* \|(\Delta\mu^*, \Delta\Sigma^*)\|^2 - \min\{\lambda, \lambda^*\} \left( \|(\Delta\mu, \Delta\Sigma)\|^2 + \|(\Delta\mu^*, \Delta\Sigma^*)\|^2 \right)$ in their formulations. However, as $\lambda \|(\Delta\mu, \Delta\Sigma)\| + \lambda^* \|(\Delta\mu^*, \Delta\Sigma^*)\| \geq \min\{\lambda, \lambda^*\} \|\|(\mu, \Sigma) - (\mu^*, \Sigma^*)\|$, the remaining term in $\mathcal{D}(G, G_*)$ is stronger than that of $W_2^2(G, G_*)$. Moreover, as $\lambda = \lambda^*$, we further obtain that

$$\mathcal{D}(G, G_*)/W_2^2(G, G_*) \asymp (\|(\Delta\mu, \Delta\Sigma)\| + \|(\Delta\mu^*, \Delta\Sigma^*)\|)/\|(\mu, \Sigma) - (\mu^*, \Sigma^*)\|.$$

Hence, as long as the right-hand side term in the above display goes to $\infty$, i.e., $\|(\Delta\mu + \Delta\mu^*, \Delta\Sigma + \Delta\Sigma^*)\| \to 0$, we have $\mathcal{D}(G, G_*)/W_2^2(G, G_*) \to \infty$. This strong refinement of the Wasserstein metric is due to the special structure of $G$ and $G_*$ as one of their components is always fixed to be $(\mu_0, \Sigma_0)$.

**(iv)** Under the setting when $G_*$ is varied, $d_1 = 1$, and $d_2 = 0$, by means of Fatou's lemma the result from Theorem 4.6 in [19] yields $V(p_G, p_{G_*}) \geq C'.W_3^3(G, G_*)$ if the kernel density function $f$ is 4-strongly identifiable (cf. Definition 2.2 in [19]) and satisfies uniform Lipschitz condition up to the fourth order where $C'$ is some positive constant depending only on $G$ and $G_*$. Since $\mathcal{D}(G, G_*) \gtrsim W_2^2(G, G_*) \geq W_1^2(G, G_*) \gtrsim W_3^3(G, G_*)$, it indicates that the bound in Theorem 3.5 is much tighter than this bound. The loss of efficiency in this bound is again due to the special structures of $G$ and $G_*$ as one of their components is always fixed to be $(\mu_0, \Sigma_0)$.

Unlike the convergence rate results from the strongly distinguishable in the first order setting between $f$ and $h_0$ in Theorem 3.3, the convergence rate of $\lambda^*$ under the setting of Theorem 3.5 depends on the rate of convergence of $\|(\Delta\mu^*, \Delta\Sigma^*)\|^2$ to 0 (cf. Theorem A.1). Additionally, the convergence rate of estimating $(\mu^*, \Sigma^*)$ will be determined based on the convergence rates of $\lambda^*$ and $(\Delta\mu^*, \Delta\Sigma^*)$ to 0.

### 3.2.2 Weakly Identifiable Settings

Thus far, as $h_0$ belongs to the family $f$, our results regarding the lower bounds between $p_G$ and $p_{G_0}$ under Total Variation distance rely on the strongly identifiable in the second order assumption of kernel $f$. However, there are various families of density functions that do not satisfy such an assumption, which we refer to as the weakly identifiable condition. To illustrate the non-uniform natures of $V(p_G, p_{G_*})$ under the weakly identifiable condition of $f$, we consider specifically a popular setting of $f$ in this section: multivariate location-covariance Gaussian kernel.

**Location-covariance multivariate Gaussian kernel:** As indicated in the previous work in the literature [7, 25, 21], if $f$ is a family of multivariate location-covariance Gaussian distributions in $d$ dimension, it exhibits the *heat partial differential equation (PDE)* with respect to the location and covariance parameter $\dfrac{\partial^2 f}{\partial \mu \partial \mu^\top}(x|\mu, \Sigma) = 2\dfrac{\partial f}{\partial \Sigma}(x|\mu, \Sigma)$, for any $x \in \mathbb{R}^d$ and $(\mu, \Sigma) \in \Theta \times \Omega$. We can verify that this structure leads to the loss of the second-order strong identifiability condition of the Gaussian kernel. Note that, the PDE structure of the Gaussian kernel has been shown to lead to very slow convergence rates of parameter estimation under general over-fitted Gaussian mixture models (cf. Theorem 1.1 in [21]). For the setting of the multivariate deviated model, since the parameters $\lambda^*$ and $(\mu^*, \Sigma^*)$ are allowed to vary with the sample size, we may expect that the estimation of these parameters will also suffer from the very slow rate. In fact, we achieve the following lower bound of $V(p_G, p_{G_*})$ under the multivariate location-covariance Gaussian kernel.

**Theorem 3.6.** *Assume that $h_0(x) = f(x|\mu_0, \Sigma_0)$ for some $(\mu_0, \Sigma_0) \in \Theta \times \Sigma$ and $f$ is a family of multivariate location-covariance Gaussian distributions. We denote*

$$
\begin{aligned}
\mathcal{Q}(G, G_*) := {} & \lambda(\|\Delta\mu\|^4 + \|\Delta\Sigma\|^2) + \lambda^*(\|\Delta\mu^*\|^4 + \|\Delta\Sigma^*\|^2) \\
& - \min\{\lambda, \lambda^*\}\left(\|\Delta\mu\|^4 + \|\Delta\Sigma\|^2 + \|\Delta\mu^*\|^4 + \|\Delta\Sigma^*\|^2\right) \\
& + \left(\lambda(\|\Delta\mu\|^2 + \|\Delta\Sigma\|) + \lambda^*(\|\Delta\mu^*\|^2 + \|\Delta\Sigma^*\|)\right)\left(\|\mu - \mu^*\|^2 + \|\Sigma - \Sigma^*\|\right),
\end{aligned}
$$

*for any $G$ and $G_*$. Then, we can find a positive constant $C$ depending only on $\Theta$, $\Omega$, and $(\mu_0, \Sigma_0)$ such that $V(p_G, p_{G_*}) \geq C.\mathcal{Q}(G, G_*)$, for any $G$ and $G_*$.*

See Appendix C.3 for the proof of Theorem 3.6. A few comments with Theorem 3.6 are in order.

**(i)** Different from the formulation of $\mathcal{D}(G, G_*)$ in Theorem 3.5 where we have the same power between $\mu$ and $\Sigma$, there is a mismatch of power between $\|\Delta\mu\|^2, \|\Delta\mu^*\|^2$ and $\|\Delta\Sigma\|, \|\Delta\Sigma^*\|$ in the formulation of $\mathcal{Q}(G, G_*)$. This interesting phenomenon is mainly due to the structure of the heat equation where the second-order derivative of the location parameter and the first-order derivative of the covariance parameter is linearly dependent.

**(ii)** If we denote $\mathcal{Q}'(G, G_*) := \lambda(\|\Delta\mu\|^4 + \|\Delta\Sigma\|^2) + \min\{\lambda, \lambda^*\}\left(\|\mu - \mu^*\|^4 + \|\Sigma - \Sigma^*\|^2\right) + \lambda^*(\|\Delta\mu^*\|^4 + \|\Delta\Sigma^*\|^2) - \min\{\lambda, \lambda^*\}\left(\|\Delta\mu\|^4 + \|\Delta\Sigma\|^2 + \|\Delta\mu^*\|^4 + \|\Delta\Sigma^*\|^2\right)$, then we can verify that $\mathcal{Q}(G, G_*) \gtrsim \mathcal{Q}'(G, G_*)$ for any $G, G_*$. If we treat $G$ and $G_*$ as two-components measures as in the remark (iii) after Theorem 3.5, we would have

$$
\mathcal{Q}'(G, G_*) \asymp W_4^4(G_1, G_{1,*}) + W_2^2(G_2, G_{2,*}), \tag{6}
$$

where $G_1 = (1 - \lambda)\delta_{\mu_0'} + \lambda\delta_\mu$, $G_2 = (1 - \lambda)\delta_{\Sigma_0'} + \lambda\delta_\Sigma$ and similarly for $G_{1,*}$ and $G_{2,*}$. Here, $(\mu_0, \Sigma_0) = (\mu_0', \Sigma_0')$, and $W_2, W_4$ are respectively second and fourth order Wasserstein metrics. The formulations of $\mathcal{Q}'(G, G_*)$, therefore, can be thought of as a combination of two Wasserstein metrics: one is with only parameter $\mu$ and another one is only with parameter $\Sigma$. The division into two Wasserstein metrics can be traced back again to the PDE structure of the heat equation.

If $\lambda = \lambda^*$ and $\left(\|\Delta\mu\|^2 + \|\Delta\mu^*\|^2 + \|\Delta\Sigma\| + \|\Delta\Sigma^*\|\right)/\left(\|\mu - \mu^*\|^2 + \|\Sigma - \Sigma^*\|\right) \to \infty$, we will have that $\mathcal{Q}(G, G_*)/\mathcal{Q}'(G, G_*) \to \infty$. It proves that the result from Theorem 3.6 under the multivariate setting of Gaussian kernel is a strong refinement of the summation of Wasserstein metrics regarding location and covariance parameter in equation (6).

A consequence of Theorem 3.6 is that the convergence rate of estimating $\lambda^*$ is determined by $\|\Delta\mu^*\|^4 + \|\Delta\Sigma^*\|^2$, instead of $\|(\Delta\mu^*, \Delta\Sigma^*)\|^2$ as in the strongly identifiable setting of $f$. Furthermore, we also encounter a phenomenon that the rate of convergence of estimating $\Sigma^*$ is much faster than that of estimating $\mu^*$. In particular, estimating $\Sigma^*$ depends on the rate in which $\lambda^*(\|\mu^*\|^2 + \|\Sigma^*\|)$ converges to 0 while estimating $\mu^*$ relies on square root of this rate (cf. Theorem A.2).

# 4 Minimax Lower Bounds and Convergence Rates of Parameter Estimation

In this section, we study the convergence rates of MLE $\widehat{G}_n$ as well as minimax lower bounds of estimating $G_*$ under various settings of $h_0$ and $f$. Due to space constraints, we present the theory in the distinguishable regime of $h_0$ and $f$. Non-distinguishable cases, though more interesting, are deferred to Appendix A.

**Theorem 4.1. (Distinguishable settings)** *Assume that classes of densities $h_0$ and $f$ satisfy the conditions in Theorem 3.3. Then, we achieve that*

*(a) (Minimax lower bound) Assume that $f$ satisfies the following assumption S.1:*

*(S.1)* $\displaystyle \sup_{\|(\mu,\Sigma)-(\mu',\Sigma')\| \leq c_0} \int \frac{\left( \partial^{|\alpha|} f(x|\mu,\Sigma)/\partial\mu^{\alpha_1}\partial\Sigma^{\alpha_2} \right)^2}{f(x|\mu',\Sigma')} dx < \infty$ *for some sufficiently small $c_0 >$*
*0, where $\alpha_1 \in \mathbb{N}^{d_1}, \alpha_2 \in \mathbb{N}^{d_2}$ in the partial derivative of $f$ take any combination such that $|\alpha| = |\alpha_1| + |\alpha_2| \leq 1$.*

*Then for any $r < 1$, there exist two universal positive constants $c_1$ and $c_2$ such that*

$$\inf_{\widehat{G}_n \in \Xi} \sup_{G \in \Xi} \mathbb{E}_{p_G} \left( \lambda^2 \|(\widehat{\mu}_n, \widehat{\Sigma}_n) - (\mu, \Sigma)\|^2 \right) \geq c_1 n^{-1/r},$$

$$\inf_{\widehat{G}_n \in \Xi} \sup_{G \in \Xi} \mathbb{E}_{p_G} \left( |\widehat{\lambda}_n - \lambda|^2 \right) \geq c_2 n^{-1/r}.$$

*Here, the infimum is taken over all sequences of estimates $\widehat{G}_n = (\widehat{\lambda}_n, \widehat{\mu}_n, \widehat{\Sigma}_n)$.*

*(b) (MLE rate) Let $\widehat{G}_n$ be the MLE defined in equation (3), and the family $\{p_G : G \in \Xi\}$ satisfies condition A2. Then, we have the convergence rate for the MLE:*

$$\sup_{G_* \in \Xi} \mathbb{E}_{p_{G_*}} \left( (\lambda^*)^2 \|(\widehat{\mu}_n, \widehat{\Sigma}_n) - (\mu^*, \Sigma^*)\|^2 \right) \lesssim \frac{\log^2 n}{n},$$

$$\sup_{G_* \in \Xi} \mathbb{E}_{p_{G_*}} \left( |\widehat{\lambda}_n - \lambda^*|^2 \right) \lesssim \frac{\log^2 n}{n}.$$

Proof of Theorem 4.1 is in Appendix D.1. The results of Theorem 4.1 imply that even though we still can estimate $\lambda^*$ at the standard rate $n^{-1/2}$, the convergence rate of $(\widehat{\mu}_n, \widehat{\Sigma}_n)$ to $(\mu^*, \Sigma^*)$ strictly depends on the vanishing rate of $\lambda^*$ to 0. Therefore, the convergence rate of estimating $(\mu^*, \Sigma^*)$ can be generally slower than $n^{-1/2}$ as long as $\lambda^*$ goes to 0 at a rate slower than $n^{-1/2}$.

We can also use the geometric inequalities developed in Section 3.2 to investigate the behaviors of $\widehat{G}_n$ in the non-distinguishable settings. We will further see how the *non-identifiability* and *singularity* of the model affect the convergence rate for density estimation. Due to space constraints, the results for this setting are presented in Appendix A.

# 5 Experiments

We now demonstrate the convergence rates of parameter estimation in the strongly distinguishable setting, where the choice of $h_0$ is a standard Cauchy distribution, and $f(\cdot|\mu, \sigma^2)$ is the normal distribution with mean $\mu$ and variance $\sigma^2$. The additional experiments with the non-distinguishable setting are deferred to Appendix F.

Assume that $X_1, \ldots, X_n$ are i.i.d. samples drawn from the true density function $p_{G_*}$ with $G_* = (\lambda^*, \mu^*, (\sigma^*)^2)$ and we obtain the MLE $(\hat{\lambda}_n, \hat{\mu}_n, \hat{\sigma}_n^2)$. We consider two following cases:

**(i)** $\lambda^* = 0.5, \mu^* = 2.5, (\sigma^*)^2 = 0.25$;

**(ii)** $\lambda^* = 0.5/n^{1/4}, \mu^* = 2.5, (\sigma^*)^2 = 0.25$.

Two histograms for samples from the density $p_{G_*}$ with $n = 10000$ corresponding to the above two cases are illustrated in Figure 1(a) and 2(a). For each case, we take into account multiple sample

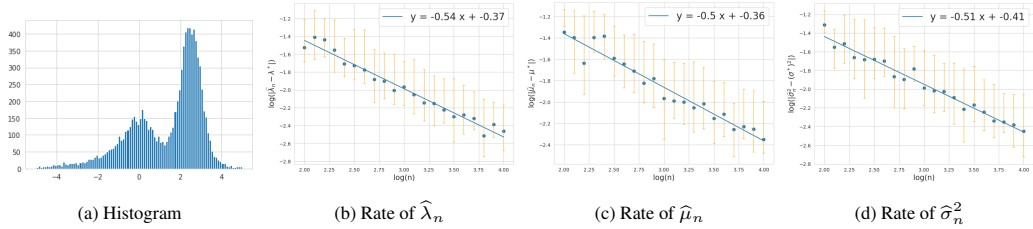

(a) Histogram  (b) Rate of $\widehat{\lambda}_n$  (c) Rate of $\widehat{\mu}_n$  (d) Rate of $\widehat{\sigma}_n^2$

Figure 1: Case (i) $\lambda^* = 0.5$.

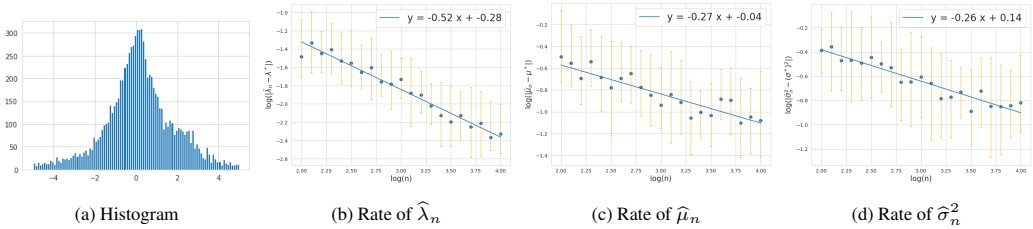

(a) Histogram  (b) Rate of $\widehat{\lambda}_n$  (c) Rate of $\widehat{\mu}_n$  (d) Rate of $\widehat{\sigma}_n^2$

Figure 2: Case (ii) $\lambda^* = 0.5/n^{1/4}$.

sizes $n$ ranging from $10^2$ to $10^4$. For each sample size $n$, we calculate the MLE $(\hat{\lambda}_n, \hat{\mu}_n, \hat{\sigma}_n^2)$ via the EM algorithm [13] and measure the errors $|\widehat{\lambda}_n - \lambda^*|$, $|\widehat{\mu}_n - \mu^*|$, and $|\widehat{\sigma}_n^2 - (\sigma^*)^2|$. We repeat this procedure 64 times and plot the mean (blue dot) and quartile error bars (yellow bar) of the logarithm of estimation errors against the log of $n$. Theorem 4.1 suggests that the log convergence rate of $\lambda^*$ is of order $-1/2$ for all cases, and so is the rate for $(\mu, (\sigma)^2)$ in the first case. Meanwhile, the convergence rates of $(\mu^*, (\sigma^*)^2)$ in the second case are slower, which is in the order of $-1/4$. The empirical rates found in the experiments match this theoretical result, where the least square line shows that the logarithm of the rate for estimating $\mu^*$ in case (i) is -0.5 and that of the case (ii) is -0.27 $\approx -1/4$ (similar for $(\sigma^*)^2$).

We once again emphasize that this interesting phenomenon of the rates of convergence is due to the *singularity* and *identifiability* of the multivariate deviated model. Our theory and simulation have accurately shown quantitative convergence rates for parameter estimation when $\lambda^*$ near the singularity point 0, where all pairs of $(\mu^*, \Sigma^*)$ in model (2) give the same model. Together with the non-distinguishable settings, we provide a comprehensive study of the large-sample theory for this type of model, thanks to the newly developed notion of distinguishability that helps to control the linear independent relation between $h_0$ and $f$. The developed optimal minimax lower bounds and convergence rates will certainly help Machine Learning practitioners understand better the role of sample sizes in the accuracy of estimation in the multivariate deviated model, and are also inspired theorists to study more about the estimation rate of complex hierarchical/mixture models.

## 6  Conclusion

In this paper, we establish the uniform rate for estimating true parameters in the multivariate deviated model by using the maximum likelihood estimation (MLE) method. During our derivation, we have to overcome two major obstacles, which are firstly the interaction between the known function $h_0$ and the Gaussian density $f$, and secondly the likelihood of the deviated proportion $\lambda^*$ vanishing to either one or zero. To this end, we introduce a notion of distinguishability to control the linearly independent relation between two functions $h_0$ and $f$. Finally, we achieve the optimal convergence rate of the MLE under both the distinguishable and non-distinguishable settings.

## Acknowledgements

NH acknowledges support from the NSF IFML 2019844 and the NSF AI Institute for Foundations of Machine Learning.

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

# Supplement to "Minimax Optimal Rate for Parameter Estimation in Multivariate Deviated Models"

In this supplementary material, we present additional results and proofs. The minimax lower bounds and convergence rates of parameter estimation in the non-distinguishable settings are presented in Section A. The general theory for the convergence rates of densities and their proofs can be found in Appendix B. Proofs of the geometric inequalities that relate the convergence of density estimation to that of parameter estimation are in Section C, while those for minimax lower bounds and convergence rates of parameter estimation are left in Appendix D. Then, we provide a necessary lemma for those results along with its proof in Appendix E. Finally, some discussion about the general setting of the paper is presented in Section F, followed by a set of simulations to support the developed theory.

## A  Minimax Lower Bounds and Convergence Rates of Parameter Estimation under the Non-distinguishable Settings

**Theorem A.1.  (Strongly identifiable and non-distinguishable settings)** *Assume that classes of densities $h_0$ and $f$ satisfy the conditions in Theorem 3.5. We define*

$$
\Xi_1(l_n) := \left\{ G = (\lambda, \mu, \Sigma) \in \Xi : \frac{l_n}{\min\limits_{\substack{1 \le i \le d_1 \\ 1 \le u,v \le d_2}} \{|(\Delta\mu)_i|^2, |(\Delta\Sigma)_{uv}|^2\}\sqrt{n}} \le \lambda \right\},
$$

*for any sequence $\{l_n\}$. Then, we achieve*

*(a) (Minimax lower bound) Assume that $f$ satisfies assumption S.1 in Theorem 4.1. Then for any $r < 1$ and sequence $\{l_n\}$, there exist two universal positive constants $c_1$ and $c_2$ such that*

$$
\inf_{\widehat{G}_n \in \Xi} \sup_{G \in \Xi_1(l_n)} \mathbb{E}_{p_G}\left( \lambda^2 \|(\Delta\mu, \Delta\Sigma)\|^2 \|(\widehat{\mu}_n, \widehat{\Sigma}_n) - (\mu, \Sigma)\|^2 \right) \ge c_1 n^{-1/r},
$$

$$
\inf_{\widehat{G}_n \in \Xi} \sup_{G \in \Xi_1(l_n)} \mathbb{E}_{p_G}\left( \|\Delta\mu, \Delta\Sigma)\|^4 |\widehat{\lambda}_n - \lambda|^2 \right) \ge c_2 n^{-1/r}.
$$

*(b) (MLE rate) Let $\widehat{G}_n$ be the MLE defined in equation (3), and the family $\{p_G : G \in \Xi\}$ satisfies condition A2. Then, for any sequence $\{l_n\}$ such that $l_n/\log n \to \infty$,*

$$
\sup_{G_* \in \Xi_1(l_n)} \mathbb{E}_{p_{G_*}}\left( (\lambda^*)^2 \|(\Delta\mu^*, \Delta\Sigma^*)\|^2 \|(\widehat{\mu}_n, \widehat{\Sigma}_n) - (\mu^*, \Sigma^*)\|^2 \right) \lesssim \frac{\log^2 n}{n},
$$

$$
\sup_{G_* \in \Xi_1(l_n)} \mathbb{E}_{p_{G_*}}\left( \|(\Delta\mu^*, \Delta\Sigma^*)\|^4 |\widehat{\lambda}_n - \lambda^*|^2 \right) \lesssim \frac{\log^2 n}{n}.
$$

Proof of Theorem A.1 is in Appendix D.2. The results of part (b) are the generalization of those in Theorem 3.1 and Theorem 3.2 in [16] to the setting of strongly identifiable in the second-order kernel. The condition regarding the lower bound of $\lambda$ in the formation of $\Xi_1(l_n)$ is necessary to guarantee that $(\widehat{\mu}_n, \widehat{\Sigma}_n)$ and $\widehat{\lambda}_n$ are consistent estimators of $(\mu^*, \Sigma^*)$ and $\lambda^*$ respectively. In particular, from the results in equation (34) of the proof of Theorem A.1, we have for any $G_* \in \Xi$ that

$$
\mathbb{E}_{p_{G_*}}(\lambda^*)^2 \|(\Delta\mu^*, \Delta\Sigma^*)\|^2 \|(\widehat{\mu}_n, \widehat{\Sigma}_n) - (\mu^*, \Sigma^*)\|^2 \lesssim \frac{\log^2 n}{n}.
$$

Therefore, for any $1 \le i \le d_1$ and $1 \le u, v \le d_2$ we get

$$
\mathbb{E}_{p_{G_*}}\left\{ \left( \frac{(\Delta\widehat{\mu}_n)_i}{(\Delta\mu^*)_i} - 1 \right)^2 \right\} \lesssim \frac{\log^2 n}{n(\lambda^*)^2 \{(\Delta\mu^*)_i\}^4},
$$

$$
\mathbb{E}_{p_{G_*}}\left\{ \left( \frac{(\Delta\widehat{\Sigma}_n)_{uv}}{(\Delta\Sigma^*)_{uv}} - 1 \right)^2 \right\} \lesssim \frac{\log^2 n}{n(\lambda^*)^2 \{(\Delta\Sigma^*)_{uv}\}^4}.
$$

It indicates that

$$\frac{\log n}{\sqrt{n}\lambda^* \min\limits_{1 \le i \le d_1, 1 \le u,v \le d_2} \{|(\Delta\mu^*)_i|^2, |(\Delta\Sigma^*)_i|^2\}} \to 0$$

for the left-hand-side terms of the above display to go to 0 for all $1 \le i \le d_1$ and $1 \le u, v \le d_2$.

The results of Theorem A.1 imply that as long as the kernel functions are strongly identifiable in the second order, the convergence rates of $\widehat{\mu}_n$ to $\mu^*$ and $\widehat{\Sigma}_n$ to $\Sigma^*$ are similar, which depend on the vanishing rate of $(\lambda^*)^2\|(\Delta\mu^*, \Delta\Sigma^*)\|^2$ to 0. In our next result of location-covariance multivariate Gaussian distribution, we will demonstrate that such uniform convergence rates of different parameters no longer hold.

**Theorem A.2. (Weakly identifiable and non-distinguishable settings)** *Assume that $f$ is a family of location-covariance multivariate Gaussian distributions, and $h_0(x) = f(x|\mu_0, \Sigma_0)$ for some $(\mu_0, \Sigma_0) \in \Theta \times \Sigma$. We define*

$$\Xi_2(l_n) := \left\{ G = (\lambda, \mu, \Sigma) \in \Xi : \frac{l_n}{\min\limits_{1 \le i \le d, 1 \le u,v \le d}\{|(\Delta\mu)_i|^4, |(\Delta\Sigma)_{uv}|^2\}\sqrt{n}} \le \lambda \right\},$$

*for any sequence $\{l_n\}$. Then, the following holds:*

*(a) (Minimax lower bound) For any $r < 1$ and sequence $\{l_n\}$, there exist two universal positive constants $c_1$ and $c_2$ such that*

$$\inf_{\widehat{G}_n \in \Xi} \sup_{G \in \Xi_2(l_n)} \mathbb{E}_{p_G}\left( \lambda^2 \left\{\|\Delta\mu\|^4 + \|\Delta\Sigma\|^2\right\} \left\{\|\widehat{\mu}_n - \mu\|^4 + \|\widehat{\Sigma}_n - \Sigma\|^2\right\} \right) \ge c_1 n^{-1/r},$$

$$\inf_{\widehat{G}_n \in \Xi} \sup_{G \in \Xi_2(l_n)} \mathbb{E}_{p_G}\left( \left\{\|\Delta\mu\|^8 + \|\Delta\Sigma\|^4\right\}|\widehat{\lambda} - \lambda|^2 \right) \ge c_2 n^{-1/r}.$$

*(b) (MLE rate) Let $\widehat{G}_n$ be the estimator defined in (3). Then, for any sequence $\{l_n\}$ such that $l_n/\log n \to \infty$ the following holds*

$$\sup_{G_* \in \Xi_2(l_n)} \mathbb{E}_{p_{G*}}\left( (\lambda^*)^2 \left\{\|\Delta\mu^*\|^4 + \|\Delta\Sigma^*\|^2\right\} \left\{\|\widehat{\mu}_n - \mu^*\|^4 + \|\widehat{\Sigma}_n - \Sigma^*\|^2\right\} \right) \lesssim \frac{\log^2 n}{n},$$

$$\sup_{G_* \in \Xi_2(l_n)} \mathbb{E}_{p_{G*}}\left( \left\{\|\Delta\mu^*\|^8 + \|\Delta\Sigma^*\|^4\right\}|\widehat{\lambda}_n - \lambda^*|^2 \right) \lesssim \frac{\log^2 n}{n}.$$

Proof of Theorem A.2 is in Appendix D.3. A few comments are in order:

**(i)** Similar to the argument after Theorem A.1, the condition regarding $\lambda$ in the formulation of $\Xi_2(l_n)$ is to guarantee that $(\widehat{\mu}_n, \widehat{\Sigma}_n)$ and $\widehat{\lambda}_n$ are consistent estimators of $(\mu^*, \Sigma^*)$ and $\lambda^*$, respectively.

**(ii)** The results of part (b) indicate that the convergence rate of estimating $\Sigma^*$ is generally much faster than that of estimating $\mu^*$ regardless of the circumstance of $(\lambda^*)^2\left\{\|\Delta\mu^*\|^4 + \|\Delta\Sigma^*\|^2\right\}$. The non-uniformity of these convergence rates is mainly due to the structure of the heat partial differential equation, where the second-order derivative of the location parameter and the first-order derivative of covariance parameter correlate.

**(iii)** From the results of part (b), it is clear that when $\|\Delta\mu^*\| + \|\Delta\Sigma^*\| \not\to 0$, i.e., $(\mu^*, \Sigma^*) \to (\overline{\mu}, \overline{\Sigma}) \ne (\mu_0, \Sigma_0)$, and $\lambda^* \not\to 0$, the convergence rate of $\widehat{\lambda}_n$ to $\lambda^*$ is $n^{-1/2}$. Furthermore, by using the result from part (a) of Proposition C.4 we can verify that

$$\sup_{G_*} \mathbb{E}_{p_{G*}}\left( (\lambda^*)^2 \left\{\|\widehat{\mu}_n - \mu^*\|^2 + \|\widehat{\Sigma}_n - \Sigma^*\|^2\right\} \right) \lesssim \frac{\log^2 n}{n},$$

where the supremum is taken over $\{G_* \in \Xi_2(l_n) : \mathcal{K}(G_*, \overline{G}) \le \epsilon\}$, and $\overline{G} = (\overline{\lambda}, \overline{\mu}, \overline{\Sigma})$, $\lambda^* \to \overline{\lambda}$, and $\epsilon$ is some sufficiently small positive constant. Since $\overline{\lambda} \ne 0$, we achieve the optimal convergence rate $n^{-1/2}$ of estimating $(\mu^*, \Sigma^*)$ within a sufficiently small neighborhood of $\overline{G}$ under metric $\mathcal{K}$. These results imply that even though the convergence rate of estimating $G_*$ may be extremely slow

when $G_*$ moves over the whole space $\Xi_2(l_n)$ (global convergence), such convergence rate can be at standard rate $n^{-1/2}$ when $G_*$ moves within a sufficiently small neighborhood of some appropriate parameters $\overline{G}$ (local convergence).

As we have seen from the convergence rate results from location-covariance multivariate Gaussian distributions, the heat PDE structure plays a key role in the slow convergence rates of location and covariance parameters as well as the mismatch of orders of these rates.

## B  Convergence Rate of Density Estimation

### B.1  General Theory and Proof of Theorem 2.3

We now describe the convergence rate of density estimation under the Hellinger distance in detail and give a general result for the multivariate deviated model. We recall some popular notions in Empirical Process theory as follows. An $\epsilon-$net for a metric space $(\mathcal{P}, d)$ is a collection of balls with radius $\epsilon$ (with respect to metric $d$) having union contains $\mathcal{P}$. The minimal cardinality of such $\epsilon-$nets is called the covering number and denoted by $N(\epsilon, \mathcal{P}, d)$. The logarithm of $N(\epsilon, \mathcal{P}, d)$ is called the entropy number and is denoted by $H(\epsilon, \mathcal{P}, d)$. The bracketing number $N_B(\epsilon, \mathcal{P}, d)$ is the minimal number $n$ such that there exists $n$ pairs $(\underline{f}_i, \overline{f}_i)_{i=1}^n$ such that $\underline{f}_i < \overline{f}_i$, $d(\underline{f}_i, \overline{f}_i) < \epsilon$, and their union covers $\mathcal{P}$. The logarithm of $N_B(\epsilon, \mathcal{P}, d)$ is called the bracketing entropy number and is denoted by $H_B(\epsilon, \mathcal{P}, d)$. In the following discussion, if $\mathcal{P}$ is a family of density and we omit $d$, we understand that $d$ is the distance associated with $L^2(m)$, where $m$ is the Lebesgue measure.

Denote $\mathcal{P}(\Xi) = \{p_G : G \in \Xi\}$ and $\overline{\mathcal{P}}(\Xi) = \{(p_{G_*} + p_G)/2 : G \in \Xi\}$ for the fixed true parameter $G_*$. The convergence rate can be deduced from the complexity of the set:

$$\overline{\mathcal{P}}^{1/2}(\Xi, \epsilon) = \left\{ \bar{p}_G^{1/2} : G \in \Xi, \ h(\bar{p}_G, p_{G_*}) \leq \epsilon \right\}, \tag{7}$$

where for any $G \in \Xi$, we denote $\bar{p}_G := (p_G + p_{G_*})/2$. We measure the complexity of this class through the bracketing entropy integral

$$\mathcal{J}_B(\epsilon, \overline{\mathcal{P}}^{1/2}(\Xi, \epsilon)) = \int_{\epsilon^2/2^{13}}^{\epsilon} H_B^{1/2}(u, \overline{\mathcal{P}}^{1/2}(\Xi, \epsilon)) du \vee \epsilon, \tag{8}$$

where $H_B(\epsilon, \mathcal{P})$ denotes the $\epsilon$-bracketing entropy number of a metric space $\mathcal{P}$. We recall assumption A2:

A2.  Given a universal constant $J > 0$, there exists $N > 0$, possibly depending on $\Theta$ and $k$, such that for all $n \geq N$ and all $\epsilon > (\log n/n)^{1/2}$,

$$\mathcal{J}_B(\epsilon, \overline{\mathcal{P}}^{1/2}(\Xi, \epsilon)) \leq J \sqrt{n} \epsilon^2.$$

**Theorem B.1.** *Assume that Assumption A2 holds, and let $k \geq 1$. Then, there exists a constant $C > 0$ depending only on $\Theta$ and $k$ such that for all $n \geq 1$,*

$$\sup_{G_* \in \Xi} \mathbb{E}_{p_{G_*}} h(p_{\widehat{G}_n}, p_{G_*}) \leq C \sqrt{\log n/n}.$$

This result can be obtained by modifying the proof of Theorem 7.4 in [33]. Recall that we defined the function class

$$\overline{\mathcal{P}}^{1/2}(\Xi, \epsilon) = \left\{ \bar{p}_G^{1/2} : G \in \Xi, \ h(\bar{p}_G, p_{G_*}) \leq \epsilon \right\}, \tag{9}$$

where for any $G \in \Xi$, we write $\bar{p}_G = (p_G + p_{G_*})/2$, and measure the complexity of this class through the bracketing entropy integral

$$\mathcal{J}_B(\epsilon, \overline{\mathcal{P}}^{1/2}(\Xi, \epsilon), \nu) = \int_{\epsilon^2/2^{13}}^{\epsilon} \sqrt{\log N_B(u, \overline{\mathcal{P}}^{1/2}(\Xi, u), \nu)} du \vee \epsilon,$$

where $N_B(\epsilon, X, \eta)$ denotes the $\epsilon$-bracketing number of a metric space $(X, \eta)$ and $\nu$ is the Lebesgue measure. We denote by $P_G$ the distribution corresponding to the density $p_G$. The technique to prove this theorem is to bound the convergence rate by the increments of an empirical process:

$$\nu_n(G) = \sqrt{n} \int_{\{p_{G_*} > 0\}} \frac{1}{2} \log \frac{\bar{p}_G}{p_{G_*}} d(P_n - P_{G_*}),$$

where $P_n = \frac{1}{n} \sum_{i=1}^{n} \delta_{X_i}$ is the empirical measure $(X_1, \ldots, X_n \overset{iid}{\sim} p_{G_*})$. We first recall Theorem 5.11 in [33] with the notations adapted from our setting:

**Theorem B.2.** *Let $R > 0$, $k \geq 1$, and $\mathcal{G}$ be a subset of $\Xi$, which contains $G_*$. Given $C_1 < \infty$, for all $C$ sufficiently large, and for $n \in \mathbb{N}$ and $t > 0$ satisfying*

$$t \leq \sqrt{n}((8R) \wedge (C_1 R^2)), \tag{10}$$

*and*

$$t \geq C^2(C_1 + 1)\left(R \vee \int_{t/(2^6\sqrt{n})}^{R} H_B^{1/2}\left(\frac{u}{\sqrt{2}}, \overline{\mathcal{P}}^{1/2}(\Xi, R), \nu\right) du\right), \tag{11}$$

*we have*

$$\mathbb{P}_{\lambda^* G_*}\left(\sup_{G \in \mathcal{G}, h(\overline{p}_G, p_{G_*}) \leq R} |\nu_n(G)| \geq t\right) \leq C \exp\left(-\frac{t^2}{C^2(C_1 + 1)R^2}\right). \tag{12}$$

Now we proceed to prove Theorem 2.3, the proof is divided into three parts: Bounding the tail probability of $h(p_{\hat{G}_n}, p_{G_*})$ by sums of empirical processes increments using the chaining technique, bounding the empirical processes increments using Theorem B.2, and bounding the expectation of $h(p_{\hat{G}_n}, p_{G_*})$ using its tail probability.

**Step 1 (Bounding the tail probability $h(p_{\hat{G}_n}, p_{G_*})$ by sums of empirical processes increments):** Firstly, by Lemma 4.1 and 4.2 of [33], we have

$$\frac{1}{16}h^2(p_{\hat{G}_n}, p_{G_*}) \leq h^2(\overline{p}_{\hat{G}_n}, p_{G_*}) \leq \frac{1}{\sqrt{n}}\nu_n(\hat{G}_n).$$

Hence, for any $\delta > \delta_n := (\log n/n)^{1/2}$, we have

$$\mathbb{P}_{G_*}(h(p_{\hat{G}_n}, p_{G_*}) \geq \delta) \leq \mathbb{P}_{G_*}\left(\nu_n(\hat{\lambda}_n \hat{G}_n) - \sqrt{n}h^2(\overline{p}_{\hat{G}_n}, p_{G_*}) \geq 0,\right.$$

$$\left. h(\overline{p}_{\hat{G}_n}, p_{G_*}) \geq \delta/4\right)$$

$$\leq \mathbb{P}_{G_*}\left(\sup_{G: h(\overline{p}_G, p_{G_*}) \geq \delta/4}[\nu_n(G) - \sqrt{n}h^2(\overline{p}_G, p_{G_*})] \geq 0\right)$$

$$\leq \sum_{s=0}^{S} \mathbb{P}_{G_*}\left(\sup_{G: 2^s\delta/4 \leq h(\overline{p}_G, p_{G_*}) \leq 2^{s+1}\delta/4} |\nu_n(G)| \geq \sqrt{n}2^{2s}(\delta/4)^2\right)$$

$$\leq \sum_{s=0}^{S} \mathbb{P}_{G_*}\left(\sup_{G: h(\overline{p}_G, p_{G_*}) \leq 2^{s+1}\delta/4} |\nu_n(G)| \geq \sqrt{n}2^{2s}(\delta/4)^2\right),$$

where $S$ is a smallest number such that $2^S\delta/4 > 1$, as $h(\overline{p}_G, p_{G_*}) \leq 1$. Now we will bound each term above using Theorem B.2.

**Step 2 (Bounding the empirical processes increments using Theorem B.2):** In Theorem B.2, choose $R = 2^{s+1}\delta, C_1 = 15$ and $t = \sqrt{n}2^{2s}(\delta/4)^2$, we can readily check that Condition (10) satisfies (because $2^{s-1}\delta/4 \leq 1$ for all $s = 0, \ldots, S$). Condition (11) satisfies thanks to Assumption A3:

$$\int_{t/(2^6\sqrt{n})}^{R} H_B^{1/2}\left(\frac{u}{\sqrt{2}}, \mathcal{P}^{1/2}(\Xi, R), \nu\right) du \vee 2^{s+1}\delta = \sqrt{2}\int_{R^2/2^{13}}^{R/\sqrt{2}} H_B^{1/2}\left(u, \mathcal{P}^{1/2}(\Xi, R), \nu\right) du \vee 2^{s+1}\delta$$

$$\leq 2\mathcal{J}_B(R, \mathcal{P}^{1/2}(\Xi, R), \nu)$$

$$\leq 2J\sqrt{n}2^{2s+1}\delta^2 = 2^6 Jt.$$

So the conclusion of Theorem B.2 gives us

$$\mathbb{P}_{G_*}(h(p_{\hat{G}_n}, p_{G_*}) > \delta) \leq C \sum_{s=0}^{\infty} \exp\left(\frac{2^{2s}n\delta^2}{J^2 2^{14}}\right) \leq c \exp\left(\frac{n\delta^2}{c^2}\right), \tag{13}$$

where $c$ is a large constants that does not depend on $G_*$.

**Step 3 (Implying the bound on supremum of expectation):** Thus, we have

$$\mathbb{E}h(p_{\hat{G}_n}, p_{G_*}) = \int_0^\infty \mathbb{P}(h(p_{\hat{G}_n}, p_{G_*}) > \delta)d\delta \leq \delta_n + c\int_{\delta_n}^\infty \exp\left(-\frac{n\delta^2}{c^2}\right) \leq \tilde{c}\delta_n,$$

for some $\tilde{c}$ does not depend on $\lambda^*, G_*$. Hence, we finally proved that

$$\sup_{G_* \in \Xi} \mathbb{E}_{G_*} h(p_{\hat{G}_n}, p_{G_*}) \leq C\sqrt{\log n/n}.$$

As a consequence, we obtain the conclusion of the theorem.

## B.2 Proof for Proposition 2.4

We further introduce some more notations that are required for the proof. Let $N(\epsilon, \mathcal{P}(\Xi), \|\cdot\|_\infty)$ be the $\epsilon-$covering number of $(\mathcal{P}(\Xi), \|\cdot\|_\infty)$ and $N_B(\epsilon, \mathcal{P}(\Xi), h)$ be the bracketing number of $\mathcal{P}(\Xi)$ measured by Hellinger metric $h$. $H_B(\epsilon, \mathcal{P}(\Xi), h) = \log N_B(\epsilon, \mathcal{P}(\Xi), h)$ is called the bracketing entropy of $\mathcal{P}(\Xi)$ under metric $h$. We want to show that

$$\mathcal{J}_B(\epsilon, \overline{\mathcal{P}}^{1/2}(\Xi, \epsilon), L^2(m)) = \left(\int_{\epsilon^2/2^{13}}^\epsilon H_B^{1/2}(\delta, \overline{\mathcal{P}}^{1/2}(\Xi, \delta), L^2(m))d\delta \vee \delta\right) \lesssim \sqrt{n}\epsilon^2, \quad (14)$$

for all $n > N$ large enough and $\epsilon > (\log n/n)^{1/2}$. We proceed to show that claim (14) will be proved if

$$\log N(\epsilon, \mathcal{P}(\Xi), \|\cdot\|_\infty) \lesssim \log(1/\epsilon), \quad (15)$$

$$H_B(\epsilon, \mathcal{P}(\Xi), h) \lesssim \log(1/\epsilon), \quad (16)$$

and then prove claim (15) and (16).

**Proof of that claim (16) implies claim (14)** Because $\overline{\mathcal{P}}^{1/2}(\Xi, \delta) \subset \overline{\mathcal{P}}^{1/2}(\Xi)$ and from the definition of Hellinger distance,

$$H_B(\delta, \overline{\mathcal{P}}^{1/2}(\Xi, \delta), \mu) \leq H_B(\delta, \overline{\mathcal{P}}^{1/2}(\Xi), \mu) = H_B(\frac{\delta}{\sqrt{2}}, \overline{\mathcal{P}}(\Xi), h).$$

Now use the fact that for densities $f_*, f_1, f_2$, we have $h^2((f_1 + f_*)/2, (f_2 + f_*)/2) \leq h^2(f_1, f_2)/2$, one can readily check that $H_B(\frac{\delta}{\sqrt{2}}, \overline{\mathcal{P}}(\Xi), h) \leq H_B(\delta, \mathcal{P}(\Xi), h)$. Hence, if claim (16) holds true, then

$$H_B(\delta, \overline{\mathcal{P}}^{1/2}(\Xi, \delta), \mu) \leq H_B(\delta, \mathcal{P}(\Xi), h) \lesssim \log(1/\delta),$$

which implies that

$$\mathcal{J}_B(\epsilon, \overline{\mathcal{P}}^{1/2}(\Xi, \delta), \mu) \lesssim \epsilon(\log(2^{13}/\epsilon^2))^{1/2} < n\epsilon^2,$$

for all $\epsilon > (\log n/n)^{1/2}$. Hence, claim (14) is proved.

**Proof of claim (15)** As $\lambda \in [0, 1]$, we can choose an $\epsilon-$net for it with the cardinality no more than $\frac{1}{\epsilon}$. Similarly, because $\Theta$ and $\Omega$ are compact, we can cover them by hypercube $[-a, a]^{d_1}$ and $[-b, b]^{d_2 \times d_2}$. Hence, there exists $\epsilon-$nets for them with the cardinality no more than $\left(\frac{2a}{\epsilon}\right)^{d_1}$ and $\left(\frac{2b}{\epsilon}\right)^{d_2^2}$. Let $\mathcal{S}$ be the Cartesian product of them. We have $\log|\mathcal{S}| \lesssim \log(1/\epsilon)$ and for every $G = (\lambda, \mu, \Sigma) \in \Xi$, there exists $G' = (\lambda', \mu', \Sigma') \in \mathcal{S}$ such that $|\lambda - \lambda'|, \|\mu - \mu'\|, \|\Sigma - \Sigma'\| \leq \epsilon$. By triangle inequalities,

$$\|p_G - p_{G'}\|_\infty \leq |\lambda - \lambda'|(\|h_0\|_\infty + \|f\|_\infty) + \lambda|f(x|\mu, \Sigma) - f(x|\mu', \Sigma')| \lesssim \epsilon,$$

thanks to the uniform bounded and Lipchitz assumptions. Hence,

$$\log N(\epsilon, \mathcal{P}(\Xi), \|\cdot\|_\infty) \lesssim \log(1/\epsilon).$$

**Proof of claim** (16)  Now, from the entropy number, we are going to bound the bracketing number, we let $\eta \leq \epsilon$ which will be chosen later. Let $f_1, \ldots, f_N$ be a $\eta$-net for $\mathcal{P}(\Xi)$, where $f_i(x) = (1 - \lambda_i)h_0(x) + \lambda_i f(x|\mu_i, \Sigma_i)$. Let

$$H(x) = \begin{cases} b_1 \exp(-b_2 \|x\|^{b_3}), & \|x\|_2 \geq B_1, \\ M, & \text{otherwise} \end{cases} \tag{17}$$

is an envelop for $f(x|\mu, \Sigma)$. We can construct brackets $[p_i^L, p_i^U]$ as follows.

$$p_i^L(x) = (1 - \lambda_i)h_0(x) + \lambda_i \max\{f(x|\mu_i, \Sigma_i) - \eta, 0\},$$
$$p_i^U(x) = (1 - \lambda_i)h_0(x) + \lambda_i \min\{f(x|\mu_i, \Sigma_i) + \eta, H(x)\}.$$

Because for each $f \in \mathcal{P}(\Xi)$, there is $f_i$ such that $\|f - f_i\|_\infty < \eta$, we have $p_i^L \leq f \leq p_i^U$. Moreover, for any $\overline{B} \geq B$,

$$\int_{\mathbb{R}^d} (p_i^U - p_i^L)d\mu \leq \lambda_i \left( \int_{\|x\| \leq \overline{B}} 2\eta dx + \int_{\|x\| \geq \overline{B}} H(x)dx \right)$$
$$\lesssim \eta \overline{B}^d + \overline{B}^d \exp\left( -b_2 \overline{B}^{b_3} \right), \tag{18}$$

where we use spherical coordinate to have

$$\int_{\|x\| \leq \overline{B}} dx = \frac{\pi^{d/2}}{\Gamma(d/2 + 1)}\overline{B}^d \lesssim \overline{B}^d,$$

and

$$\int_{\|x\| \geq \overline{B}} \exp\left( -b_2 \|x\|^{b_3} \right) \lesssim \int_{r \geq \overline{B}} r^{d-1} \exp\left( -b_2 r^{b_3} \right) dr$$
$$= \frac{1}{b_3 b_2^{1/b_3}} \int_{\overline{B}^{b_3}}^\infty u^{d/b_3 - 1} \exp(-u)du \quad \text{(change of variable } u = b_2 r^{b_3})$$
$$\leq \frac{1}{b_3 b_2^{1/b_3}} \overline{B}^{d-b_3} \exp(-\overline{B}^{b_3}).$$

Hence, in (18), if we choose $\overline{B} = B(\log(1/\eta))^{1/b_3}$ then

$$\int_{\mathbb{R}^d} (p_i^U - p_i^L)d\mu \lesssim \eta \left( \log\left(\frac{1}{\eta}\right) \right)^{d/b_3}. \tag{19}$$

Therefore, there exists a positive constant $c$ which does not depend on $\eta$ such that

$$H_B(c\eta \log(1/\eta)^{d/b_3}, \mathcal{P}(\Xi), \|\cdot\|_1) \lesssim \log(1/\eta).$$

Let $\epsilon = c\eta(\log(1/\eta))^{d/b_3}$, we have $\log(1/\epsilon) \asymp \log(1/\eta)$, which combines with inequality $\|\cdot\|_1 \leq h^2$ leads to

$$H_B(\epsilon, \mathcal{P}(\Xi), h) \leq H_B(\epsilon^2, \mathcal{P}(\Xi), \|\cdot\|_1) \lesssim \log(1/\epsilon^2) \lesssim \log(1/\epsilon).$$

Thus, we have proved claim (16).

## C  Proofs for Geometric Inverse Bounds

### C.1  Proof of Theorem 3.3

The second inequality in Theorem 3.3 is straightforward from the equivalent form of $W_1(G, G_*)$ in Lemma E.1 (see Appendix E). Therefore, we will only focus on establishing the first inequality in that theorem. We start with the following key result:

**Proposition C.1.** *Given the assumptions in Theorem 3.5 and $\overline{G} = (\overline{\lambda}, \overline{\mu}, \overline{\Sigma})$ such that $\overline{\lambda} \in [0, 1]$ and $(\overline{\mu}, \overline{\Sigma})$ can be equal to $(\mu_0, \Sigma_0)$. Then, we have*

$$\lim_{\epsilon \to 0} \inf_{G, G_*} \left\{ \frac{V(p_G, p_{G_*})}{\mathcal{K}(G, G_*)} : \mathcal{K}(G, \overline{G}) \vee \mathcal{K}(G_*, \overline{G}) \leq \epsilon \right\} > 0.$$

*Proof.* The high level idea of the proof of Proposition C.3 is to utilize the Taylor expansion techniques previously employed in [9, 28, 22, 19]. Indeed, following Fatou's argument from Theorem 3.1 in [22], to obtain the conclusion of Proposition C.3 it suffices to demonstrate that

$$\lim_{\epsilon \to 0} \inf_{G, G_*} \left\{ \frac{\|p_G - p_{G_*}\|_\infty}{\mathcal{K}(G, G_*)} : \mathcal{K}(G, \overline{G}) \vee \mathcal{K}(G_*, \overline{G}) \leq \epsilon \right\} > 0.$$

Assume that the above conclusion does not hold. It implies that we can find two sequences $G_n = (\lambda_n, \mu_n, \Sigma_n)$ and $G_{*,n} = (\lambda_n^*, \mu_n^*, \Sigma_n^*)$ such that $\mathcal{K}(G_n, \overline{G}) \to 0$, $\mathcal{K}(G_{*,n}, \overline{G}) \to 0$, and $\|p_{G_n} - p_{G_{*,n}}\|_\infty / \mathcal{K}(G_n, G_{*,n}) \to 0$ as $n \to \infty$. Now, we only consider the most challenging setting of $(\mu_n, \Sigma_n)$ and $(\mu_n^*, \Sigma_n^*)$ when they share the same limit point $(\mu', \Sigma')$. The other settings of these two components can be argued in the same fashion. Here, $(\mu', \Sigma')$ is not necessarily equal to $(\mu_0, \Sigma_0)$ or $(\overline{\mu}, \overline{\Sigma})$ as $\lambda_n, \lambda_n^*$ can go to 0 or 1 in the limit. Under that setting, by means of Taylor expansion up to the first order we obtain

$$\frac{p_{G_n}(x) - p_{G_{*,n}}(x)}{\mathcal{K}(G_n, G_{*,n})} = \frac{(\lambda_n^* - \lambda_n)[h_0(x|\mu_0, \Sigma_0) - f(x|\mu_n^*, \Sigma_n^*)] + \lambda_n[f(x|\mu_n, \Sigma_n) - f(x|\mu_n^*, \Sigma_n^*)]}{\mathcal{K}(G_n, G_{*,n})}$$

$$= \frac{(\lambda_n^* - \lambda_n)[h_0(x|\mu_0, \Sigma_0) - f(x|\mu_n^*, \Sigma_n^*)]}{\mathcal{K}(G_n, G_{*,n})}$$

$$+ \frac{\lambda_n \left( \sum_{|\alpha|=1} \frac{(\mu_n - \mu_n^*)^{\alpha_1}(\Sigma_n - \Sigma_n^*)^{\alpha_2}}{\alpha!} \frac{\partial^{|\alpha|} f}{\partial \mu^{\alpha_1} \partial \Sigma^{\alpha_2}}(x|\mu_n^*, \Sigma_n^*) + R_1(x) \right)}{\mathcal{K}(G_n, G_{*,n})}$$

where $R_1(x)$ is Taylor remainder and $\alpha = (\alpha_1, \alpha_2)$ in the summation of the second equality satisfies $\alpha_1 = (\alpha_1^{(1)}, \ldots, \alpha_{d_1}^{(1)}) \in \mathbb{N}^{d_1}$, $\alpha_2 = (\alpha_{uv}^{(2)}) \in \mathbb{N}^{d_2 \times d_2}$, $|\alpha| = \sum_{i=1}^{d_1} \alpha_i^{(1)} + \sum_{1 \leq u,v \leq d_2} \alpha_{uv}^{(2)}$, and $\alpha! = \prod_{i=1}^{d_1} \alpha_i^{(1)}! \prod_{1 \leq u,v \leq d_2} \alpha_{uv}^{(2)}!$. As $f$ admits the first order uniform Lipschitz condition, we have $R_1(x) = O(\|(\mu_n, \Sigma_n) - (\mu_n^*, \Sigma_n^*)\|^{1+\gamma})$ for some $\gamma > 0$, which implies that

$$\lambda_n |R_1(x)| / \mathcal{K}(G_n, G_{*,n}) = O(\|(\mu_n, \Sigma_n) - (\mu_n^*, \Sigma_n^*)\|^\gamma) \to 0$$

as $n \to \infty$. Therefore, we can treat $[p_{G_n}(x) - p_{G_{*,n}}(x)]/\mathcal{K}(G_n, G_{*,n})$ as the linear combination of $h_0(x|\theta_0, \Sigma_0)$ and $\frac{\partial^{|\alpha|} f}{\partial \mu^{\alpha_1} \partial \Sigma^{\alpha_2}}(x|\mu_n^*, \Sigma_n^*)$ when $|\alpha| \leq 1$. Assume that the coefficients of these terms go to 0. Then, by studying the coefficients of $h_0(x|\theta_0, \Sigma_0)$, $\frac{\partial f}{\partial \mu_i}(x|\mu_0, \Sigma_0)$, and $\frac{\partial f}{\partial \Sigma_{uv}}(x|\mu_0, \Sigma_0)$, we achieve

$(\lambda_n^* - \lambda_n)/\mathcal{K}(G_n, G_{*,n}) \to 0$, $\lambda_n(\mu_n - \mu_n^*)_i/\mathcal{K}(G_n, G_{*,n}) \to 0$, $\lambda_n(\Sigma_n - \Sigma_n^*)_{uv}/\mathcal{K}(G_n, G_{*,n}) \to 0$

for all $1 \leq i \leq d_1$ and $1 \leq u, v \leq d_2$ where $(a)_i$ denotes the $i$-th element of vector $a$ and $A_{uv}$ denotes the $(u, v)$-th element of matrix $A$. It would imply that

$$(\lambda_n + \lambda_n^*)\|(\mu_n, \Sigma_n) - (\mu_n^*, \Sigma_n^*)\|/\mathcal{K}(G_n, G_{*,n}) \to 0.$$

Therefore, we achieve

$$1 = \left( |\lambda_n^* - \lambda_n| + (\lambda_n + \lambda_n^*)\|(\mu_n, \Sigma_n) - (\mu_n^*, \Sigma_n^*)\| \right)/\mathcal{K}(G_n, G_{*,n}) \to 0,$$

a contradiction. Therefore, not all the coefficients of $h_0(x|\theta_0, \Sigma_0)$ and $\frac{\partial^{|\alpha|} f}{\partial \mu^{\alpha_1} \partial \Sigma^{\alpha_2}}(x|\mu_n^*, \Sigma_n^*)$ go to 0. If we denote $m_n$ to be the maximum of the absolute values of the coefficients of $h_0(x|\theta_0, \Sigma_0)$ and $\frac{\partial^{|\alpha|} f}{\partial \mu^{\alpha_1} \partial \Sigma^{\alpha_2}}(x|\mu_n^*, \Sigma_n^*)$, then we get $1/m_n \not\to \infty$ as $n \to \infty$, i.e., $1/m_n$ is uniformly bounded. Hence, we achieve for all $x$ that

$$\frac{1}{m_n} \frac{p_{G_n}(x) - p_{G_{*,n}}(x)}{K(G_n, G_{*,n})} \to \eta f(|\mu_0, \Sigma_0) + \sum_{|\alpha| \leq 1} \tau_\alpha \frac{\partial^{|\alpha|} f}{\partial \mu^{\alpha_1} \partial \Sigma^{\alpha_2}}(x|\mu', \Sigma') = 0$$

for some coefficients $\eta$ and $\tau_\alpha$ such that they are not all 0. However, as $f$ is distinguishable from $h_0$ up to the first order, the above equation indicates that $\eta = \tau_\alpha = 0$ for all $|\alpha| \leq 1$, a contradiction. As a consequence, we achieve the conclusion of the proposition. $\square$

Now, assume that the conclusion of Theorem (3.3) does not hold. It implies that we can find two sequences $G'_n$ and $G'_{*,n}$ such that $A_n = \|p_{G'_n} - p_{G'_{*,n}}\|_2 / \mathcal{K}(G'_n, G'_{*,n}) \to 0$ as $n \to \infty$. Since $\Theta$ and $\Omega$ are two bounded subsets, we can find subsequences of $G'_n$ and $G'_{*,n}$ such that $\mathcal{K}(G'_n, \overline{G}_1)$ and $\mathcal{K}(G'_{*,n}, \overline{G}_2)$ vanish to 0 as $n \to \infty$ where $\overline{G}_1, \overline{G}_2$ are some discrete measures having one component to be $(\mu_0, \Sigma_0)$. Because $A_n \to 0$, we obtain $V(p_{G'_n}, p_{G'_{*,n}}) \to 0$ as $n \to \infty$. By means of Fatou's lemma, we have

$$0 = \lim_{n \to \infty} \int \left| (p_{G'_n}(x) - p_{G'_{*,n}}(x)) \right| dx \geq \int \liminf_{n \to \infty} \left| (p_{G'_n}(x) - p_{G'_{*,n}}(x)) \right| dx = V(p_{\overline{G}_1}(x), p_{\overline{G}_2}(x)).$$

Due to the fact that $f$ is distinguishable from $h_0$ up to the first order, the above equation implies that $\overline{G}_1 \equiv \overline{G}_2$. However, from the result of Proposition C.1, regardless of the value of $\overline{G}_1$ we would have $A_n \nrightarrow 0$ as $n \to \infty$, which is a contradiction. Therefore, we obtain the conclusion of the theorem.

## C.2 Proof of Theorem 3.5

Prior to presenting the proof of Theorem 3.5, we introduce the definition of second-order uniform Lipschitz:

**Definition C.2** (Second-order Uniform Lipschitz). We say that $f$ is uniformly Lipschitz up to the second order if the following holds: there are positive constants $\delta_3, \delta_4$ such that for any $R_4, R_5, R_6 > 0$, $\gamma_1 \in \mathbb{R}^{d_1}$, $\gamma_2 \in \mathbb{R}^{d_2 \times d_2}$, $R_4 \leq \sqrt{\lambda_1(\Sigma)} \leq \sqrt{\lambda_{d_2}(\Sigma)} \leq R_5$, $\|\theta\| \leq R_6$, $\theta_1, \theta_2 \in \Theta$, $\Sigma_1, \Sigma_2 \in \Omega$, there are positive constants $C_1$ depending on $(R_4, R_5)$ and $C_2$ depending on $R_6$ such that for all $x \in \mathcal{X}$,

$$\left| \gamma_1^\top \left( \frac{\partial^2 f}{\partial \theta^2}(x|\theta_1, \Sigma) - \frac{\partial^2 f}{\partial \theta^2}(x|\theta_2, \Sigma) \right) \gamma_1 \right| \leq \|\theta_1 - \theta_2\|_1^{\delta_3} \|\gamma_1\|_2^2,$$

$$\left| \mathrm{tr} \left( \left[ \frac{\partial}{\partial \Sigma} \left( \mathrm{tr} \left( \frac{\partial f}{\partial \Sigma_1}(x|\theta, \Sigma)^\top \gamma_2 \right) \right) - \frac{\partial}{\partial \Sigma} \left( \mathrm{tr} \left( \frac{\partial f}{\partial \Sigma}(x|\theta, \Sigma_2)^\top \gamma_2 \right) \right) \right]^\top \gamma_2 \right) \right|$$
$$\leq C_2 \|\Sigma_1 - \Sigma_2\|_2^{\delta_4} \|\gamma\|_2^2.$$

Now, we are back to the main proof. Utilizing the same Fatou's argument as that of Proposition C.1 , to achieve the conclusion of the first inequality in Theorem 3.5 it suffices to demonstrate the following result

**Proposition C.3.** *Given the assumptions in Theorem 3.5 and $\overline{G} = (\overline{\lambda}, \overline{\mu}, \overline{\Sigma})$ such that $\overline{\lambda} \in [0, 1]$ and $(\overline{\mu}, \overline{\Sigma})$ can be identical to $(\mu_0, \Sigma_0)$. Then, the following holds*

*(a) If $(\mu_0, \Sigma_0) \neq (\overline{\mu}, \overline{\Sigma})$ and $\overline{\lambda} > 0$, then*

$$\lim_{\epsilon \to 0} \inf_{G, G_*} \left\{ \frac{\|p_G - p_{G_*}\|_\infty}{\mathcal{K}(G, G_*)} \; : \; \mathcal{K}(G, \overline{G}) \vee \mathcal{K}(G_*, \overline{G}) \leq \epsilon \right\} > 0.$$

*(b) If $(\mu_0, \Sigma_0) \equiv (\overline{\mu}, \overline{\Sigma})$ or $(\mu_0, \Sigma_0) \neq (\overline{\mu}, \overline{\Sigma})$ and $\overline{\lambda} = 0$, then*

$$\lim_{\epsilon \to 0} \inf_{G, G_*} \left\{ \frac{\|p_G - p_{G_*}\|_\infty}{\mathcal{D}(G, G_*)} \; : \; \mathcal{D}(G, \overline{G}) \vee \mathcal{D}(G_*, \overline{G}) \leq \epsilon \right\} > 0.$$

*Proof.* The proof of part (a) is essentially similar to that of Proposition C.1; therefore, we only provide the proof for the challenging settings of part (b). Here, we only consider the setting that $(\mu_0, \Sigma_0) \equiv (\overline{\mu}, \overline{\Sigma})$ as the proof for other possibilities of $(\mu_0, \Sigma_0)$ can be argued in the similar fashion. Under this assumption, $(\mu_0, \Sigma_0) = (\mu_0, \Sigma_0)$, $\overline{G} = (\overline{\lambda}, \theta_0, \Sigma_0)$, and $h_0(x|\theta_0, \Sigma_0) = f(x|\theta_0, \Sigma_0)$ for all $x \in \mathcal{X}$. Assume that the conclusion of Proposition C.3 does not hold. It implies that we can find two sequences $G_n = (\lambda_n, \mu_n, \Sigma_n)$ and $G_{*,n} = (\lambda_n^*, \mu_n^*, \Sigma_n^*)$ such that $\mathcal{D}(G_n, \overline{G}) = \lambda_n \|(\Delta \mu_n, \Delta \Sigma_n)\|^2 \to 0$, $\mathcal{D}(G_{*,n}, \overline{G}) = \lambda_n^* \|(\Delta \mu_n^*, \Delta \Sigma_n^*)\|^2 \to 0$, and $\|p_{G_n} - p_{G_{*,n}}\|_\infty / \mathcal{D}(G_n, G_{*,n}) \to 0$ as $n \to \infty$. For the transparency of presentation, we denote $A_n = \|(\Delta \mu_n, \Delta \Sigma_n)\|$, $B_n = \|(\Delta \mu_n^*, \Delta \Sigma_n^*)\|$, and $C_n = \|(\mu_n, \Sigma_n) - (\mu_n^*, \Sigma_n^*)\| = \|(\Delta \mu_n, \Delta \Sigma_n) - (\Delta \mu_n^*, \Delta \Sigma_n^*)\|$. Now, we have three main cases regarding the convergence behaviors of $(\mu_n, \Sigma_n)$ and $(\mu_n^*, \Sigma_n^*)$

**Case 1:** Both $A_n \to 0$ and $B_n \to 0$, i.e., $(\mu_n, \Sigma_n)$ and $(\mu_n^*, \Sigma_n^*)$ vanish to $(\mu_0, \Sigma_0)$ as $n \to \infty$. Due to the symmetry between $\lambda_n$ and $\lambda_n^*$, we assume without loss of generality that $\lambda_n^* \geq \lambda_n$ for infinite values of $n$. Without loss of generality, we replace these subsequences of $G_n, G_{*,n}$ by the whole sequences of $G_n$ and $G_{*,n}$. Now, the formulation of $\mathcal{D}(G_n, G_{*,n})$ is

$$\mathcal{D}(G_n, G_{*,n}) = (\lambda_n^* - \lambda_n)B_n^2 + \left(\lambda_n A_n + \lambda_n^* B_n\right)C_n.$$

Now, by means of Taylor expansion up to the second order, we get

$$
\begin{aligned}
\frac{p_{G_n}(x) - p_{G_{*,n}}(x)}{\mathcal{D}(G_n, G_{*,n})} &= \frac{(\lambda_n^* - \lambda_n)[f(x|\mu_0, \Sigma_0) - f(x|\mu_n^*, \Sigma_n^*)] + \lambda_n[f(x|\mu_n, \Sigma_n) - f(x|\mu_n^*, \Sigma_n^*)]}{\mathcal{D}(G_n, G_{*,n})} \\[2mm]
&= \frac{(\lambda_n^* - \lambda_n)\left(\sum_{|\alpha|=1}^{2} \frac{(-\Delta\mu_n^*)^{\alpha_1}(-\Delta\Sigma_n^*)^{\alpha_2}}{\alpha!} \frac{\partial^{|\alpha|} f}{\partial\mu^{\alpha_1}\partial\Sigma^{\alpha_2}}(x|\mu_n^*, \Sigma_n^*) + R_1(x)\right)}{\mathcal{D}(G_n, G_{*,n})} \\[2mm]
&\quad + \frac{\lambda_n\left(\sum_{|\alpha|=1}^{2} \frac{(\Delta\mu_n - \Delta\mu_n^*)^{\alpha_1}(\Delta\Sigma_n - \Delta\Sigma_n^*)^{\alpha_2}}{\alpha!} \frac{\partial^{|\alpha|} f}{\partial\mu^{\alpha_1}\partial\Sigma^{\alpha_2}}(x|\mu_n^*, \Sigma_n^*) + R_2(x)\right)}{\mathcal{D}(G_n, G_{*,n})}
\end{aligned}
$$

where $R_1(x)$ and $R_2(x)$ are Taylor remainders that satisfy $R_1(x) = O(B_n^{2+\gamma})$ and $R_2(x) = O(C_n^{2+\gamma})$ for some positive number $\gamma$ due to the second order uniform Lipschitz condition of kernel density function $f$. From the formation of $\mathcal{D}(G_n, G_{*,n})$, since $A_n + B_n \geq C_n$ (triangle inequality), as $A_n \to 0$ and $B_n \to 0$ it is clear that

$$(\lambda_n - \lambda_n^*)|R_1(x)|/\mathcal{D}(G_n, G_{*,n}) \leq |R_1(x)|/B_n^2 = O(B_n^\gamma) \to 0$$

$$\lambda_n|R_2(x)|/\mathcal{D}(G_n, G_{*,n}) \leq |R_2(x)|/\{(A_n + B_n)C_n\} = O\left(C_n^{2+\gamma}/C_n^2\right) = O(C_n^\gamma) \to 0$$

as $n \to \infty$ for all $x \in \mathcal{X}$. Therefore, we achieve for all $x \in \mathcal{X}$ that

$$\left((\lambda_n - \lambda_n^*)|R_1(x)| + \lambda_n|R_2(x)|\right)/\mathcal{D}(G_n, G_{*,n}) \to 0.$$

Hence, we can treat $[p_{G_n}(x) - p_{G_{*,n}}(x)]/\mathcal{D}(G_n, G_{*,n})$ as a linear combination of $\dfrac{\partial^{|\alpha|} f}{\partial\mu^{\alpha_1}\partial\Sigma^{\alpha_2}}(x|\mu_n^*, \Sigma_n^*)$ for all $x$ and $\alpha = (\alpha_1, \alpha_2)$ such that $1 \leq |\alpha| \leq 2$. Assume that all the coefficients of these terms go to 0 as $n \to \infty$. By studying the vanishing behaviors of the coefficients of $\dfrac{\partial^{|\alpha|} f}{\partial\mu^{\alpha_1}\partial\Sigma^{\alpha_2}}(x|\mu_n^*, \Sigma_n^*)$ as $|\alpha| = 1$, we achieve the following limits

$$\left(\lambda_n(\Delta\mu_n)_i - \lambda_n^*(\Delta\mu_n^*)_i\right)/\mathcal{D}(G_n, G_{*,n}) \to 0, \quad \left(\lambda_n(\Delta\Sigma_n)_{uv} - \lambda_n^*(\Delta\Sigma_n^*)_{uv}\right)/\mathcal{D}(G_n, G_{*,n}) \to 0$$

for all $1 \leq i \leq d_1$ and $1 \leq u, v \leq d_2$ where $(a)_i$ denotes the $i$-th element of vector $a$ and $A_{uv}$ denotes the $(u, v)$-th element of matrix $A$. Furthermore, for any $1 \leq i, j \leq d$ ($i$ and $j$ can be equal), the coefficient of $\dfrac{\partial^{|\alpha|} f}{\partial\mu^{\alpha_1}\partial\Sigma^{\alpha_2}}(x|\mu_n^*, \Sigma_n^*)$ when $(\alpha_1)_i = (\alpha_1)_j = 1$ and $\alpha_2 = 0$ leads to

$$\left[(\lambda_n^* - \lambda_n)(\Delta\mu_n^*)_i(\Delta\mu_n^*)_j + \lambda_n(\Delta\mu_n - \Delta\mu_n^*)_i(\Delta\mu_n - \Delta\mu_n^*)_j\right]/\mathcal{D}(G_n, G_{*,n}) \to 0. \qquad (20)$$

When $i = j$, the above limits lead to

$$\left[(\lambda_n^* - \lambda_n)\{(\Delta\mu_n^*)_i\}^2 + \lambda_n\{(\Delta\mu_n - \Delta\mu_n^*)_i\}^2\right]/\mathcal{D}(G_n, G_{*,n}) \to 0.$$

Therefore, we would have

$$\left[(\lambda_n^* - \lambda_n)\|\Delta\mu_n^*\|^2 + \lambda_n\|\Delta\mu_n - \Delta\mu_n^*\|^2\right]/\mathcal{D}(G_n, G_{*,n}) \to 0. \qquad (21)$$

Now, as $\left(\lambda_n(\Delta\mu_n)_i - \lambda_n^*(\Delta\mu_n^*)_i\right)/\mathcal{D}(G_n, G_{*,n}) \to 0$ we obtain that

$$\left(\lambda_n(\Delta\mu_n)_i(\Delta\mu_n)_j - \lambda_n^*(\Delta\mu_n^*)_i(\Delta\mu_n)_j\right)/\mathcal{D}(G_n, G_{*,n}) \quad \to \quad 0,$$

$$\left(\lambda_n(\Delta\mu_n)_i(\Delta\mu_n^*)_j - \lambda_n^*(\Delta\mu_n^*)_i(\Delta\mu_n^*)_j\right)/\mathcal{D}(G_n, G_{*,n}) \quad \to \quad 0. \tag{22}$$

Plugging the results from (22) into (20), we ultimately achieve for any $1 \le i, j \le d$ that

$$(\lambda_n^* - \lambda_n)(\Delta\mu_n^*)_i(\Delta\mu_n)_j/\mathcal{D}(G_n, G_{*,n}) \to 0. \tag{23}$$

Using the results from (20) and (23), we would have

$$\frac{\lambda_n(\Delta\mu_n)_i(\Delta\mu_n - \Delta\mu_n^*)_j}{\mathcal{D}(G_n, G_{*,n})} \to \frac{(\lambda_n^* - \lambda_n)(\Delta\mu_n)_i(\Delta\mu_n^*)_j}{\mathcal{D}(G_n, G_{*,n})} \to 0,$$

$$\frac{\lambda_n^*(\Delta\mu_n^*)_i(\Delta\mu_n - \Delta\mu_n^*)_j}{\mathcal{D}(G_n, G_{*,n})} \to \frac{(\lambda_n^* - \lambda_n)(\Delta\mu_n^*)_i(\Delta\mu_n)_j}{\mathcal{D}(G_n, G_{*,n})} \to 0$$

for any $1 \le i, j \le d$. Therefore, it leads to

$$\frac{\sum\limits_{1 \le i,j \le d} \lambda_n|(\Delta\mu_n)_i||(\Delta\mu_n - \Delta\mu_n^*)_j|}{\mathcal{D}(G_n, G_{*,n})} = \frac{\lambda_n \sum\limits_{1 \le i \le d} |(\Delta\mu_n)_i| \sum\limits_{1 \le i \le d} |(\Delta\mu_n - \Delta\mu_n^*)_i|}{\mathcal{D}(G_n, G_{*,n})} \to 0,$$

$$\frac{\sum\limits_{1 \le i,j \le d} \lambda_n^*|(\Delta\mu_n^*)_i||(\Delta\mu_n - \Delta\mu_n^*)_j|}{\mathcal{D}(G_n, G_{*,n})} = \frac{\lambda_n^* \sum\limits_{1 \le i \le d} |(\Delta\mu_n)_i^*| \sum\limits_{1 \le i \le d} |(\Delta\mu_n - \Delta\mu_n^*)_i|}{\mathcal{D}(G_n, G_{*,n})} \to 0.$$

The above results mean that

$$\lambda_n\|\Delta\mu_n\|\|\Delta\mu_n - \Delta\mu_n^*\|/\mathcal{D}(G_n, G_{*,n}) \to 0, \quad \lambda_n^*\|\Delta\mu_n^*\|\|\Delta\mu_n - \Delta\mu_n^*\|/\mathcal{D}(G_n, G_{*,n}) \to 0. \tag{24}$$

By applying the above argument with the coefficients of $\dfrac{\partial^{|\alpha|} f}{\partial\mu^{\alpha_1}\partial\Sigma^{\alpha_2}}(x|\mu_n^*, \Sigma_n^*)$ when $\alpha_1 = 0$ and $(\alpha_2)_{u_1 v_1} = (\alpha_2)_{u_2 v_2} = 1$ for any two pairs $(u_1, v_1), (u_2, v_2)$ (not neccessarily distinct) such that $1 \le u_1, u_2, v_1, v_2 \le d$ or $(\alpha_1)_i = 1$ and $(\alpha_2)_{uv} = 1$ for any $1 \le i \le d$ and $1 \le u, v \le d$, we respectively obtain that

$$\left[(\lambda_n^* - \lambda_n)\|\Delta\Sigma_n^*\|^2 + \lambda_n\|\Delta\Sigma_n - \Delta\Sigma_n^*\|^2\right]/\mathcal{D}(G_n, G_{*,n}) \to 0,$$

$$\lambda_n\|\Delta\Sigma_n\|\|\Delta\Sigma_n - \Delta\Sigma_n^*\|/\mathcal{D}(G_n, G_{*,n}) \to 0, \quad \lambda_n^*\|\Delta\Sigma_n^*\|\|\Delta\Sigma_n - \Delta\Sigma_n^*\|/\mathcal{D}(G_n, G_{*,n}) \to 0,$$

$$\lambda_n\|\Delta\mu_n\|\|\Delta\Sigma_n - \Delta\Sigma_n^*\|/\mathcal{D}(G_n, G_{*,n}) \to 0, \quad \lambda_n^*\|\Delta\mu_n^*\|\|\Delta\Sigma_n - \Delta\Sigma_n^*\|/\mathcal{D}(G_n, G_{*,n}) \to 0. \tag{25}$$

Combining the results from (21), (24), and (25) leads to

$$1 = \mathcal{D}(G_n, G_{*,n})/\mathcal{D}(G_n, G_{*,n}) \to 0,$$

which is a contradiction. As a consequence, not all the coefficients of $\dfrac{\partial^{|\alpha|} f}{\partial\mu^{\alpha_1}\partial\Sigma^{\alpha_2}}(x|\mu_n^*, \Sigma_n^*)$ go to 0 as $1 \le |\alpha| \le 2$. Follow the argument of Proposition C.1, by denoting $m_n$ to be the maximum of the absolute values of the coefficients of $\dfrac{\partial^{|\alpha|} f}{\partial\mu^{\alpha_1}\partial\Sigma^{\alpha_2}}(x|\mu_n^*, \Sigma_n^*)$ we achieve for all $x$ that

$$\frac{1}{m_n}\frac{p_{G_n}(x) - p_{G_{*,n}}(x)}{W_2^2(G_n, G_{*,n})} \to \sum_{|\alpha|=1}^{2} \tau_\alpha \frac{\partial^{|\alpha|} f}{\partial\mu^{\alpha_1}\partial\Sigma^{\alpha_2}}(x|\mu_0, \Sigma_0) = 0$$

where $\tau_\alpha \in \mathbb{R}$ are some coefficients such that not all of them are 0. Due to the second order identifiability condition of $f$, the above equation implies that $\tau_\alpha = 0$ for all $\alpha$ such that $|\alpha| = 2$, which is a contradiction. As a consequence, Case 1 cannot happen.

**Case 2:** Exactly one of $A_n$ and $B_n$ goes to 0, i.e., there exists at least one component among $(\mu_n, \Sigma_n)$ and $(\mu_n^*, \Sigma_n^*)$ that does not converge to $(\mu_0, \Sigma_0)$ as $n \to \infty$. Due to the symmetry of $A_n$ and $B_n$, we assume without loss of generality that $A_n \not\to 0$ and $B_n \to 0$, which is equivalent to $(\mu_n, \Sigma_n) \to (\mu', \Sigma') \neq (\mu_0, \Sigma_0)$ while $(\mu_n^*, \Sigma_n^*) \to (\mu_0, \Sigma_0)$ as $n \to \infty$. We denote

$$\mathcal{D}'(G_n, G_{*,n}) = |\lambda_n^* - \lambda_n| B_n + \lambda_n A_n + \lambda_n^* B_n.$$

Since $[p_{G_n}(x) - p_{G_{*,n}}(x)]/\mathcal{D}(G_n, G_{*,n}) \to 0$, we achieve that $[p_{G_n}(x) - p_{G_{*,n}}(x)]/\mathcal{D}'(G_n, G_{*,n}) \to 0$ for all $x$ as $\mathcal{D}(G_n, G_{*,n}) \lesssim \mathcal{D}'(G_n, G_{*,n})$. By means of Taylor expansion up to the first order, we have

$$
\frac{p_{G_n}(x) - p_{G_{*,n}}(x)}{\mathcal{D}'(G_n, G_{*,n})} = \frac{(\lambda_n^* - \lambda_n)[f(x|\mu_0, \Sigma_0) - f(x|\mu_n^*, \Sigma_n^*)] + \lambda_n f(x|\mu_n, \Sigma_n) - \lambda_n f(x|\mu_n^*, \Sigma_n^*)}{\mathcal{D}'(G_n, G_{*,n})}
$$

$$
= \frac{(\lambda_n^* - \lambda_n)\left( \sum_{|\alpha|=1} \frac{(-\Delta\mu_n^*)^{\alpha_1}(-\Delta\Sigma_n^*)^{\alpha_2}}{\alpha!} \frac{\partial f}{\partial\mu^{\alpha_1}\partial\Sigma^{\alpha_2}}(x|\mu_n^*, \Sigma_n^*) + R_1'(x) \right)}{\mathcal{D}'(G_n, G_{*,n})}
$$

$$
+ \frac{\lambda_n f(x|\mu_n, \Sigma_n) - \lambda_n f(x|\mu_n^*, \Sigma_n^*)}{\mathcal{D}'(G_n, G_{*,n})}
$$

where $R_1'(x)$ is Taylor remainder that satisfies $(\lambda_n^* - \lambda_n)|R_1'(x)|/\mathcal{D}'(G_n, G_{*,n}) = O(B_n^{\gamma'}) \to 0$ for some positive number $\gamma' > 0$. Since $(\mu_n, \Sigma_n)$ and $(\mu_n^*, \Sigma_n^*)$ do not have the same limit, they will be different when $n$ is large enough, i.e., $n \geq M'$ for some value of $M'$. Now, as $n \geq M'$, $[p_{G_n}(x) - p_{G_{*,n}}(x)]/\mathcal{D}'(G_n, G_{*,n})$ becomes a linear combination of $\frac{\partial f}{\partial\mu^{\alpha_1}\partial\Sigma^{\alpha_2}}(x|\mu_n^*, \Sigma_n^*)$ for all $|\alpha| \leq 1$ and $f(x|\mu_n, \Sigma_n)$. If all of the coefficients of these terms go to 0, we would have $\lambda_n/\mathcal{D}'(G_n, G_{*,n}) \to 0$, $(\lambda_n^* - \lambda_n)(-\Delta\mu_n^*)_i/\mathcal{D}'(G_n, G_{*,n}) \to 0$, and $(\lambda_n^* - \lambda_n)(-\Delta\Sigma_n^*)_{uv}/\mathcal{D}'(G_n, G_{*,n}) \to 0$ for all $1 \leq i \leq d_1$ and $1 \leq u, v \leq d_2$. It would imply that $(\lambda_n^* - \lambda_n)B_n/\mathcal{D}'(G_n, G_{*,n}) \to 0$, $\lambda_n A_n/\mathcal{D}'(G_n, G_{*,n}) \to 0$, and $\lambda_n B_n/\mathcal{D}'(G_n, G_{*,n}) \to 0$. These results lead to

$$1 = \left( |\lambda_n^* - \lambda_n| B_n + \lambda_n A_n + \lambda_n^* B_n \right)/\mathcal{D}'(G_n, G_{*,n}) \to 0,$$

a contradiction. Therefore, not all the coefficients of $\frac{\partial f}{\partial\mu^{\alpha_1}\partial\Sigma^{\alpha_2}}(x|\mu_n^*, \Sigma_n^*)$ and $f(x|\mu_n, \Sigma_n)$ go to 0. By defining $m_n'$ to be the maximum of these coefficients, we achieve for all $x$ that

$$\frac{1}{m_n'} \frac{p_{G_n}(x) - p_{G_{*,n}}(x)}{\mathcal{D}'(G_n, G_{*,n})} \to \eta' f(x|\mu_0, \Sigma_0) + \sum_{|\alpha|=0}^{1} \tau_\alpha' \frac{\partial^{|\alpha|} f}{\partial\mu^{\alpha_1}\partial\Sigma^{\alpha_2}}(x|\mu', \Sigma') = 0,$$

where $\eta'$ and $\tau_\alpha'$ are coefficients such that not all of them are 0, which is a contradiction to the first order identifiability of $f$. As a consequence, Case 2 cannot hold.

**Case 3:** Both $A_n$ and $B_n$ do not go to 0, i.e., $(\mu_n, \Sigma_n)$ and $(\mu_n^*, \Sigma_n^*)$ do not converge to $(\mu_0, \Sigma_0)$ as $n \to \infty$. Since $\mathcal{D}_n(G_n, G_{*,n}) \lesssim \mathcal{K}(G_n, G_{*,n}) = |\lambda_n - \lambda_n^*| + (\lambda_n + \lambda_n^*)C_n$ and $[p_{G_n}(x) - p_{G_{*,n}}(x)]/\mathcal{D}(G_n, G_{*,n}) \to 0$, we achieve that $[p_{G_n}(x) - p_{G_{*,n}}(x)]/\mathcal{K}(G_n, G_{*,n}) \to 0$ for all $x$. From here, by using the same argument as that of the proof of Proposition C.1, we also reach the contradiction. Therefore, Case 3 cannot happen.

In sum, we achieve the conclusion of the proposition. $\qquad\square$

### C.3 Proof of Theorem 3.6

For the simplicity of proof argument, we will only consider the univariate setting of Gaussian kernel, i.e., when both $\mu$ and $\Sigma = \sigma^2$ are scalars. The argument for the multivariate setting of Gaussian kernel can be argued in the rather similar fashion, which is omitted. Throughout this proof, we denote $v := \sigma^2$. Now, according to the proof argument of Proposition C.1 and Proposition C.3, to achieve the conclusion of the theorem it suffices to demonstrate the following result:

**Proposition C.4.** *Given $\overline{G} = (\overline{\lambda}, \overline{\mu}, \overline{v})$ such that $\overline{\lambda} \in [0, 1]$ and $(\overline{\mu}, \overline{v})$ can be identical to $(\mu_0, v_0)$. Then, the following holds*

(a) If $(\mu_0, v_0) \neq (\overline{\mu}, \overline{v})$ and $\overline{\lambda} > 0$, then

$$\lim_{\epsilon \to 0} \inf_{G, G_*} \left\{ \frac{\|p_G - p_{G_*}\|_\infty}{\mathcal{K}(G, G_*)} \; : \; \mathcal{K}(G, \overline{G}) \vee \mathcal{K}(G_*, \overline{G}) \leq \epsilon \right\} > 0.$$

(b) If $(\mu_0, v_0) \equiv (\overline{\mu}, \overline{v})$ or $(\mu_0, v_0) \neq (\overline{\mu}, \overline{v})$ and $\overline{\lambda} = 0$, then

$$\lim_{\epsilon \to 0} \inf_{G, G_*} \left\{ \frac{\|p_G - p_{G_*}\|_\infty}{\mathcal{Q}(G, G_*)} \; : \; \mathcal{Q}(G, \overline{G}) \vee \mathcal{Q}(G_*, \overline{G}) \leq \epsilon \right\} > 0.$$

*Proof.* We will only provide the proof for part (b) since the proofs for part (a) can be argued in similar fashion as that of Proposition C.1. Assume that the conclusion of Proposition C.4 does not hold. It implies that we can find two sequences $G_n = (\lambda_n, \mu_n, v_n)$ and $G_{*,n} = (\lambda_n^*, \mu_n^*, v_n^*)$ such that $\mathcal{Q}(G_n, \overline{G}) \to 0$, $\mathcal{Q}(G_{*,n}, \overline{G}) \to 0$, and $\|p_{G_n} - p_{G_{*,n}}\|_\infty / \mathcal{Q}(G_n, G_{*,n}) \to 0$ as $n \to \infty$. Due to the symmetry between $\lambda_n$ and $\lambda_n^*$, we can assume without loss of generality that $\lambda_n^* \geq \lambda_n$. Therefore, we achieve that

$$\mathcal{Q}(G_n, G_{*,n}) = (\lambda_n^* - \lambda_n)(|\Delta\mu_n^*|^4 + |\Delta v_n^*|^2) + \left( \lambda_n(|\Delta\mu_n|^2 + |\Delta v_n|) + \lambda_n^*(|\Delta\mu_n^*|^2 + |\Delta v_n^*|) \right) \times$$

$$\times \left( |\mu_n - \mu_n^*|^2 + |v_n - v_n^*| \right).$$

In this proof, we only consider the scenario when $\|(\Delta\mu_n, \Delta v_n)\| \to 0$ and $\|(\Delta\mu_n^*, \Delta v_n^*)\| \to 0$ since the arguments for other settings of these two terms are similar to those of Case 2 and Case 3 in the proof of Proposition C.3. As being indicated in Section 3.2.2, the univariate Gaussian kernel contains the partial differential equation structure $\frac{\partial^2 f}{\partial \mu^2}(x|\mu, v) = 2\frac{\partial f}{\partial v}(x|\mu, v)$ for all $\mu \in \Theta$ and $v \in \Omega$. Therefore, for any $\alpha = (\alpha_1, \alpha_2)$ we can check that

$$\frac{\partial^{|\alpha|} f}{\partial \mu^{\alpha_1} \partial v^{\alpha_2}}(x|\mu, v) = \frac{1}{2^{\alpha_2}} \frac{\partial^\beta f}{\partial \mu^\beta}(x|\mu, v)$$

where $\beta = \alpha_1 + 2\alpha_2$. Now, by means of Taylor expansion up to the fourth order, we obtain

$$\frac{p_{G_n}(x) - p_{G_{*,n}}(x)}{\mathcal{Q}(G_n, G_{*,n})} = \frac{(\lambda_n^* - \lambda_n)\left( \sum_{|\alpha|=1}^{4} \frac{(-\Delta\mu_n^*)^{\alpha_1}(-\Delta v_n^*)^{\alpha_2}}{\alpha_1!\alpha_2!} \frac{\partial^{|\alpha|} f}{\partial \mu^{\alpha_1} \partial v^{\alpha_2}}(x|\mu_n^*, v_n^*) + R_1(x) \right)}{\mathcal{Q}(G_n, G_{*,n})}$$

$$+ \frac{\lambda_n\left( \sum_{|\alpha|=1}^{4} \frac{(\Delta\mu_n - \Delta\mu_n^*)^{\alpha_1}(\Delta v_n - \Delta v_n^*)^{\alpha_2}}{\alpha_1!\alpha_2!} \frac{\partial^{|\alpha|} f}{\partial \mu^{\alpha_1} \partial v^{\alpha_2}}(x|\mu_n^*, v_n^*) + R_2(x) \right)}{\mathcal{Q}(G_n, G_{*,n})}$$

$$= \sum_{\beta=1}^{8} \sum_{\alpha_1, \alpha_2} \frac{(\lambda_n^* - \lambda_n)(-\Delta\mu_n^*)^{\alpha_1}(-\Delta v_n^*)^{\alpha_2} + \lambda_n(\Delta\mu_n - \Delta\mu_n^*)^{\alpha_1}(\Delta v_n - \Delta v_n^*)^{\alpha_2}}{2^{\alpha_2}\alpha_1!\alpha_2!\mathcal{Q}(G_n, G_{*,n})}$$

$$\times \frac{\partial^\beta f}{\partial \mu^\beta}(x|\mu_n^*, v_n^*) + \frac{(\lambda_n^* - \lambda_n)R_1(x) + \lambda_n R_2(x)}{\mathcal{Q}(G_n, G_{*,n})}$$

where $R_1(x), R_2(x)$ are Taylor remainders and the range of $\alpha_1, \alpha_2$ in the summation of the second equality satisfies $\beta = \alpha_1 + 2\alpha_2$. As Gaussian kernel admits fourth-order uniform Lipschitz condition, it is clear that

$$\frac{(\lambda_n^* - \lambda_n)|R_1(x)| + \lambda_n|R_2(x)|}{\mathcal{Q}(G_n, G_{*,n})} = \mathcal{O}(\|(\Delta\mu_n^*, \Delta v_n^*)\|^\gamma + \|(\mu_n, v_n) - (\mu_n^*, v_n^*)\|^\gamma) \to 0$$

as $n \to \infty$ for some $\gamma > 0$. Therefore, we can consider $[p_{G_n}(x) - p_{G_{*,n}}(x)]/\mathcal{Q}(G_n, G_{*,n})$ as a linear combination of $\frac{\partial^\beta f}{\partial \mu^\beta}(x|\mu_n^*, v_n^*)$ for $1 \leq \beta \leq 8$. If all of the coefficients of these terms go to 0, then we obtain

$$L_\beta = \frac{\sum_{\alpha_1, \alpha_2} \frac{(\lambda_n^* - \lambda_n)(-\Delta\mu_n^*)^{\alpha_1}(-\Delta v_n^*)^{\alpha_2} + \lambda_n(\Delta\mu_n - \Delta\mu_n^*)^{\alpha_1}(\Delta v_n - \Delta v_n^*)^{\alpha_2}}{2^{|\alpha_2|}\alpha_1!\alpha_2!}}{\mathcal{Q}(G_n, G_{*,n})} \to 0$$

for any $1 \leq \beta \leq 8$. Now, we divide our argument with $L_\beta$ into two key cases

**Case 1:** $\left(\lambda_n(|\Delta\mu_n|^2 + |\Delta v_n|) + \lambda_n^*(|\Delta\mu_n^*|^2 + |\Delta v_n^*|)\right)/\left\{\lambda_n(|\mu_n - \mu_n^*|^2 + |v_n - v_n^*|)\right\} \not\to \infty.$

It implies that as $n$ is large enough, we would have

$$Q(G_n, G_{*,n}) \lesssim (\lambda_n^* - \lambda_n)(|\Delta\mu_n^*|^4 + |\Delta v_n^*|^2) + \lambda_n(|\Delta\mu_n - \Delta\mu_n^*|^4 + |\Delta v_n - \Delta v_n^*|^2).$$

Combining the above result with $L_\beta \to 0$ for all $1 \le \beta \le 8$, we get

$$H_\beta = \frac{\displaystyle\sum_{\alpha_1,\alpha_2} \frac{(\lambda_n^* - \lambda_n)(-\Delta\mu_n^*)^{\alpha_1}(-\Delta v_n^*)^{\alpha_2} + \lambda_n(\Delta\mu_n - \Delta\mu_n^*)^{\alpha_1}(\Delta v_n - \Delta v_n^*)^{\alpha_2}}{2^{|\alpha_2|}\alpha_1!\alpha_2!}}{(\lambda_n^* - \lambda_n)(|\Delta\mu_n^*|^4 + |\Delta v_n^*|^2) + \lambda_n(|\Delta\mu_n - \Delta\mu_n^*|^4 + |\Delta v_n - \Delta v_n^*|^2)} \to 0$$

Note that, when the denominator of the above limits is $(\lambda_n^* - \lambda_n)(|\Delta\mu_n^*|^4 + |\Delta v_n^*|^4) + \lambda_n(|\Delta\mu_n - \Delta\mu_n^*|^4 + |\Delta v_n - \Delta v_n^*|^4)$, the technique for studying the above system of limits with this denominator has been considered in Proposition 2.3 in [21]. However, since the current denominator of $H_\beta$ strongly dominates by the previous denominator, we must develop a more sophisticated control of $H_\beta$ as $1 \le \beta \le 8$ to obtain a concrete understanding of their limits. Due to the symmetry between $\lambda_n^* - \lambda_n$ and $\lambda_n$, we assume without loss of generality that $\lambda_n^* - \lambda_n \le \lambda_n$ for all $n$ (by the subsequence argument). We have two possibilities regarding $\lambda_n$ and $\lambda_n^*$

**Case 1.1:** $(\lambda_n^* - \lambda_n)/\lambda_n \not\to 0$ as $n \to \infty$. Under that setting, we define $p_n = \max\{\lambda_n^* - \lambda_n, \lambda_n\}$ and

$$M_n = \max\left\{|\Delta\mu_n^*|, |\Delta\mu_n^* - \Delta\mu_n|, |\Delta v_n^*|^{1/2}, |\Delta v_n^* - \Delta v_n|^{1/2}\right\}$$

Additionally, we let $(\lambda_n^* - \lambda_n)/p_n \to c_1^2$, $\lambda_n/p_n \to c_2^2$, $\Delta\mu_n^*/M_n \to -a_1$, $(\Delta\mu_n^* - \Delta\mu_n)/M_n \to a_2$, $\Delta v_n^*/M_n^2 \to -2b_1$, and $(\Delta v_n - \Delta v_n^*)/M_n^2 \to 2b_2$. From here, at least one among $a_1, a_2, b_1, b_2$ and both $c_1, c_2$ are different from 0. Now, by dividing both the numerators and the denominators of $H_\beta$ as $1 \le \beta \le 4$ by $p_n M_n^\beta$, we achieve the following system of polynomial equations

$$c_1^2 a_1 + c_2^2 a_2 = 0$$
$$\frac{1}{2}(c_1^2 a_1^2 + c_2^2 a_2^2) + c_1^2 b_1 + c_2^2 b_2 = 0$$
$$\frac{1}{3!}(c_1^2 a_1^3 + c_2^2 a_2^3) + c_1^2 a_1 b_1 + c_2^2 a_2 b_2 = 0$$
$$\frac{1}{4!}(c_1^2 a_1^4 + c_2^2 a_2^4) + \frac{1}{2!}(c_1^2 a_1^2 b_1 + c_2^2 a_2^2 b_2) + \frac{1}{2!}(c_1^2 b_1^2 + c_2^2 b_2^2) = 0,$$

As being indicated in Proposition 2.1 in [21], this system will only admits the trivial solution, i.e., $a_1 = a_2 = b_1 = b_2 = 0$, which is a contradiction. Therefore, Case 1.1 cannot happen.

**Case 1.2:** $(\lambda_n^* - \lambda_n)/\lambda_n \to 0$, i.e., $\lambda_n^*/\lambda_n \to 1$, as $n \to \infty$. Under that setting, if $M_n \in \max\{|\Delta\mu_n - \Delta\mu_n^*|, |\Delta v_n - \Delta v_n^*|^{1/2}\}$, then we have

$$\lambda_n M_n^4 = \max\left\{(\lambda_n^* - \lambda_n)|\Delta\mu_n^*|^4, (\lambda_n^* - \lambda_n)|\Delta\mu_n - \Delta\mu_n^*|^4, \lambda_n|\Delta v_n^*|^2, \lambda_n|\Delta v_n - \Delta v_n^*|^2\right\}.$$

By dividing both the numerator and the denominator of $H_1$ by $\lambda_n M_n$, given that the new denominator of $H_1$ goes to 0, its new numerator also goes to 0, i.e., we obtain

$$(\lambda_n^* - \lambda_n)(-\Delta\mu_n^*)/\{\lambda_n M_n\} + (\Delta\mu_n - \Delta\mu_n^*)/M_n \to 0.$$

Since $(\lambda_n^* - \lambda_n)/\lambda_n \to 0$ and $|\Delta\mu_n^*| \le M_n$, we have $(\lambda_n^* - \lambda_n)(-\Delta\mu_n^*)/\{\lambda_n M_n\} \to 0$. Therefore, we have $(\Delta\mu_n - \Delta\mu_n^*)/M_n \to 0$. With the previous results, by dividing both the numerator and the denominator of $H_2$ by $\lambda_n M_n^2$ and given that the new denominator goes to 0, we have

$$(\lambda_n^* - \lambda_n)(-\Delta v_n^*)/\{\lambda_n M_n^2\} + (\Delta v_n - \Delta v_n^*)/M_n^2 \to 0.$$

As $(\lambda_n^* - \lambda_n)(-\Delta v_n^*)/\{\lambda_n M_n^2\} \to 0$ (due to the assumption of $M_n$), we get $(\Delta v_n - \Delta v_n^*)/M_n^2 \to 0$. These results imply that

$$1 = \frac{\max\{|\Delta\mu_n - \Delta\mu_n^*|^2, |\Delta v_n - \Delta v_n^*|\}}{M_n^2} \to 0,$$

which is a contradiction. Therefore, we would only have $M_n \in \max\left\{|\Delta\mu_n^*|, |\Delta v_n^*|^{1/2}\right\}$. For the simplicity of the proof, we only consider the setting when $M_n = |\Delta\mu_n^*|$ for all $n$ (by subsequence argument). The setting that $M_n = |\Delta v_n^*|^{1/2}$ for all $n$ can be argued in the similar fashion. Now, if we have

$$\max\left\{|\Delta\mu_n - \Delta\mu_n^*|, |\Delta v_n - \Delta v_n^*|^{1/2}\right\}/M_n \not\to 0,$$

then by dividing the numerator and denominator of $H_i$ with $\lambda_n\left(\max\left\{|\Delta\mu_n - \Delta\mu_n^*|, |\Delta v_n - \Delta v_n^*|^{1/2}\right\}\right)^i$ as $1 \leq i \leq 2$, we would achieve

$$1 = \frac{\max\left\{|\Delta\mu_n - \Delta\mu_n^*|^2, |\Delta v_n - \Delta v_n^*|\right\}}{\max\left\{|\Delta\mu_n - \Delta\mu_n^*|^2, |\Delta v_n - \Delta v_n^*|\right\}} \to 0,$$

a contradiction. Therefore, we must have

$$\max\left\{|\Delta\mu_n - \Delta\mu_n^*|, |\Delta v_n - \Delta v_n^*|^{1/2}\right\}/M_n \not\to 0 \tag{26}$$

as $n \to \infty$. Now, we further divide the argument under that setting of $M_n$ into two small cases

**Case 1.2.1:** $(\lambda_n^* - \lambda_n)|\Delta\mu_n^*|^4 \leq \lambda_n|\Delta\mu_n - \Delta\mu_n^*|^4$ for all $n$ (by subsequence argument). Since $M_n = |\Delta\mu_n^*|$, we would have $(\lambda_n^* - \lambda_n)|\Delta\mu_n^*|^i \leq \lambda_n|\Delta\mu_n - \Delta\mu_n^*|^i$ for all $n$ and $1 \leq l \leq 4$. From here, we obtain that

$$\frac{(\lambda_n^* - \lambda_n)|\Delta\mu_n^*|^4}{\lambda_n|\Delta\mu_n - \Delta\mu_n^*|} \leq \frac{\lambda_n|\Delta\mu_n - \Delta\mu_n^*|^4}{\lambda_n|\Delta\mu_n - \Delta\mu_n^*|} \to 0,$$

$$\frac{(\lambda_n^* - \lambda_n)|\Delta v_n^*|^2}{\lambda_n|\Delta\mu_n - \Delta\mu_n^*|} \leq \frac{(\lambda_n^* - \lambda_n)|\Delta\mu_n^*|^4}{\lambda_n|\Delta\mu_n - \Delta\mu_n^*|} \to 0.$$

If $|\Delta\mu_n - \Delta\mu_n^*|/|\Delta v_n - \Delta v_n^*|^{1/2} \not\to 0$, by diving both the numerator and the denominator of $H_1$ by $\lambda_n|\Delta\mu_n - \Delta\mu_n^*|$ and given that the new denominator goes to 0, the new numerator must converge to 0, i.e. we have

$$(\lambda_n^* - \lambda_n)\Delta\mu_n^*/\left\{\lambda_n(\Delta\mu_n - \Delta\mu_n^*)\right\} \to -1.$$

However, since we have $|(\Delta\mu_n - \Delta\mu_n^*)/\Delta\mu_n^* \to 0$, the above result would imply that

$$(\lambda_n^* - \lambda_n)|\Delta\mu_n^*|^4/\left\{\lambda_n|\Delta\mu_n - \Delta\mu_n^*|^4\right\} \to \infty,$$

which is a contradiction to the assumption of Case 1.2.1.1. As a consequence, we must have $|\Delta\mu_n - \Delta\mu_n^*|/|\Delta v_n - \Delta v_n^*|^{1/2} \to 0$. Now, we also have that

$$\frac{(\lambda_n^* - \lambda_n)|\Delta\mu_n^*|^4}{\lambda_n|\Delta v_n - \Delta v_n^*|^{i/2}} \lesssim \frac{\lambda_n|\Delta v_n - \Delta v_n^*|^2}{\lambda_n|\Delta\mu_n - \Delta\mu_n^*|^{i/2}} \to 0,$$

$$\frac{(\lambda_n^* - \lambda_n)|\Delta v_n^*|^4}{\lambda_n|\Delta v_n - \Delta v_n^*|^{i/2}} \leq \frac{(\lambda_n^* - \lambda_n)|\Delta\mu_n^*|^4}{\lambda_n|\Delta v_n - \Delta v_n^*|^{i/2}} \to 0.$$

for all $1 \leq i \leq 3$. Without loss of generality, we assume that $\Delta v_n - \Delta v_n^* > 0$ for all $n$. We denote $(-\Delta\mu_n^*) = q_1^n(\Delta v_n - \Delta v_n^*)$ and $\Delta v_n^* = q_2^n(\Delta v_n - \Delta v_n^*)$ for all $n$. From the result of (26), we would have $|q_1^n| \to \infty$. Given the above results, by dividing the numerators and the denominators of $H_\beta$ by $\lambda_n(\Delta v_n - \Delta v_n^*)^{\beta/2}$ for any $1 \leq \beta \leq 3$, we would have the new denominators go to 0. Therefore, all the new numerators of these $H_\beta$ also go to 0, i.e. we achieve the following system of limits

$$\frac{\lambda_n^* - \lambda_n}{\lambda_n}q_1^n \to 0, \quad \frac{\lambda_n^* - \lambda_n}{\lambda_n}\left\{(q_1^n)^2 + q_2^n\right\} + 1 \to 0, \quad \frac{\lambda_n^* - \lambda_n}{\lambda_n}\left(\frac{(q_1^n)^3}{6} + \frac{q_1^n q_2^n}{2}\right) \to 0.$$

Since $|q_1^n| \to \infty$, the last limit in the above system implies that $(\lambda_n^* - \lambda_n)\left(\frac{(q_1^n)^2}{3} + q_2^n\right)/\lambda_n \to 0$.

Combining this result with the second limit in the above system yields that $(\lambda_n^* - \lambda_n)(q_1^n)^2/\lambda_n + 3/2 \to 0$, which cannot happen. Therefore, Case 1.2.1 does not hold.

**Case 1.2.2:** $(\lambda_n^* - \lambda_n)|\Delta\mu_n^*|^4 > \lambda_n|\Delta\mu_n - \Delta\mu_n^*|^4$ for all $n$ (by subsequence argument). If $(\lambda_n^* - \lambda_n)|\Delta\mu_n^*|^4 \leq \lambda_n|\Delta v_n - \Delta v_n^*|^2$ for all $n$, the by using the same argument as that of Case 1.2.1, we quickly achieve the contradiction. Therefore, we must have $(\lambda_n^* - \lambda_n)|\Delta\mu_n^*|^4 > \lambda_n|\Delta v_n - \Delta v_n^*|^2$. Denote $(\Delta\mu_n - \Delta\mu_n^*) = m_1^n(-\Delta\mu_n^*)$, $(-\Delta v_n^*) = m_2^n(\Delta\mu_n^*)^2$, and $(\Delta v_n - \Delta v_n^*) = m_3^n(\Delta\mu_n^*)^2$. Since $M_n = |\Delta\mu_n^*|$, we would have $|m_i^n| \leq 1$ for all $1 \leq i \leq 3$. Denote $m_i^n \to m_i$ for all $1 \leq i \leq 3$ (by subsequence argument). The results of (26) lead to $m_1 = m_3 = 0$. Now by dividing both the numerator and denominator of $H_\beta$ by $(\lambda_n^* - \lambda_n)(-\Delta\mu_n^*)^\beta$ for any $1 \leq \beta \leq 4$, as the new denominators of $H_{|\beta|}$ do not go to $\infty$, we would also achieve that the new numerators of $H_{|\beta|}$ go to 0, i.e. the following system of limits hold

$$1 + \frac{\lambda_n^* - \lambda_n}{\lambda_n}m_1^n \to 0, \quad \left[1 + \frac{\lambda_n^* - \lambda_n}{\lambda_n}(m_1^n)^2\right] + m_2^n + \frac{\lambda_n^* - \lambda_n}{\lambda_n}m_3^n \to 0,$$

$$\left(1 + \frac{\lambda_n^* - \lambda_n}{\lambda_n}(m_1^n)^3\right)/6 + \left(m_2^n + \frac{\lambda_n^* - \lambda_n}{\lambda_n}m_1^n m_3^n\right)/2 \to 0,$$

$$\left(1 + \frac{\lambda_n^* - \lambda_n}{\lambda_n}(m_1^n)^4\right)/24 + \left(m_2^n + \frac{\lambda_n^* - \lambda_n}{\lambda_n}(m_3^n)^2\right)/4 + \left((m_2^n)^2 + \frac{\lambda_n^* - \lambda_n}{\lambda_n}(m_3^n)^2\right)/8 \to 0.$$

Combining with $m_1^n \to 0$, the first and third limit of the above system of limits imply that $m_2 = -1/3$. From here, the second and fourth limit yields that $1/6 + m_2 + m_2^2/2 = 0$, which is a contradiction. Therefore, Case 1.2.2 cannot hold.

**Case 2:** $\left(\lambda_n(|\Delta\mu_n|^2 + |\Delta v_n|) + \lambda_n^*(|\Delta\mu_n^*|^2 + |\Delta v_n^*|)\right)/\left\{\lambda_n(|\mu_n - \mu_n^*|^2 + |v_n - v_n^*|)\right\} \to \infty.$

We define

$$\overline{\mathcal{Q}}(G_n, G_{*,n}) = (\lambda_n^* - \lambda_n)(|\Delta\mu_n|^2 + |\Delta v_n|)(|\Delta\mu_n^*|^2 + |\Delta v_n^*|) + \left(\lambda_n(|\Delta\mu_n|^2 + |\Delta v_n|)\right.$$

$$+ \left.\lambda_n^*(|\Delta\mu_n^*|^2 + |\Delta v_n^*|)\right)\left(|\mu_n - \mu_n^*|^2 + |v_n - v_n^*|\right).$$

We will demonstrate that $\mathcal{Q}(G_n, G_{*,n}) \asymp \overline{\mathcal{Q}}(G_n, G_{*,n})$. In fact, from the above formulation of $\overline{\mathcal{Q}}(G_n, G_{*,n})$, we would have that

$$\overline{\mathcal{Q}}(G_n, G_{*,n}) \leq 2(\lambda_n^* - \lambda_n)(|\Delta\mu_n^*|^2 + |\Delta\mu_n - \Delta\mu_n^*|^2 + |\Delta v_n^*| + |\Delta v_n - \Delta v_n^*|)(|\Delta\mu_n^*|^2 + |\Delta v_n^*|)$$

$$+ 2\left(\lambda_n(|\Delta\mu_n|^2 + |\Delta v_n|) + \lambda_n^*(|\Delta\mu_n^*|^2 + |\Delta v_n^*|)\right)\left(|\mu_n - \mu_n^*|^2 + |v_n - v_n^*|\right)$$

$$\leq 2\mathcal{Q}(G_n, G_{*,n})$$

where the first inequality is due to the triangle inequality and basic inequality $(a + b)^2 \leq 2(a^2 + b^2)$ and the second inequality is due to the following result

$$(\lambda_n^* - \lambda_n)(|\Delta\mu_n - \Delta\mu_n^*|^2 + |\Delta v_n - \Delta v_n^*|) \leq \lambda_n^*\left(|\mu_n - \mu_n^*|^2 + |v_n - v_n^*|\right)$$

On the other hand, we also have that

$$2\overline{\mathcal{Q}}(G_n, G_{*,n}) \geq (\lambda_n^* - \lambda_n)(|\Delta\mu_n^*|^2 + |\Delta v_n^*|)(|\Delta\mu_n|^2 + |\Delta v_n| + |\mu_n - \mu_n^*|^2 + |v_n - v_n^*|)$$

$$+ \left(\lambda_n(|\Delta\mu_n|^2 + |\Delta v_n|) + \lambda_n^*(|\Delta\mu_n^*|^2 + |\Delta v_n^*|)\right)\left(|\mu_n - \mu_n^*|^2 + |v_n - v_n^*|\right)$$

$$\geq \mathcal{Q}(G_n, G_{*,n})/2$$

where the last inequality is due to triangle inequality and basic inequality $(a + b)^2 \leq 2(a^2 + b^2)$. Therefore, we conclude that $\mathcal{Q}(G_n, G_{*,n}) \asymp \overline{\mathcal{Q}}(G_n, G_{*,n})$. Now, since $H_\beta \to 0$ for all $1 \leq \beta \leq 8$, we would have that

$$F_\beta = \frac{\displaystyle\sum_{\alpha_1,\alpha_2} \frac{(\lambda_n^* - \lambda_n)(-\Delta\mu_n^*)^{\alpha_1}(-\Delta v_n^*)^{\alpha_2} + \lambda_n(\Delta\mu_n - \Delta\mu_n^*)^{\alpha_1}(\Delta v_n - \Delta v_n^*)^{\alpha_2}}{2^{|\alpha_2|}\alpha_1!\alpha_2!}}{\overline{\mathcal{Q}}(G_n, G_{*,n})} \to 0.$$

Similar to Case 1, under Case 2 we also consider two distincts setting of $\lambda_n^*/\lambda_n$

**Case 2.1:** $\lambda_n^*/\lambda_n \not\to \infty$. Under this case, we denote

$$M_n' := \max \left\{ |\Delta \mu_n|^2, |\Delta v_n|, |\Delta \mu_n^*|^2, |\Delta v_n^*| \right\}.$$

From the assumption of Case 2, we would have

$$|\Delta \mu_n - \Delta \mu_n^*|^2 / M_n' \to 0, \ |\Delta v_n - \Delta v_n^*| / M_n' \to 0. \tag{27}$$

Due to the symmetry between $(|\Delta \mu_n|^2, |\Delta v_n|)$ and $(|\Delta \mu_n^*|^2, |\Delta v_n^*|)$, we assume without loss of generality that $M_n' \in \max \left\{ |\Delta \mu_n|^2, |\Delta v_n| \right\}$. Under that assumption, we have two distinct cases

**Case 2.1.1:** $M_n' = |\Delta \mu_n|^2$ for all $n$ (by the subsequence argument). From (27), we have $|\Delta \mu_n - \Delta \mu_n^*| / |\Delta \mu_n| \to 0$, i.e., $\Delta \mu_n / \Delta \mu_n^* \to 1$. To be able to utilize the assumptions of Case 2, we will need to study the formulations of $F_\beta$ more deeply. In fact, when $\beta = 1$ simple calculation yields

$$A_1 := (\lambda_n \Delta \mu_n - \lambda_n^* \Delta \mu_n^*) / \overline{\mathcal{Q}}(G_n, G_{*,n}) \to 0.$$

When $\beta = 2$, we have

$$F_2 = \frac{(\lambda_n^* - \lambda_n)(\Delta \mu_n^*)^2 + \lambda_n(\Delta \mu_n - \Delta \mu_n^*)^2 + (\lambda_n^* - \lambda_n)(-\Delta v_n^*) + \lambda_n(\Delta v_n - \Delta v_n^*)}{\overline{\mathcal{Q}}(G_n, G_{*,n})} \to 0.$$

Combining with the result of $A_1$, it is clear that

$$\frac{(\lambda_n^* - \lambda_n)(\Delta \mu_n^*)^2 + \lambda_n(\Delta \mu_n - \Delta \mu_n^*)^2}{\overline{\mathcal{Q}}(G_n, G_{*,n})} \to \frac{(\lambda_n^* - \lambda_n)\Delta \mu_n \Delta \mu_n^*}{\overline{\mathcal{Q}}(G_n, G_{*,n})}.$$

Combining the above result with $F_2 \to 0$, we would have

$$A_2 := \frac{(\lambda_n^* - \lambda_n)\Delta \mu_n \Delta \mu_n^* + \lambda_n \Delta v_n - \lambda_n^* \Delta v_n^*}{\overline{\mathcal{Q}}(G_n, G_{*,n})} \to 0.$$

Now, we have two small cases

**Case 2.1.1.1:** $\Delta v_n / (\Delta \mu_n)^2 \to 0$ as $n \to \infty$. From (27), since we have $|\Delta v_n - \Delta v_n^*| / (\Delta \mu_n)^2 \to 0$, it implies that $\Delta v_n^* / (\Delta \mu_n)^2 \to 0$. Since $\Delta \mu_n / \Delta \mu_n^* \to 1$, we also have that $\Delta v_n^* / (\Delta \mu_n^*)^2 \to 0$. Now, from the formulations of $\overline{\mathcal{Q}}(G_n, G_{*,n})$ we have

$$(\lambda_n^* - \lambda_n)|\Delta v_n \Delta v_n^*| / \overline{\mathcal{Q}}(G_n, G_{*,n}) \leq |\Delta v_n| / |\Delta \mu_n|^2 \to 0,$$
$$(\lambda_n^* - \lambda_n)|\Delta v_n (\Delta \mu_n^*)^2| / \overline{\mathcal{Q}}(G_n, G_{*,n}) \leq |\Delta v_n| / |\Delta \mu_n|^2 \to 0,$$
$$(\lambda_n^* - \lambda_n)|\Delta v_n^* (\Delta \mu_n)^2| / \overline{\mathcal{Q}}(G_n, G_{*,n}) \leq |\Delta v_n^*| / |\Delta \mu_n^*|^2 \to 0. \tag{28}$$

From the result that $A_2 \to 0$, by multiplying $A_2$ with $\Delta \mu_n \Delta \mu_n^*$, we would also have that

$$\frac{(\lambda_n^* - \lambda_n)(\Delta \mu_n \Delta \mu_n^*)^2 + (\lambda_n \Delta v_n - \lambda_n^* \Delta v_n^*)\Delta \mu_n \Delta \mu_n^*}{\overline{\mathcal{Q}}(G_n, G_{*,n})} \to 0. \tag{29}$$

As $\lambda_n^*/\lambda_n \not\to \infty$, we have two distinct settings of $\lambda_n^*/\lambda_n$

**Case 2.1.1.1.1:** $\lambda_n^*/\lambda_n \not\to 1$. Using the result from (28) and the fact that $\Delta \mu_n / \Delta \mu_n^* \to 1$, we would obtain that

$$(\lambda_n \Delta v_n - \lambda_n^* \Delta v_n^*)\Delta \mu_n \Delta \mu_n^* / \overline{\mathcal{Q}}(G_n, G_{*,n}) \to 0.$$

Combining the above result with (29), it leads to

$$(\lambda_n^* - \lambda_n)(\Delta \mu_n \Delta \mu_n^*)^2 / \overline{\mathcal{Q}}(G_n, G_{*,n}) \to 0. \tag{30}$$

Combining (28) and (30), we would achieve that

$$\frac{(\lambda_n^* - \lambda_n)(|\Delta \mu_n|^2 + |\Delta v_n|)(|\Delta \mu_n^*|^2 + |\Delta v_n^*|)}{\overline{\mathcal{Q}}(G_n, G_{*,n})} \to 0.$$

From the formulation of $\overline{\mathcal{Q}}(G_n, G_{*,n})$, the above limit implies that

$$E := \frac{\left(\lambda_n(|\Delta\mu_n|^2 + |\Delta v_n|) + \lambda_n^*(|\Delta\mu_n^*|^2 + |\Delta v_n^*|)\right)\left(|\mu_n - \mu_n^*|^2 + |v_n - v_n^*|\right)}{\overline{\mathcal{Q}}(G_n, G_{*,n})} \to 1.$$

Due to the previous assumptions, we obtain that

$$E \lesssim \frac{\max\left\{\lambda_n(\Delta\mu_n)^2(\Delta\mu_n - \Delta\mu_n^*)^2, \lambda_n(\Delta\mu_n)^2(\Delta v_n - \Delta v_n^*)\right\}}{\overline{\mathcal{Q}}(G_n, G_{*,n})}.$$

By combining the results from (28) and (30), we can verify that

$$\frac{\lambda_n(\Delta\mu_n)^2(\Delta\mu_n - \Delta\mu_n^*)^2}{\overline{\mathcal{Q}}(G_n, G_{*,n})} \to \frac{(\lambda_n^* - \lambda_n)\left(-(\Delta\mu_n)^2(\Delta\mu_n^*)^2 + (\Delta\mu_n)^3\Delta\mu_n^*\right)}{\overline{\mathcal{Q}}(G_n, G_{*,n})} \to 0,$$

$$\frac{\lambda_n(\Delta\mu_n)^2(\Delta v_n - \Delta v_n^*)}{\overline{\mathcal{Q}}(G_n, G_{*,n})} \to \frac{(\lambda_n^* - \lambda_n)(\Delta\mu_n)^2\Delta v_n^*}{\overline{\mathcal{Q}}(G_n, G_{*,n})} \to 0.$$

Therefore, we achieve $E \to 0$, which is a contradiction. As a consequence, Case 2.1.1.1.1 cannot happen.

**Case 2.1.1.1.2:** $\lambda_n^*/\lambda_n \to 1$. Under this case, if we have

$$\max\left\{\frac{\lambda_n|\Delta\mu_n - \Delta\mu_n^*|^2}{(\lambda_n^* - \lambda_n)|\Delta\mu_n|^2}, \frac{\lambda_n|\Delta v_n - \Delta v_n^*|}{(\lambda_n^* - \lambda_n)|\Delta\mu_n|^2}\right\} \to \infty,$$

then we will achieve that

$$(\lambda_n^* - \lambda_n)(\Delta\mu_n\Delta\mu_n^*)^2/\overline{\mathcal{Q}}(G_n, G_{*,n}) \leq \min\left\{\frac{(\lambda_n^* - \lambda_n)|\Delta\mu_n^*|^2}{\lambda_n|\Delta\mu_n - \Delta\mu_n^*|^2}, \frac{(\lambda_n^* - \lambda_n)|\Delta\mu_n^*|^2}{\lambda_n|\Delta v_n - \Delta v_n^*|}\right\} \to 0.$$

From here, by using the same argument as that of Case 2.1.1.1.1, we will obtain $E \to 0$, which is a contradiction. Therefore, we would have that

$$\max\left\{\frac{\lambda_n|\Delta\mu_n - \Delta\mu_n^*|^2}{(\lambda_n^* - \lambda_n)|\Delta\mu_n|^2}, \frac{\lambda_n|\Delta v_n - \Delta v_n^*|}{(\lambda_n^* - \lambda_n)|\Delta\mu_n|^2}\right\} \not\to \infty. \tag{31}$$

With that assumption, it leads to $\overline{\mathcal{Q}}(G_n, G_{*,n}) \asymp (\lambda_n^* - \lambda_n)(\Delta\mu_n^*)^2(\Delta\mu_n)^2 \asymp (\lambda_n^* - \lambda_n)(\Delta\mu_n^*)^4$ as $\Delta\mu_n^*/\Delta\mu_n \to 1$. Now, we denote $\Delta\mu_n - \Delta\mu_n^* = \tau_1^n\Delta\mu_n^*$ and $\Delta v_n - \Delta v_n^* = \tau_2^n(\Delta\mu_n^*)^2$. From the assumption of Case 2.1.1.1, we would have that $\tau_1^n \to 0$ and $\tau_2^n \to 0$. By dividing both the numerator and the denominator of $F_3$ by $(\lambda_n^* - \lambda_n)(\Delta\mu_n^*)^3$, as the new denominators of $F_3$ goes to 0, we also obtain the numerator of this term goes to 0, i.e., the following holds

$$\left\{-1 + \frac{\lambda_n}{\lambda_n^* - \lambda_n}(\tau_1^n)^3\right\}/6 + \frac{\lambda_n}{2(\lambda_n^* - \lambda_n)}\tau_1^n\tau_2^n \to 0.$$

From (31), we have that $\lambda_n(\tau_1^n)^2/(\lambda_n^* - \lambda_n) \not\to \infty$ and $\lambda_n\tau_2^n/(\lambda_n^* - \lambda_n) \not\to \infty$. Therefore, since $\tau_1^n \to 0$ and $\tau_2^n \to 0$, we would achieve that $\lambda_n(\tau_1^n)^3/(\lambda_n^* - \lambda_n) \to 0$ and $\lambda_n\tau_1^n\tau_2^n/(\lambda_n^* - \lambda_n) \to 0$. By plugging these results to the above limit, it implies that $-1/6 = 0$, which is a contradiction. As a consequence, Case 2.1.1.1.2 cannot hold.

**Case 2.1.1.2:** $\Delta v_n/(\Delta\mu_n)^2 \not\to 0$ as $n \to \infty$. Under that case, we will only consider the setting that $\lambda_n^*/\lambda_n \to 1$ as the argument for other settings of that ratio can be argued in the similar fashion. Since we have $|\Delta v_n - \Delta v_n^*|/(\Delta\mu_n)^2 \to 0$, it leads to $\Delta v_n^*/(\Delta\mu_n)^2 \not\to 0$. Combining with $\Delta\mu_n/\Delta\mu_n^* \to 1$, it implies that as $n$ is large enough we would have

$$\max\left\{(\Delta\mu_n)^2, (\Delta\mu_n^*)^2\right\} \lesssim \min\left\{|\Delta v_n|, |\Delta v_n^*|\right\}. \tag{32}$$

According the formulation of $\overline{\mathcal{Q}}(G_n, G_{*,n})$, we achieve

$$\frac{(\lambda_n^* - \lambda_n)|\Delta v_n\Delta v_n^*|}{\overline{\mathcal{Q}}(G_n, G_{*,n})} \leq \min\left\{\frac{(\lambda_n^* - \lambda_n)|\Delta v_n^*|}{\lambda_n|\Delta v_n - \Delta v_n^*|}, \frac{(\lambda_n^* - \lambda_n)|\Delta v_n|}{\lambda_n^*|\Delta v_n - \Delta v_n^*|}, \frac{(\lambda_n^* - \lambda_n)|\Delta v_n^*|}{\lambda_n|\Delta\mu_n - \Delta\mu_n^*|^2},\right.$$

$$\left.\frac{(\lambda_n^* - \lambda_n)|\Delta v_n|}{\lambda_n^*|\Delta\mu_n - \Delta\mu_n^*|^2}\right\} = B.$$

If we have $B \to 0$, we would get $\dfrac{(\lambda_n^* - \lambda_n)|\Delta v_n \Delta v_n^*|}{\overline{\mathcal{Q}}(G_n, G_{*,n})} \to 0$. Combining with (32), we can check that all the results in (28), (30), and (31) hold. With similar argument as Case 2.1.1.1, we achieve $\overline{\mathcal{Q}}(G_n, G_{*,n})/\underline{\mathcal{Q}}(G_n, G_{*,n}) \to 0$, a contradiction. Therefore, we must have $B \not\to 0$. It implies that as $n$ is large enough we must have

$$\max\{\lambda_n, \lambda_n^*\} \max\{|\Delta\mu_n - \Delta\mu_n^*|^2, |\Delta v_n - \Delta v_n^*|\} \lesssim (\lambda_n^* - \lambda_n) \min\{|\Delta v_n|, |\Delta v_n^*|\}.$$

Furthermore, as $(\lambda_n^* - \lambda_n)/\lambda_n \to 0$, we obtain $|\Delta v_n^*|/|\Delta v_n - \Delta v_n^*| \to \infty$ and $|\Delta v_n^*|/|\Delta\mu_n - \Delta\mu_n^*|^2 \to \infty$, i.e., $\Delta v_n/\Delta v_n^* \to 1$. With all of these results, we can check that $\overline{\mathcal{Q}}(G_n, G_{*,n}) \lesssim (\lambda_n^* - \lambda_n)|\Delta v_n^*|^2$. Denote $(\Delta v_n - \Delta v_n^*) = k_1^n |\Delta v_n^*|$, $(\Delta\mu_n - \Delta\mu_n^*) = k_2^n |\Delta v_n^*|^{1/2}$, and $\Delta\mu_n^* = k_3^n |\Delta v_n^*|^{1/2}$ for all $n$. From all the assumptions we have thus far, we get $k_1^n \to 0$, $k_2^n \to 0$, and $|k_3^n| \not\to \infty$. Additionally, as $B \not\to 0$, we further have $\lambda_n |k_1^n|/(\lambda_n^* - \lambda_n) \not\to \infty$ and $\lambda_n(k_2^n)^2/(\lambda_n^* - \lambda_n) \not\to \infty$. By dividing both the numerator and the denominator of $F_3$ and $F_4$ respectively by $(\lambda_n^* - \lambda_n)|\Delta v_n^*|^{3/2}$ and $(\lambda_n^* - \lambda_n)|\Delta v_n^*|^2$, as the new denominators of $F_3, F_4$ do not go to infinity, we obtain the new numerators of these terms go to 0, i.e., the following holds

$$\left\{-(k_3^n)^3 + \frac{\lambda_n}{\lambda_n^* - \lambda_n}(k_2^n)^3\right\}/6 + \left\{k_3^n + \frac{\lambda_n}{\lambda_n^* - \lambda_n}k_1^n k_2^n\right\}/2 \to 0,$$

$$\left\{(k_3^n)^4 + \frac{\lambda_n}{\lambda_n^* - \lambda_n}(k_2^n)^4\right\}/24 + \left\{-(k_3^n)^2 + \frac{\lambda_n}{\lambda_n^* - \lambda_n}k_1^n(k_2^n)^2\right\}/4 + \left\{1 + \frac{\lambda_n}{\lambda_n^* - \lambda_n}(k_1^n)^2\right\}/8 \to 0.$$

With the assumptions with $k_1^n, k_2^n$, and $k_3^n$, we would have

$$\frac{\lambda_n}{\lambda_n^* - \lambda_n}(k_2^n)^i \to 0, \quad \frac{\lambda_n}{\lambda_n^* - \lambda_n}k_1^n(k_2^n)^j \to 0, \quad \frac{\lambda_n}{\lambda_n^* - \lambda_n}(k_1^n)^2 \to 0$$

for any $3 \le i \le 4$ and $1 \le j \le 2$. If we denote $k_3^n \to k_3$, by combining all the above results we achieve the following system of equations

$$-k_3^3/6 + k_3/2 = 0, \quad k_3^4/24 - k_3^2/4 + 1/8 = 0,$$

which does not admit a solution, a contradiction. Hence, Case 2.1.1.2 cannot hold.

**Case 2.1.2:** $M_n' = |\Delta v_n|$ for all $n$ (by the subsequence argument). From (27), we would have $|\Delta v_n - \Delta v_n^*|/|\Delta v_n| \to 0$, i.e., $\Delta v_n/\Delta v_n^* \to 1$, and $|\Delta\mu_n - \Delta\mu_n^*|^2/|\Delta v_n| \to 0$. The argument under this case is rather similar to that of Case 2.1; therefore, we only sketch the key steps. By using the result that $A_1 \to 0$ and $A_2 \to 0$, we would obtain that

$$\frac{(\lambda_n^* - \lambda_n)(\Delta\mu_n^*)^4 + \lambda_n(\Delta\mu_n - \Delta\mu_n^*)^4}{24\overline{\mathcal{Q}}(G_n, G_{*,n})} \to \frac{(\lambda_n^* - \lambda_n)\Delta\mu_n\Delta\mu_n^*\left[(\Delta\mu_n)^2 - 3\Delta\mu_n\Delta\mu_n^* + 3(\Delta\mu_n^*)^2\right]}{24\overline{\mathcal{Q}}(G_n, G_{*,n})},$$

$$\frac{(\lambda_n^* - \lambda_n)(\Delta v_n^*)^2 + \lambda_n(\Delta v_n - \Delta v_n^*)^2}{8\overline{\mathcal{Q}}(G_n, G_{*,n})} \to \frac{(\lambda_n^* - \lambda_n)\Delta\mu_n^*\left[\Delta\mu_n\Delta v_n - \Delta\mu_n^*\Delta v_n - \Delta u_n\Delta v_n^*\right]}{8\overline{\mathcal{Q}}(G_n, G_{*,n})},$$

$$\frac{\lambda_n(\Delta\mu_n - \Delta\mu_n^*)^2(\Delta v_n - \Delta v_n^*)}{4\overline{\mathcal{Q}}(G_n, G_{*,n})} \to \frac{(\lambda_n^* - \lambda_n)\left[\Delta\mu_n\Delta\mu_n^*\Delta v_n^* - \Delta\mu_n\Delta\mu_n^*\Delta v_n + \Delta v_n\Delta v_n^*\right]}{4\overline{\mathcal{Q}}(G_n, G_{*,n})}.$$

As $F_4 \to 0$, we equivalently have

$$\begin{aligned}
A_4 \ := \ &(\lambda_n^* - \lambda_n)\left\{\frac{\Delta\mu_n\Delta\mu_n^*\left[(\Delta\mu_n)^2 - 3\Delta\mu_n\Delta\mu_n^* + 3(\Delta\mu_n^*)^2\right]}{24\overline{\mathcal{Q}}(G_n, G_{*,n})}\right.\\
&+ \ \left.\frac{\Delta\mu_n\Delta\mu_n^*\Delta v_n - 2(\Delta\mu_n^*)^2\Delta v_n - \Delta\mu_n\Delta\mu_n^*\Delta v_n^* + \Delta v_n\Delta v_n^*}{8\overline{\mathcal{Q}}(G_n, G_{*,n})}\right\} \to 0.
\end{aligned}$$

Under Case 2.1.2, we only consider the setting when $(\Delta\mu_n)^2/\Delta v_n \to 0$ as other settings of this term can be argued in the similar fashion as that of Case 2.1.2. Since $|\Delta\mu_n - \Delta\mu_n^*|^2/|\Delta v_n| \to 0$,

we have $(\Delta\mu_n^*)/\Delta v_n \to 0$. As $\Delta v_n/\Delta v_n^* \to 1$, we also further have that $(\Delta\mu_n^*)^2/\Delta v_n^* \to 0$ and $(\Delta\mu_n)^2/\Delta v_n \to 0$. Therefore, we have $\Delta\mu_n\Delta\mu_n^*/\Delta v_n \to 0$ and $\Delta\mu_n\Delta\mu_n^*/\Delta v_n^* \to 0$. Now, from the formulation of $\overline{Q}(G_n, G_{*,n})$, we achieve

$$(\lambda_n^* - \lambda_n)|\Delta\mu_n\Delta\mu_n^*|^2/\overline{Q}(G_n, G_{*,n}) \le |\Delta\mu_n|^2/|\Delta v_n^*|^2 \to 0,$$

$$(\lambda_n^* - \lambda_n)|\Delta v_n(\Delta\mu_n^*)^2|/\overline{Q}(G_n, G_{*,n}) \le |\Delta\mu_n^*|^2/|\Delta v_n^*| \to 0,$$

$$(\lambda_n^* - \lambda_n)|\Delta v_n^*(\Delta\mu_n)^2|/\overline{Q}(G_n, G_{*,n}) \le |\Delta\mu_n|^2/|\Delta v_n| \to 0,$$

$$(\lambda_n^* - \lambda_n)|\Delta\mu_n|^3|\Delta\mu_n^*|/\overline{Q}(G_n, G_{*,n}) \le |\Delta\mu_n||\Delta\mu_n^*|/|\Delta v_n^*| \to 0,$$

$$(\lambda_n^* - \lambda_n)|\Delta\mu_n||\Delta\mu_n^*|^3/\overline{Q}(G_n, G_{*,n}) \le |\Delta\mu_n||\Delta\mu_n^*|/|\Delta v_n| \to 0.$$

Combining these results with $A_4 \to 0$, we achieve $(\lambda_n^* - \lambda_n)\Delta v_n\Delta v_n^*/\overline{Q}(G_n, G_{*,n}) \to 0$. From here, we can easily verify that all the results in (31) hold. Thus, by using the same argument as that of Case 2.1.1, we would get $\overline{Q}(G_n, G_{*,n})/\underline{Q}(G_n, G_{*,n}) \to 0$, a contradiction. As a consequence, Case 2.1.2 cannot happen.

**Case 2.2:** $\lambda_n^*/\lambda_n \to \infty$. Remind that $M_n' = \max\{|\Delta\mu_n|^2, |\Delta v_n|, |\Delta\mu_n^*|^2, |\Delta v_n^*|\}$. We can verify that $\overline{Q}(G_n, G_{*,n}) \lesssim \lambda_n^*(M_n')^4$. By dividing both the numerator and the denominator of $A_1$ and $A_2$ respectively by $\lambda_n^*(M_n')^{1/2}$ and $\lambda_n^* M_n'$, given that the new denominators go to 0 we would obtain the new numerators also go to 0, i.e., we have the following results

$$\lambda_n\Delta\mu_n^n/\left\{\lambda_n^*(M_n')^{1/2}\right\} - \Delta\mu_n^*/(M_n')^{1/2} \to 0,$$

$$\left[(\lambda_n^* - \lambda_n)\Delta\mu_n\Delta\mu_n^* + \lambda_n\Delta v_n - \lambda_n^*\Delta v_n^*\right]/\{\lambda_n^* M_n'\} \to 0.$$

Since $\lambda_n/\lambda_n^* \to 0$, the first limit implies that $\Delta\mu_n^*/M_n' \to 0$. Combining this result with the second limit, we obtain $\Delta v_n^*/M_n' \to 0$. Therefore, we would have $M_n' = \max\{|\Delta\mu_n|^2, |\Delta v_n|\}$. Without loss of generality, we assume that $M_n' = |\Delta\mu_n|^2$ as the argument for other possibility of $M_n'$ can be argued in the similar fashion. With these assumptions, $|\Delta v_n - \Delta v_n^*|/|\Delta\mu_n|^2 \not\to \infty$, i.e., as $n$ is large enough we get $|\Delta v_n - \Delta v_n^*| \lesssim |\Delta\mu_n|^2$. Now, we have two distinct cases

**Case 2.2.1:** $\lambda_n^* \max\{|\Delta\mu_n^*|^2, |\Delta v_n^*|\}/(\lambda_n|\Delta\mu_n|^2) \to \infty$. Due to this assumption, we can check that as $n$ is large enough, $\overline{Q}(G_n, G_{*,n}) \asymp \lambda_n^*|\Delta\mu_n|^2 \max\{|\Delta\mu_n^*|^2, |\Delta v_n^*|\}$. If $\max\{|\Delta\mu_n^*|^2, |\Delta v_n^*|\} = |\Delta\mu_n^*|^2$ for all $n$, then by dividing both the numerator and denominator of $A_1$ by $\lambda_n^*\Delta\mu_n^*$, given that the new denominator of $A_1$ goes to 0, its new numerator must go to 0, i.e., we have

$$\lambda_n\Delta\mu_n/(\lambda_n^*\Delta\mu_n^*) \to 1,$$

which cannot hold since $\lambda|\Delta\mu_n|^2/(\lambda_n^*|\Delta\mu_n^*|^2) \to 0$ (assumption of Case 2.2.1) and $|\Delta\mu_n|/|\Delta\mu_n^*| \to \infty$. Therefore, we must have $\max\{|\Delta\mu_n^*|^2, |\Delta v_n^*|\} = |\Delta v_n^*|$ for all $n$. By dividing both the numerator and denominator of $A_2$ by $\lambda_n^*\Delta v_n^*$, as the new denominator of $A_2$ goes to 0, we would have

$$\frac{(\lambda_n^* - \lambda_n)\Delta\mu_n\Delta\mu_n^*}{\lambda_n^*\Delta v_n^*} + \frac{\lambda_n\Delta v_n}{\lambda_n^*\Delta v_n^*} - 1 \to 0.$$

Since $\frac{\lambda_n|\Delta v_n|}{\lambda_n^*|\Delta v_n^*|} \le \frac{\lambda_n|\Delta\mu_n|^2}{\lambda_n^*|\Delta v_n^*|} \to 0$ and $(\lambda_n^* - \lambda_n)/\lambda_n^* \to 1$, the above limit shows that $\Delta\mu_n\Delta\mu_n^*/\Delta v_n^* \to 1$. Since $(\Delta\mu_n)^2/|\Delta v_n^*| \to \infty$, it implies that $(\Delta\mu_n^*)^2/\Delta v_n^* \to 0$. Now, by combing the result that $A_1 \to 0$ and $A_2 \to 0$, since $F_3 \to 0$, we can verify that it is equivalent to

$$A_3 := \frac{\left[(\lambda_n^* - \lambda_n)\Delta\mu_n\Delta\mu_n^*(\Delta\mu_n - 2\Delta\mu_n^*)\right]/3 + (\lambda_n^* - \lambda_n)\Delta\mu_n^*\Delta v_n}{\overline{Q}(G_n, G_{*,n})} \to 0.$$

By dividing both the numerator and the denominator of $A_3$ by $\lambda_n^*\Delta\mu_n\Delta v_n^*$, we obtain

$$\frac{\left[(\lambda_n^* - \lambda_n)\Delta\mu_n\Delta\mu_n^*(\Delta\mu_n - 2\Delta\mu_n^*)\right]/3 + (\lambda_n^* - \lambda_n)\Delta\mu_n^*\Delta v_n}{\lambda_n^*\Delta\mu_n\Delta v_n^*} \to 0.$$

As $(\Delta\mu_n^*)^2/\Delta v_n^* \to 0$ and $\Delta\mu_n\Delta\mu_n^*/\Delta v_n^* \to 1$, the above limit leads to $\Delta\mu_n^*\Delta v_n/(\Delta\mu_n\Delta v_n^*) \to -1/3$. Now, by studying $A_4 \to 0$ with the assumption that $\overline{\mathcal{Q}}(G_n, G_{*,n}) \asymp \lambda_n^*|\Delta\mu_n|^2|\Delta v_n^*|$, we eventually get the equation $1/24 - 1/12 = 0$, which is a contradiction. Therefore, Case 2.2.1 cannot hold.

**Case 2.2.2:** $\lambda_n^* \max\left\{|\Delta\mu_n^*|^2, |\Delta v_n^*|\right\}/\lambda_n|\Delta\mu_n|^2 \not\to \infty$. Therefore, as $n$ is large enough, we would have $\lambda_n^* \max\left\{|\Delta\mu_n^*|^2, |\Delta v_n^*|\right\} \lesssim (\lambda_n|\Delta\mu_n|^2)$. Hence, we achieve under this case that $\overline{\mathcal{Q}}(G_n, G_{*,n}) \asymp \lambda_n|\Delta\mu_n|^4$. Denote $\Delta\mu_n^* = l_1^n\Delta\mu_n$, $\Delta v_n = l_2^n(\Delta\mu_n)^2$, and $\Delta v_n^* = l_3^n(\Delta\mu_n)^2$. From the assumptions of Case 2.2.2, we would have $l_1^n \to 0$ and $l_3^n \to 0$ while $l_2^n \not\to \infty$. Additionally, $\lambda_n^* \max\left\{(l_1^n)^2, |l_3^n|\right\}/\lambda_n \not\to 0$. By dividing the numerators and denominators of $A_i$ by $\lambda_n(\Delta\mu_n)^i$ for $1 \le i \le 3$, we achieve the following system of limits

$$\frac{\lambda_n^* l_1^n}{\lambda_n} - 1 \to 0, \quad \frac{(\lambda_n^* - \lambda_n)l_1^n}{\lambda_n} + l_2^n - \frac{\lambda_n^* l_3^n}{\lambda_n} \to 0, \quad \frac{\lambda_n^* - \lambda_n}{\lambda_n}\left\{\frac{l_1^n - (l_1^n)^2}{3} + l_1^n l_2^n\right\} \to 0. \quad (33)$$

As $l_1^n \to 0$, the first limit in the above system implies that $\lambda_n^*(l_1^n)^2/\lambda_n \to 0$. If we have $\max\left\{(l_1^n)^2, |l_2^n|\right\} = |l_1^n|^2$ for all $n$, the previous result would mean that $\lambda_n^* l_3^n/\lambda_n \to 0$. Therefore, the second limit in (33) demonstrates that $l_2^n \to -1$. However, plugging these results to the third limit in this system would yield $1/3 - 1 = 0$, which is a contradiction. Hence, we must have $\max\left\{(l_1^n)^2, |l_2^n|\right\} = |l_3^n|$ for all $n$. Under this setting, by denoting $\frac{\lambda_n^* l_3^n}{\lambda_n} \to a$ as $n \to \infty$, the first and second limit in (33) leads to $l_2^n \to a - 1$. With this result, the third limit in this system shows that $a = 2/3$. With these results, by dividing both the numerator and denominator of $A_4$ by $\lambda_n(\Delta\mu_n)^4$, we quickly achieve the equation $1/24 - 5/72 = 0$, which is a contradiction. Therefore, Case 2.2.2 cannot hold.

In sum, not all the coefficients of $\frac{\partial^{|\beta|}f}{\partial\mu^\beta}(x|\mu_n^*, v_n^*)$ as $1 \le |\beta| \le 8$ go to 0. From here, by using the same argument as that of Proposition C.1 and Proposition C.3, we achieve the result of part (b) of the proposition. As a consequence, we reach the conclusion of the theorem. $\qquad\square$

# D  Proofs for Convergence Rates of Parameter Estimation and Minimax Lower Bounds

In this appendix, we provide the proofs for the convergence rates of the MLE as well as the corresponding minimax lower bounds introduced in Section D.

## D.1  Proof of Theorem 4.1

(a) For any $G_1 = G_1(\lambda_1, \mu_1, \Sigma_1)$ and $G_2 = G_2(\lambda_2, \mu_2, \Sigma_2)$, we denote the following distance

$$\begin{aligned} d_1(G_1, G_2) &= \lambda_1||(\mu_1, \Sigma_1) - (\mu_2, \Sigma_2)||, \\ d_2(G_1, G_2) &= |\lambda_1 - \lambda_2|^2. \end{aligned}$$

Even though $d_2(G_1, G_2)$ is a proper distance, it is clear that $d(G_1, G_2)$ is not symmetric and only satisfies a weak triangle inequality, i.e. we have

$$d_1(G_1, G_3) + d_1(G_2, G_3) \ge \min\left\{d_1(G_1, G_2), d_1(G_2, G_1)\right\}.$$

Therefore, we will utilize the modification of Le Cam method for nonsymmetric loss in Lemma 6.1 of [16] to deal with such distance. We start with the following proposition

**Proposition D.1.** *Given that $f$ satisfies assumption (S.1) in Theorem 4.1, we achieve for any $r < 1$ that*

   *(i)* $\displaystyle\lim_{\epsilon\to 0}\inf_{G_1=(\lambda,\mu_1,\Sigma_1), G_2=(\lambda,\mu_2,\Sigma_2)}\left\{h(p_{G_1}, p_{G_2})/d_1^r(G_1, G_2) : d_1(G_1, G_2) \le \epsilon\right\} = 0.$

   *(ii)* $\displaystyle\lim_{\epsilon\to 0}\inf_{G_1=(\lambda_1,\mu,\Sigma), G_2=(\lambda_2,\mu,\Sigma)}\left\{h(p_{G_1}, p_{G_2})/d_2^r(G_1, G_2) : d_2(G_1, G_2) \le \epsilon\right\} = 0.$

*Proof.* (i) For any sequences $G_{1,n} = (\lambda_n, \mu_{1,n}, \Sigma_{1,n})$ and $G_{2,n} = (\lambda_n, \mu_{2,n}, \Sigma_{2,n})$, we have

$$
\begin{aligned}
h^2(p_{G_{1,n}}, p_{G_{2,n}}) &\leq \frac{1}{\lambda_n} \int \frac{(p_{G_{1,n}}(x) - p_{G_{2,n}}(x))^2}{f(x|\mu_{2,n}, \Sigma_{2,n})} dx \\
&= \lambda_n \int \frac{(f(x|\mu_{1,n}, \Sigma_{1,n}) - f(x|\mu_{2,n}, \Sigma_{2,n}))^2}{f(x|\mu_{2,n}, \Sigma_{2,n})} dx
\end{aligned}
$$

where the first inequality is due to $\sqrt{p_{G_{1,n}}(x)} + \sqrt{p_{G_{2,n}}(x)} > \sqrt{\lambda_n f(x|\mu_{2,n}, \Sigma_{2,n})}$. By Taylor expansion up to the first order, we have

$$
f(x|\mu_{1,n}, \Sigma_{1,n}) - f(x|\mu_{2,n}, \Sigma_{2,n}) = \sum_{|\alpha|=1} \frac{(\mu_{1,n} - \mu_{2,n})^{\alpha_1}(\Sigma_{1,n} - \Sigma_{2,n})^{\alpha_2}}{\alpha_1! \alpha_2!} \frac{\partial f}{\partial \mu^{\alpha_1} \partial \Sigma^{\alpha_2}}(x|\mu_{2,n}, \Sigma_{2,n})
$$

$$
+ \sum_{|\alpha|=1} \frac{(\mu_{1,n} - \mu_{2,n})^{\alpha_1}(\Sigma_{1,n} - \Sigma_{2,n})^{\alpha_2}}{\alpha_1! \alpha_2!} \int_0^1 \frac{\partial f}{\partial \mu^{\alpha_1} \partial \Sigma^{\alpha_2}}(x|\mu_{2,n} + t(\mu_{1,n} - \mu_{2,n}), \Sigma_{2,n} + t(\Sigma_{1,n} - \Sigma_{2,n})) dt
$$

Now, by choosing $\lambda_n^{1-2r} \|(\mu_{1,n}, \Sigma_{1,n}) - (\mu_{2,n}, \Sigma_{2,n})\|^{2-2r} \to 0$, and $\|(\mu_{1,n}, \Sigma_{1,n}) - (\mu_{2,n}, \Sigma_{2,n})\| \to 0$ and using condition (S.1), we can easily verify that $h(p_{G_{1,n}}, p_{G_{2,n}})/d_1^r(G_{1,n}, G_{2,n}) \to 0$. Therefore, we achieve the conclusion of part (i).

(ii) The argument for this part is essentially similar to that in part (i). In fact, for any two sequences $G'_{1,n} = (\lambda_{1,n}, \mu_n, \Sigma_n)$ and $G'_{2,n} = (\lambda_{2,n}, \mu_n, \Sigma_n)$, we also obtain

$$
\begin{aligned}
\frac{h^2(p_{G'_{1,n}}, p_{G'_{2,n}})}{d_2^{2r}(G'_{1,n}, G'_{2,n})} &\leq \frac{(\lambda_{1,n} - \lambda_{2,n})^{2-2r}}{(1 - \lambda_{1,n}) \wedge \lambda_{1,n}} \int \frac{(h_0(x|\mu_0, \Sigma_0) - f(x|\mu_n, \Sigma_n))^2}{h_0(x|\mu_0, \Sigma_0) + f(x|\mu_n, \Sigma_n)} dx \\
&\leq \frac{2(\lambda_{1,n} - \lambda_{2,n})^{2-2r}}{(1 - \lambda_{1,n}) \wedge \lambda_{1,n}}
\end{aligned}
$$

By choosing $(\lambda_{1,n} - \lambda_{2,n})^{2-2r}/\{(1 - \lambda_{1,n}) \wedge \lambda_{1,n}\} \to 0$, we also achieve the conclusion of part (ii). $\square$

Now, given $G_* = (\lambda^*, \mu^*, \Sigma^*)$ and $r < 1$. Let $C_0$ be any fixed constant. According to part (i) of Proposition D.1, for any sufficiently small $\epsilon > 0$, there exists $G'_* = (\lambda^*, \mu_1^*, \Sigma_1^*)$ such that $d_1(G_*, G'_*) = d_1(G'_*, G_*) = \epsilon$ and $h(p_{G_*}, p_{G'_*}) \leq C_0 \epsilon^r$. By means of Lemma 6.1 of [16], we achieve

$$
\inf_{\widehat{G}_n \in \Xi} \sup_{G \in \Xi} \mathbb{E}_{p_G}\left(\lambda^2 \|(\widehat{\mu}_n, \widehat{\Sigma}_n) - (\mu, \Sigma)\|^2\right) \geq \frac{\epsilon^2}{2}\left(1 - V(p_{G_*}^n, p_{G'_*}^n)\right).
$$

where $p_{G_*}^n$ denotes the density of the $n$-iid sample $X_1, \ldots, X_n$. From there,

$$
\begin{aligned}
V(p_{G_*}^n, p_{G'_*}^n) &\leq h(p_{G_*}^n, p_{G'_*}^n) \\
&= \sqrt{1 - \left(1 - h^2(p_{G_*}, p_{G'_*})\right)^n} \\
&\leq \sqrt{1 - (1 - C_0^2 \epsilon^{2r})^n}.
\end{aligned}
$$

Hence, we obtain

$$
\inf_{\widehat{G}_n \in \Xi} \sup_{G \in \Xi} \mathbb{E}_{p_G}\left(\lambda^2 \|(\widehat{\mu}_n, \widehat{\Sigma}_n) - (\mu, \Sigma)\|^2\right) \geq \frac{\epsilon^2}{2}\sqrt{1 - (1 - C_0^2 \epsilon^{2r})^n}.
$$

By choosing $\epsilon^{2r} = \dfrac{1}{C_0^2 n}$, we achieve

$$
\inf_{\widehat{G}_n \in \Xi} \sup_{G \in \Xi} \mathbb{E}_{p_G}\left(\lambda^2 \|(\widehat{\mu}_n, \widehat{\Sigma}_n) - (\mu, \Sigma)\|^2\right) \geq c_1 n^{-1/r}.
$$

for any $r < 1$ where $c_1$ is some positive constant. Using the similar argument, with the result of (ii) in Proposition D.1 we also immediately obtain the result $\inf_{\widehat{G}_n \in \Xi} \sup_{G \in \Xi} \mathbb{E}_{p_G}\left(|\widehat{\lambda}_n - \lambda|^2\right) \geq c_2 n^{-1/r}$. As a consequence, we reach the conclusion of part (a) of the theorem.

(b) The proof of this part is a direct consequence of Theorem 3.3 and Theorem 2.3. Indeed, for $\widehat{G}_n = (\widehat{\lambda}_n, \widehat{\mu}_n, \widehat{\Sigma}_n)$ being the MLE as in equation (3), we have

$$\mathbb{E}_{p_{G_*}}\left(|\widehat{\lambda}_n - \lambda^*| + \lambda^* \|(\widehat{\mu}_n, \widehat{\Sigma}_n) - (\mu^*, \Sigma^*)\|\right) \overset{\text{Thm 3.3}}{\lesssim} \mathbb{E}_{p_{G_*}} V(p_{\widehat{G}_n}, p_{G_*}) \leq \mathbb{E}_{p_{G_*}} h(p_{\widehat{G}_n}, p_{G_*})$$

$$\overset{\text{Thm 2.3}}{\lesssim} \frac{\log n}{\sqrt{n}}$$

Because all inequalities are uniform in $G_*$, we achieve the conclusion of part (b) of the theorem.

## D.2 Proof of Theorem A.1

(a) Similar to the proof argument of part (a) of Theorem 4.1, we define

$$d_3(G_1, G_2) = \lambda_1 \|(\Delta\mu_1, \Delta\Sigma_1)\| \|(\mu_1, \Sigma_1) - (\mu_2, \Sigma_2)\|,$$
$$d_4(G_1, G_2) = |\lambda_1 - \lambda_2| \|(\Delta\mu_1, \Delta\Sigma_1)\|^2.$$

for any $G_1 = G_1(\lambda_1, \mu_1, \Sigma_1)$ and $G_2 = G_2(\lambda_2, \mu_2, \Sigma_2)$. It is clear that both $d_3(G_1, G_2)$ and $d_4(G_1, G_2)$ still satisfy weak triangle inequality. To achieve the conclusion of this part, it suffices to demonstrate the following results

(i) There exists two sequences $G_{1,n} = (\lambda_n, \mu_{1,n}, \Sigma_{1,n}) \in \Xi_1(l_n)$ and $G_{2,n} = (\lambda_n, \mu_{2,n}, \Sigma_{2,n}) \in \Xi_1(l_n)$ such that $d_3(G_{1,n}, G_{2,n}) \to 0$ and $h(p_{G_{1,n}}, p_{G_{2,n}})/d_3^r(G_{1,n}, G_{2,n})$ as $n \to \infty$.

(ii) There exists two sequences $G'_{1,n} = (\lambda_{1,n}, \mu_n, \Sigma_n) \in \Xi_1(l_n)$ and $G'_{2,n} = (\lambda_{2,n}, \mu_n, \Sigma_n) \in \Xi_1(l_n)$ such that $d_4(G_{1,n}, G_{2,n}) \to 0$ and $h(p_{G'_{1,n}}, p_{G'_{2,n}})/d_4^r(G_{1,n}, G_{2,n})$ as $n \to \infty$.

for any $r < 1$. The proof argument for the above results can proceed in a similar fashion as that of Proposition D.1; therefore, it is omitted. We achieve the conclusion of part (a) of the theorem.

(b) Combining the result of Theorem 3.5 and the fact that $D(G, G_*) \asymp \overline{D}(G, G_*)$ for any $G$ and $G_*$, we immediately achieve the following convergence rates

$$\sup_{G_* \in \Xi} \mathbb{E}_{p_{G*}}\left((\lambda^*)^2 \|(\Delta\mu^*, \Delta\Sigma^*)\|^2 \|(\widehat{\mu}_n, \widehat{\Sigma}_n) - (\mu^*, \Sigma^*)\|^2\right) \lesssim \frac{\log^2 n}{n},$$

$$\sup_{G_* \in \Xi} \mathbb{E}_{p_{G*}}\left(\|(\Delta\widehat{\mu}_n, \Delta\widehat{\Sigma}_n)\|^2 \|(\Delta\mu^*, \Delta\Sigma^*)\|^2 |\widehat{\lambda}_n - \lambda^*|^2\right) \lesssim \frac{\log^2 n}{n}. \tag{34}$$

It is clear that the second result in (34) does not match with the second result in the conclusion of part (b) of the theorem. To circumvent this issue, we utilize the fact that $G_* \in \Xi_1(l_n)$. Indeed, notice that $(\widehat{\mu}_n, \widehat{\Sigma}_n) - (\mu^*, \Sigma^*) = (\Delta\widehat{\mu}_n, \Delta\widehat{\Sigma}_n) - (\Delta\mu^*, \Delta\Sigma^*)$, we have

$$\sup_{G_* \in \Xi} \frac{\mathbb{E}_{p_{G*}}\left\|(\Delta\widehat{\mu}_n, \Delta\widehat{\Sigma}_n) - (\Delta\mu^*, \Delta\Sigma^*)\right\|^2}{\|(\Delta\mu^*, \Delta\Sigma^*)\|^2} \lesssim \frac{\log^2 n}{n(\lambda^*)^2 \|(\Delta\mu^*, \Delta\Sigma^*)\|^4} \to 0. \tag{35}$$

Hence, by the AM-GM inequality, we have

$$\mathbb{E}_{p_{G*}} \|(\Delta\widehat{\mu}_n, \Delta\widehat{\Sigma}_n)\|^2 (\widehat{\lambda}_n - \lambda^*)^2$$

$$\geq \frac{1}{2}\|(\Delta\mu^*, \Delta\Sigma^*)\|^2 \mathbb{E}_{p_{G*}}(\widehat{\lambda}_n - \lambda^*)^2 - \mathbb{E}_{p_{G*}} \|(\Delta\widehat{\mu}_n, \Delta\widehat{\Sigma}_n) - (\Delta\mu^*, \Delta\Sigma^*)\|^2 (\widehat{\lambda}_n - \lambda^*)^2$$

$$= \frac{1}{2}\|(\Delta\mu^*, \Delta\Sigma^*)\|^2 \left(\mathbb{E}_{p_{G*}}(\widehat{\lambda}_n - \lambda^*)^2 - \frac{\mathbb{E}_{p_{G*}} \left\|(\Delta\widehat{\mu}_n, \Delta\widehat{\Sigma}_n) - (\Delta\mu^*, \Delta\Sigma^*)\right\|^2 (\widehat{\lambda}_n - \lambda^*)^2}{\|(\Delta\mu^*, \Delta\Sigma^*)\|^2}\right)$$

$$\gtrsim \|(\Delta\mu^*, \Delta\Sigma^*)\| \mathbb{E}_{p_{G*}}(\widehat{\lambda}_n - \lambda^*)^2, \tag{36}$$

uniformly in $G_*$, where in the last inequality we use (35) combining with the fact that $|\widehat{\lambda}_n - \lambda^*|$ is uniformly bounded by 2. Hence,

$$\mathbb{E}_{p_{G*}}\left(\|(\Delta\mu^*, \Delta\Sigma^*)\|^4 |\widehat{\lambda}_n - \lambda^*|^2\right) \lesssim \mathbb{E}_{p_{G*}}\left(\|(\Delta\widehat{\mu}_n, \Delta\widehat{\Sigma}_n)\|^2 \|(\Delta\mu^*, \Delta\Sigma^*)\|^2 |\widehat{\lambda}_n - \lambda^*|^2\right) \lesssim \frac{\log^2(n)}{n},$$

which is the conclusion of the theorem.

### D.3 Proof of Theorem A.2

(a) Similar to the proof argument of part (a) of Theorem 4.1, we define

$$
\begin{aligned}
d_5(G_1, G_2) &= \lambda_1 \|(\mu_1, \Sigma_1) - (\mu_2, \Sigma_2)\|^4, \\
d_6(G_1, G_2) &= |\lambda_1 - \lambda_2| \|(\Delta\mu_1, \Delta\Sigma_1)\|^4.
\end{aligned}
$$

for any $G_1 = G_1(\lambda_1, \mu_1, \Sigma_1)$ and $G_2 = G_2(\lambda_2, \mu_2, \Sigma_2)$. It is clear that $d_6(G_1, G_2)$ satisfies weak triangle inequality while $d_5(G_1, G_2)$ no longer satisfies weak triangle inequality. In particular, we have

$$
d_5(G_1, G_3) + d_5(G_2, G_3) \geq \frac{\min\{d_5(G_1, G_2), d_5(G_2, G_1)\}}{8}.
$$

A close investigation of Lemma 6.1 of [16] reveals that modified Le Cam method still works under this setting of $d_5$ metric. More specifically, for any $\epsilon > 0$ the following holds

$$
\inf_{\widehat{G}_n \in \Xi} \sup_{G \in \Xi_2(l_n)} \mathbb{E}_{p_G}\left( d_5^2(G, \widehat{G}_n) \right) \geq \frac{\epsilon^2}{128}\left\{ 1 - V(p_{G_1}^n, p_{G_2}^n) \right\}
$$

where $G_1, G_2 \in \Xi_2(l_n)$ such that $d_5(G_1, G_2) \wedge d_5(G_1, G_2) \geq \epsilon/4$. From here, to achieve the conclusion of part (a), it suffices to demonstrate for any $r < 1$ that

  (i) There exists two sequences $G_{1,n} = (\lambda_n, \mu_{1,n}, \Sigma_{1,n}) \in \Xi_2(l_n)$ and $G_{2,n} = (\lambda_n, \mu_{2,n}, \Sigma_{2,n}) \in \Xi_1(l_n)$ such that $d_5(G_{1,n}, G_{2,n}) \to 0$ and $h(p_{G_{1,n}}, p_{G_{2,n}})/d_5^r(G_{1,n}, G_{2,n})$ as $n \to \infty$.

  (ii) There exists two sequences $G'_{1,n} = (\lambda_{1,n}, \mu_n, \Sigma_n) \in \Xi_2(l_n)$ and $G'_{2,n} = (\lambda_{2,n}, \mu_n, \Sigma_n) \in \Xi_1(l_n)$ such that $d_6(G_{1,n}, G_{2,n}) \to 0$ and $h(p_{G'_{1,n}}, p_{G'_{2,n}})/d_6^r(G_{1,n}, G_{2,n})$ as $n \to \infty$.

Following the proof argument of Proposition D.1, we can quickly verify the above results. As a consequence, we reach the conclusion of part (a) of the theorem.

(b) From the discussion after Theorem 3.5, we can show that:

$$
\mathcal{Q}(G, G_*) \asymp |\lambda - \lambda^*|(\|\Delta\mu\|^2\|\Delta\Sigma\|)(\|\Delta\mu^*\|^2\|\Delta\Sigma^*\|) + (\|\mu - \mu^*\|^2 + \|\Sigma - \Sigma^*\|)(\lambda(\|\Delta\mu\|^2 + \|\Delta\Sigma\|) + \lambda^*(\|\Delta\mu^*\|^2 + \|\Delta\Sigma^*\|)).
$$

Hence, from Theorem 3.6 combining with Theorem 2.3, we have

$$
\sup_{G_*} \mathbb{E}_{p_{G_*}} (\lambda^*)^2 (\|\widehat{\mu}_n - \mu^*\|^4 + \|\widehat{\Sigma}_n - \Sigma^*\|^2)(\|\Delta\mu^*\|^4 + \|\Delta\Sigma^*\|^2) \lesssim \frac{\log^2(n)}{n}
$$

$$
\sup_{G_*} \mathbb{E}_{p_{G_*}} |\widehat{\lambda}_n - \lambda^*|^2 (\|\Delta\widehat{\mu}_n\|^4 \|\Delta\widehat{\Sigma}_n\|^2)(\|\Delta\mu^*\|^4 \|\Delta\Sigma^*\|^2) \lesssim \frac{\log^2(n)}{n}.
$$

Similar to the proof of Theorem A.1 and with the definition of $\Xi_2(l_2)$, we have

$$
\mathbb{E}_{p_{G_*}} |\widehat{\lambda}_n - \lambda^*|^2 (\|\Delta\widehat{\mu}_n\|^4 \|\Delta\widehat{\Sigma}_n\|^2) \gtrsim (\|\Delta\mu^*\|^4 \|\Delta\Sigma^*\|^2)\mathbb{E}_{p_{G_*}} |\widehat{\lambda}_n - \lambda^*|^2
$$

uniformly in $G_* \in \Xi_2(l_2)$. Hence,

$$
\sup_{G_* \in \Xi_2(l_2)} \mathbb{E}_{p_{G_*}} |\widehat{\lambda}_n - \lambda^*|^2 (\|\Delta\mu^*\|^8 \|\Delta\Sigma^*\|^4) \lesssim \frac{\log^2(n)}{n}.
$$

As a consequence, we obtain the conclusion of the theorem.

## E  Proofs for Auxiliary Results

**Lemma E.1.** *For any $r \geq 1$, we define*

$$
\begin{aligned}
D_r(G, G_*) &= \lambda\|(\Delta\mu, \Delta\Sigma)\|^r + \lambda^*\|(\Delta\mu^*, \Delta\Sigma^*)\|^r \\
&\quad - \min\{\lambda, \lambda^*\}\left( \|(\Delta\mu, \Delta\Sigma)\|^r + \|(\Delta\mu^*, \Delta\Sigma^*)\|^r - \|(\mu, \Sigma) - (\mu^*, \Sigma^*)\|^r \right)
\end{aligned}
$$

*for any $G$ and $G_*$. Then, we have $W_r^r(G, G_*) \asymp D_r(G, G_*)$ for any $r \geq 1$ where $W_r$ is the $r$-th order Wasserstein distance.*

*Proof.* Without loss of generality, we assume throughout the lemma that $\lambda < \lambda^*$. Therefore, we obtain from the formulation of $D_r(G, G_*)$ that

$$D_r(G, G_*) = (\lambda^* - \lambda)||(\Delta\mu^*, \Delta\Sigma^*)||^r + \lambda||(\mu, \Sigma) - (\mu^*, \Sigma^*)||^r.$$

Direct computation of $W_r^r(G, G_*)$ yields three distinct cases:

**Case 1:** If $||(\Delta\mu, \Delta\Sigma)||^r + ||(\Delta\mu^*, \Delta\Sigma^*)||^r \geq ||(\mu, \Sigma) - (\mu^*, \Sigma^*)||^r$, then

$$
\begin{aligned}
W_r^r(G, G_*) &= \lambda||(\Delta\mu, \Delta\Sigma)||^r + \lambda^*||(\Delta\mu^*, \Delta\Sigma^*)||^r \\
&\quad - \min\{\lambda, \lambda^*\}\left(||(\Delta\mu, \Delta\Sigma)||^r + ||(\Delta\mu^*, \Delta\Sigma^*)||^r - ||(\mu, \Sigma) - (\mu^*, \Sigma^*)||^r\right) \\
&= D_r(G, G_*).
\end{aligned}
$$

**Case 2:** If $||(\Delta\mu, \Delta\Sigma)||^r + ||(\Delta\mu^*, \Delta\Sigma^*)||^r < ||(\mu, \Sigma) - (\mu^*, \Sigma^*)||^r$ and $\lambda + \lambda^* \leq 1$, then

$$
\begin{aligned}
W_r^r(G, G_*) &= \lambda||(\Delta\mu, \Delta\Sigma)||^r + \lambda^*||(\Delta\mu^*, \Delta\Sigma^*)||^r \\
&= (\lambda^* - \lambda)||(\Delta\mu^*, \Delta\Sigma^*)||^r + \lambda(||(\Delta\mu, \Delta\Sigma)||^r + ||(\Delta\mu^*, \Delta\Sigma^*)||^r).
\end{aligned}
$$

From Cauchy-Schartz's inequality, we have $||(\Delta\mu, \Delta\Sigma)||^r + ||(\Delta\mu^*, \Delta\Sigma^*)||^r \gtrsim ||(\mu, \Sigma) - (\mu^*, \Sigma^*)||^r$. Therefore, under Case 2 we have $||(\Delta\mu, \Delta\Sigma)||^r + ||(\Delta\mu^*, \Delta\Sigma^*)||^r \asymp ||(\mu, \Sigma) - (\mu^*, \Sigma^*)||^r$, which directly implies that $W_r^r(G, G_*) \asymp D_r(G, G_*)$.

**Case 3:** If $||(\Delta\mu, \Delta\Sigma)||^r + ||(\Delta\mu^*, \Delta\Sigma^*)||^r < ||(\mu, \Sigma) - (\mu^*, \Sigma^*)||^r$ and $\lambda + \lambda^* > 1$, then

$$
\begin{aligned}
W_r^r(G, G_*) &= (1 - \lambda^*)||(\Delta\mu, \Delta\Sigma)||^r + (1 - \lambda)||(\Delta\mu^*, \Delta\Sigma^*)||^r \\
&\quad + (\lambda + \lambda^* - 1)||(\mu, \Sigma) - (\mu^*, \Sigma^*)||^r \\
&= (\lambda^* - \lambda)||(\Delta\mu^*, \Delta\Sigma^*)||^r + (1 - \lambda^*)(||(\Delta\mu, \Delta\Sigma)||^r + ||(\Delta\mu^*, \Delta\Sigma^*)||^r) \\
&\quad + (\lambda^* + \lambda - 1)||(\mu, \Sigma) - (\mu^*, \Sigma^*)||^r.
\end{aligned}
$$

Since $||(\Delta\mu, \Delta\Sigma)||^r + ||(\Delta\mu^*, \Delta\Sigma^*)||^r \asymp ||(\mu, \Sigma) - (\mu^*, \Sigma^*)||^r$, we achieve

$$(1 - \lambda^*)(||(\Delta\mu, \Delta\Sigma)||^r \asymp (1 - \lambda^*)||(\mu, \Sigma) - (\mu^*, \Sigma^*)||^r.$$

Therefore, we also have $W_r^r(G, G_*) \asymp D_r(G, G_*)$ under Case 3.

Combining the results from these cases, we reach the conclusion of the lemma. $\qquad\square$

## F    Discussion and Additional Experiments

### F.1    Parameter Changes with the Sample Size

In statistics and machine learning, researchers often want to know how many samples are enough to achieve some pre-specified $\epsilon$ error for the estimation of parameter $\theta$ in the fitted model. In the language of probability, we want to find an inequality such as $E||\widehat{\theta}_n - \theta|| < C * rate(n)$, where $rate(n)$ is a decreasing sequence in $n$ and $C$ does not depend on $n$. Usually, in the parametric models, we have $rate(n) = 1/\sqrt{n}$, and therefore it takes $C^2/\epsilon^2$ samples to achieve average $\epsilon$ error in estimation. In complex models such as hierarchical models or the multivariate deviated model that we consider in this paper, difficulties arise because of the *singularity* and *identifiability* of the model. For example, in Eq. (2), if $\lambda^* = 0$, any pair of $(\mu^*, \Sigma^*)$ yields the same model. Hence, when $\lambda^* \approx 0$, it should be harder to estimate $(\mu^*, \Sigma^*)$, and researchers may need more samples to have an accurate estimation for them. Notably, we have shown, for example in Theorem 4.1, the precise dependence of the convergence rate of $(\mu^*, \Sigma^*)$ on the magnitude of $\lambda^*$. In particular, we have

$$\mathbb{E}|\widehat{\lambda}_n - \lambda^*| \leq C\frac{\log n}{\sqrt{n}}, \quad \mathbb{E}\lambda^*\left\|(\widehat{\mu}_n, \widehat{\Sigma}_n) - (\mu^*, \Sigma^*)\right\| \leq C\frac{\log n}{\sqrt{n}},$$

where *C is a constant that does not depend on* $\lambda^*, \mu^*, \Sigma^*$ *and* $n$. Hence, one can have a good estimation (with error $\epsilon$) for $\lambda^*$ with $C^2/\epsilon^2$ samples, while he needs $C^2/(\epsilon * \lambda^*)^2$ samples to achieve such a good estimation for $(\mu^*, \Sigma^*)$. The simulation studies in the next section will make this clearer.

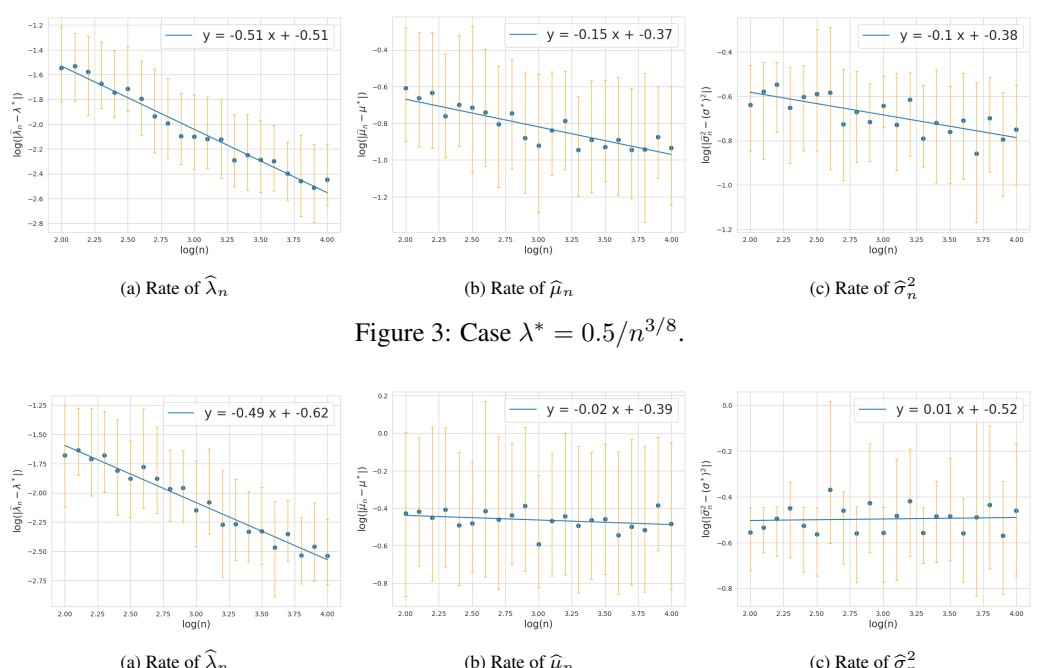

(a) Rate of $\widehat{\lambda}_n$          (b) Rate of $\widehat{\mu}_n$          (c) Rate of $\widehat{\sigma}_n^2$

Figure 3: Case $\lambda^* = 0.5/n^{3/8}$.

(a) Rate of $\widehat{\lambda}_n$          (b) Rate of $\widehat{\mu}_n$          (c) Rate of $\widehat{\sigma}_n^2$

Figure 4: Case $\lambda^* = 0.5/n^{1/2}$.

## F.2    Additional Experiments for the Distinguishable Settings

We have seen in the main text that in two cases where $\lambda^*$ is either fixed or decreasing with rate $n^{-1/4}$, the convergence rate of $\lambda^*$ is $C \times n^{-1/2}$, where the constants $C$ is the same for both cases. The convergence rate for $(\mu^*, \Sigma^*)$ is $C \times n^{-1/4}$ for the latter case, which is much slower than the parametric rate in the former case. This phenomenon is quite rare for parametric models. We want to further bring readers' attention to two more extreme cases:

1. $\lambda^* = 0.5/n^{3/8}$ as $n$ increases;

2. $\lambda^* = 0.5/n^{1/2}$ as $n$ increases,

where we consider the same $(\mu^*, \Sigma^*)$ with the experiments in the main text. The convergence rate for $(\lambda, \mu, \sigma^2)$ in both cases in the log domain can be seen in Figure 3 and Figure 4. Hence, in all cases, the rate of convergence for $\lambda^*$ is always of order $n^{-1/2}$, meanwhile, the rate for $(\mu^*, \Sigma^*)$ becomes slower as $\lambda^*$ tends to 0 faster. From the theoretical result, when $\lambda^* = 0.5/n^{3/8}$, we expect the rate for $(\mu^*, \Sigma^*)$ to be of order $n^{-1/8}$, which is demonstrated in Figure 3(b)&(c). At the extreme case $\lambda^* = 0.5/n^{1/2}$, it is even impossible to recover $(\mu^*, \Sigma^*)$ as $n \to \infty$ (cf. Figure 4(b)&(c)). This suggests practitioners collect more data when $\hat{\lambda}_n$ is small to have a good estimate for $(\mu^*, \Sigma^*)$. Finally, in the case $\hat{\lambda}_n$ is extremely small (of order $n^{-1/2}$), we suggest not to report the estimated values $(\widehat{\mu}_n, \widehat{\Sigma}_n)$, as they are highly uncertain.

## F.3    Non-distinguishable Settings

Finally, we consider the weakly identifiable and non-distinguishable setting here to demonstrate that the convergence rate for $\lambda^*$ can be slower than the parametric rate when $f$ is near $h_0$. Let both $h_0$ and $f$ belong to the location-scale Gaussian family. $h_0(x) = f(x|0, 1)$ and consider two cases of $(\lambda^*, \mu^*, \Sigma^*)$:

1. $\lambda^* = 0.25, \mu^* = 0.$ are fixed and $\sigma^* = 1 + n^{-1/8}$ as $n$ increases;

2. $\lambda^* = 0.25, \sigma^* = 1.$ are fixed and $\mu^* = n^{-1/8}$ as $n$ increases;

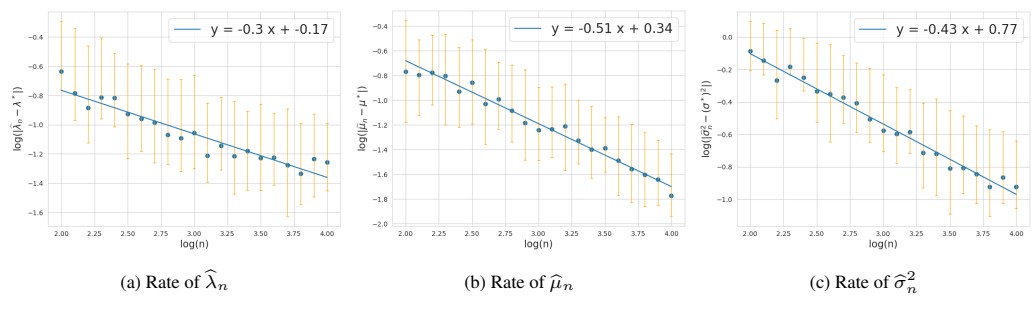

(a) Rate of $\widehat{\lambda}_n$         (b) Rate of $\widehat{\mu}_n$         (c) Rate of $\widehat{\sigma}_n^2$

Figure 5: Case 1: $\mu^* = \mu_0$ and $(\sigma^*)^2 \to \sigma_n^2$ in the rate $n^{-1/8}$

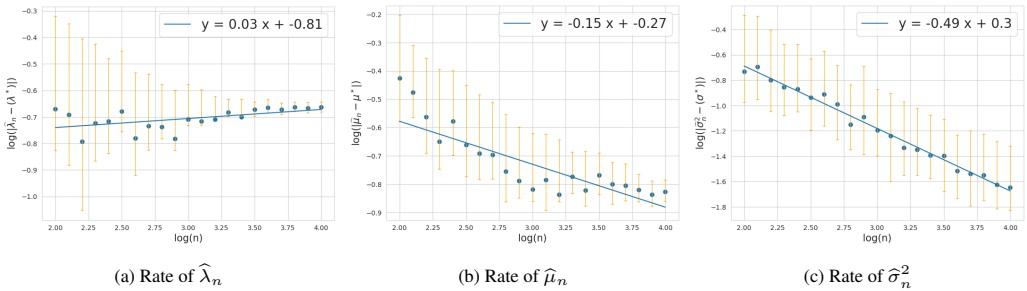

(a) Rate of $\widehat{\lambda}_n$         (b) Rate of $\widehat{\mu}_n$         (c) Rate of $\widehat{\sigma}_n^2$

Figure 6: Case 2: $\sigma^* = \sigma_0$ and $\mu^* \to \mu_0$ in the rate $n^{-1/8}$.

Recall that we have proved that $\left\{ \|\Delta\mu^*\|^4 + \|\Delta\Sigma^*\|^2 \right\} |\widehat{\lambda}_n - \lambda^*| = \mathcal{O}(n^{-1/2})$ and $\lambda^*(\|\Delta\mu^*\|^2 + \|\Delta\widehat{\mu}_n\|^2 + \|\Delta\Sigma^*\| + \|\Delta\widehat{\Sigma}_n\|)(\|\widehat{\mu}_n - \mu^*\|^2 + \|\widehat{\Sigma}_n - \Sigma^*\|) = \mathcal{O}(n^{-1/2})$, where there is a mismatch in the orders of convergence rates of the location and scale parameter. Notably, the rate of convergence for $\lambda^*$ also depends on the rate $\Delta\mu^*$ and $\Delta(\sigma^*)^2 \to 0$. The experiments do support this theoretical finding, where we have the rate for $\lambda^*$ is $\approx n^{-1/4}$ is the first case (as $\left\|\Delta(\sigma^*)^2\right\|^2 = O(n^{-1/4})$) and it does not convergence in the second case $\lambda^*$ is $\approx n^{-1/4}$ is the first case (as $\|\Delta\mu^*\|^4 = O(n^{-1/2})$). The mismatch rate for $\|\widehat{\mu}_n - \mu^*\|$ and $\|\widehat{\sigma}_n^2 - (\sigma^*)^2\|$ can also be seen clearly in the second case, where the rate for the scale parameter is still of the parametric rate, whereas it is slower for the mean.

