# OpenReview forum: "Minimax Optimal Rate for Parameter Estimation in Multivariate Deviated Models"
_NeurIPS.cc/2023/Conference — NeurIPS 2023 poster_

### Official Review · Reviewer_HL4W · 2023-07-02

**Soundness:** 4 excellent
**Presentation:** 3 good
**Contribution:** 3 good
**Rating:** 6
**Confidence:** 2

**Summary:**

The paper studies the optimal rate for multivariate deviated models. Specifically, they consider the model $(1-\lambda) h(x) + \lambda f(x|\mu,\Sigma)$, where $h$ is known and the goal is to estimate the other parameters. The authors propose to use the notion of *distinguishability* and study the convergence rate of parameters using MLE under both distinguishable and non-distinguishable cases. The authors present three pairs of upper and lower bounds for distinguishable, non-distinguishable but $f$ is strongly-identifiable, and distinguishable but $f$ is a family of location-covariance multivariate Gaussian distributions. Experiments are also provided to corroborate their theoretical results.

**Strengths:**

- The paper is clear. The notations are very consistent for such a lengthy work.
- There are plenty of explanations and discussions around each main result, making it an interesting paper to read.
- To the best of my knowledge, the proofs are technically sound and highly sophisticated. The notion of distinguishability and identifiability seems very suitable and intuitive.
- Lower bounds are also presented and match their convergence rate for all three cases considered.

**Weaknesses:**

- The idea of distinguishability is not wholly novel. Similar notions have been used in [1] for a different model, which in turn is derived from the notion of *identifiability* adopted in [2] and many other previous works.
- The section for related works is very concise.
- Given the classical parametric setting and the MLE estimation, I am not very sure if this paper would be a good match for neurips rather than a more statistically-focused journal.

[1] Do, Dat, Nhat Ho, and XuanLong Nguyen. "Beyond black box densities: Parameter learning for the deviated components." Advances in Neural Information Processing Systems 35 (2022): 28167-28178.
[2] Nguyen, XuanLong. "Convergence of latent mixing measures in finite and infinite mixture models." (2013): 370-400.

**Questions:**

- I totally understand the logic behind the organization of the sections but I would suggest shortening Section 3 and moving the results in Appendix A to the main content. Theorem 3.3/3.5/3.6 seem to be intermediate results for proving the convergence rates with artificial distances $\mathcal{K}$, $\mathcal{D}$, and $\mathcal{G}$. So I don't quite understand why they should take up nearly three pages, forcing the main results to be postponed to the appendix.
- Say we would like to adopt the model to fit some data in practice. How should we obtain $h_0$ which is claimed to be known in this paper? Or $h_0$ could just be good enough and then this model may take over? Would you please comment on how the results may provide possible guidance to the methodology?

---

> ### Author Rebuttal · Authors · 2023-08-09
>
> **Q1: The idea of distinguishability is not wholly novel. Similar notions have been used in [1] for a different model, which in turn is derived from the notion of identifiability adopted in [2] and many other previous works.**
>
> Thank you for raising this concern. We would like to emphasize that the other main contribution of the paper is to develop "uniform" convergence rates for parameters $(\lambda^*, \mu^*, \Sigma^*)$, compared to "point-wise" convergence rates from previous work. To see further why this is significant for hierarchical/mixture models such as the one in this paper, let us reconsider the deviated model:
> $$p^*(x) = (1-\lambda^*) h_0(x) + \lambda^* f(x|\mu^*, \Sigma^*).$$
> Suppose that $\lambda^* > 0$ is fixed, $\mu^*, \Sigma^*$ are fixed, and $h_0$ is distinguishable from $f$. Using the proof technique from previous work; it is possible to prove that the MLE $(\widehat{\lambda}_n, \widehat{\mu}_n, \widehat{\Sigma}_n)$ satisfies:
> $$\mathbb{E} |\widehat{\lambda}_n - \lambda^*| \leq C \left(\dfrac{\log(n)}{n}\right)^{1/2}, \mathbb{E} \|\widehat{\mu}_n - \mu^*\| \leq C \left(\dfrac{\log(n)}{n}\right)^{1/2}, $$
> and
> $$\mathbb{E} \|\widehat{\Sigma}_n - \Sigma^*\| \leq C \left(\dfrac{\log(n)}{n}\right)^{1/2},$$
> where $C$ is a constant that depends on $(\lambda^*, \mu^*, \Sigma^*)$ and does not depend on $n$. This is called point-wise convergence rates. However, when $\lambda^* = 0$, any pair of $(\mu^*, \Sigma^*)$ will give the same density, so parameter estimation is not possible. Hence, there must be a transition of convergence rates for $(\mu^*, \Sigma^*)$ as $\lambda^* \to 0$. We demonstrate this point precisely in Theorem 4.1, where the main result claims that
> $$\mathbb{E}||(\widehat{\mu}_n, \widehat{\Sigma}_n) - (\mu^*, \Sigma^*)|| \leq C \dfrac{1}{\lambda^*}\left(\dfrac{\log n}{n}\right)^{1/2},$$
> where $C$ is a constant does not depend on $(\lambda^*, \mu^*, \Sigma^*)$ and $n$. Hence, the convergence rates of $\mu^*$ and $\Sigma^*$
> are slower as $\lambda^*\to 0$. This uniform convergence rate allows scientists to obtain more concise confident intervals for parameter estimation and design better experiments in real life.
>
> To obtain those uniform bounds, the bounds between density distances and parameter distances that we develop in Section 3 need to be more refined compared to Wasserstein distances in the previous work. It leads to the divergence $\mathcal{K}, \mathcal{D}$ and $\mathcal{G}$, which seem to be quite artificial but technically meaningful. The novelty in the presented technique lies in the careful examination of the bound for each element $\lambda^*$, $\mu^*$, and $\Sigma^*$.
>
> In the paper, there are more challenging settings where $h_0$ can belong to the same family of distributions as $f$ and when the distinguishable condition does not hold. Each leads to different convergence rates for $\lambda^*, \mu^*$, and $\Sigma^*$. Then those uniform convergence rates are supported by simulation studies in Section 5 and Appendix F. The rates we obtain are even better than the rates obtained by using the method of moments in the literature. Please kindly refer to the general comment for more details.
>
> **Q2: The section for related works is very concise.**
>
> This is also a common concern raised by other reviewers. We decided to use the general response to explain the literature further. Please kindly refer to that. We will add the detailed literature review to the revision version.
>
> **Q3: Given the classical parametric setting and the MLE estimation, I am not very sure if this paper would be a good match for neurips rather than a more statistically-focused journal.**
>
> Thanks for your comment. In our opinion, MLE is the principle estimation for several Machine Learning models including Diffusion models and Transformers, therefore it is not outdated. The idea of deviated models is actually can be used in the big data regime, where $h_0$ may be a pre-trained large model such as Transformer, and $f(\cdot | \mu, \Sigma)$ is a small, low-rank model that is trained to adapt to some downstream task while freezing $h_0$ in training. The deviated proportion $\lambda^*$ can be studied to check how much the model needs to update to study the new and smaller downstream task. This idea is popular in Domain Adaptation recently [1, 2].
>
> **Q4. I would suggest shortening Section 3 and moving the results in Appendix A to the main content. Theorem 3.3/3.5/3.6 seem to be intermediate results for proving the convergence rates with artificial distances $\mathcal{K}, \mathcal{D}$ and $\mathcal{G}$.**
>
> Thanks for your suggestion. We will consider it and edit our paper accordingly.
>
> **Q5: Assume we would like to adopt the model to fit some data in practice. How should we obtain $h_0$ which is claimed to be known in this paper?
>
> In practice, we can see $h_0$ as rising from the domain adaptation problem, where it is estimated from a relevant data set. We then try to modify $h_0$ by a distribution in the vector-matrix family $f(\cdot | \mu, \Sigma)$ to estimate the density of the data set that we are working with [1, 2]. Also, kindly refer to the general response for another example in multiple testing problems, where we consider the distribution of the $p$ value that is obtained from numerous (independent) hypotheses tests. Hence, $h_0$ is uniform distribution on $[0, 1]$. The distribution under $H_1$ is unknown and required to estimate using the deviated component $f^*(x)$.
>
> **References**
>
> [1] Hu, Edward J., Yelong Shen, Phillip Wallis, Zeyuan Allen-Zhu, Yuanzhi Li, Shean Wang, Lu Wang, and Weizhu Chen. "Lora: Low-rank adaptation of large language models." arXiv preprint arXiv:2106.09685 (2021).
>
> [2] Jiang, Ziyu, Tianlong Chen, Xuxi Chen, Yu Cheng, Luowei Zhou, Lu Yuan, Ahmed Awadallah, and Zhangyang Wang. "DnA: Improving Few-Shot Transfer Learning with Low-Rank Decomposition and Alignment." In European Conference on Computer Vision, pp. 239-256. Cham: Springer Nature Switzerland, 2022.

---

> > ### Comment · Reviewer_HL4W · 2023-08-12
> >
> > Authors' response addressed my concerns. I am maintaining my score (6).

---

> > > ### Author Response · Authors · 2023-08-12
> > >
> > > We thank Reviewer HL4W for your positive evaluation of our paper after the rebuttal and for maintaining your score (6).

---

### Official Review · Reviewer_UU3o · 2023-07-05

**Soundness:** 3 good
**Presentation:** 3 good
**Contribution:** 3 good
**Rating:** 7
**Confidence:** 3

**Summary:**

In this paper, the authors establish the rate for estimating true parameters in the multivariate deviated model by using the MLE method. They mainly try to address two challenges encountered in deriving the rate of convergence for MLE estimators, i.e. 1) the interaction between the null hypothesis density $h_0$ and the alternative density function $f$, 2) the likelihood of the deviated proportion $\lambda$ vanishing to either extreme points of the interval [0, 1]. To this end, they develop the distinguishability condition to capture the linear independent relation between the function $h_0$ and the density function $f$, and derived the optimal convergence rate of the MLE under both distinguishable and non-distinguishable settings.

**Strengths:**

The paper is well-structured and effectively presents the problem setup, theoretical framework, and main results. The definitions and explanations of key concepts are well presented. The authors address a fundamental statistical problem and provide insights into the behavior of the MLE in the multivariate deviated model. The derived convergence rates and minimax rates contribute to the understanding of parameter estimation and hypothesis testing in complex data scenarios.

**Weaknesses:**

I feel that in the paper the comparison to existing literature is a bit limited. Particularly, how does this paper compare with the current literature on heterogeneous mixture detection?
In the experiment section, it seems that the setup is a bit limited with f being Gaussian. It would be great if the authors could show some more numerical results with more expanded scenarios.

**Questions:**

How does this paper compare with the current literature on heterogeneous mixture detection in terms of assumptions and results?

Can the results in Section 3.2.2 be extended to non-Gaussian distributions for f?

**Limitations:**

No potential negative societal impact

---

> ### Author Rebuttal · Authors · 2023-08-09
>
> **Q1: How does this paper compare with the current literature on heterogeneous mixture detection in terms of assumptions and results?**
>
> Thanks for your question. Different from the heterogeneous mixture detection literature where most of the results are under specific settings of $h_{0}$ and $f(x|\theta, \Sigma)$, in our paper we are able to provide general theories for more general settings of $h_{0}$ and $f(x|\theta, \Sigma)$. In particular, we develop the novel notion of distinguishability between $h_{0}$ and $f(x|\theta, \Sigma)$, which allows us to characterize the minimax uniform rates of parameter estimation under the general distinguishable settings of these functions. For the settings when these functions are not distinguishable, our results also cover a wide range of $h_{0}$ and $f(x|\theta, \Sigma)$ in practice, including strongly and weakly identifiable cases (e.g., multivariate Gaussian distribution and multivariate Student's t distribution with general covariance matrices).
>
> **Q2: In the experiment section, it seems that the setup is a bit limited with $f$ being Gaussian. It would be great if the authors could show some more numerical results with more expanded scenarios.**
>
> Thanks for your question. It is indeed true that the mixture model can be used with a much wider range of kernel densities. Although the Gaussian kernel is arguably the most popular, the mixture of Gamma distributions is also of interest when it comes to modeling heterogeneous data on the real line. We can consider the true generative model:
> $$p^*(x) = (1-\lambda^*) G(x | \alpha_0, \beta_0) + \lambda^* G(x | \alpha^*, \beta^*), $$
> where the shape-rate density Gamma is defined by:
> $$G(x|\alpha, \beta) = \dfrac{\beta^{\alpha}}{\Gamma(\alpha)} x^{\alpha - 1} e^{-\beta x},
> \quad \alpha, \beta > 0.$$
> The model is known to be strongly identifiable when $\alpha^* \neq \alpha_0$ or $|\beta^* - \beta_0| \neq 1$. So that the result in Theorem A.1 applies in this case. We will provide a simulation study for this model in Appendix F.
>
> **Q3: Can the results in Section 3.2.2 be extended to non-Gaussian distributions for $f$?**
>
> Thanks for your question. Most kernel densities, such as multivariate Student's t distribution and multivariate Laplace distribution, satisfy the second-order identifiability so that the results in Section 3.2.1 apply to them. For weakly identifiable families, the most popular kernel may be the location-scale Gaussians. Other examples are the Gamma distribution that we discussed previously or the skew normal distribution (cf. [1]). For Gamma distribution, it is interesting that the Gamma densities are only weakly identifiable in a zero Lebesgue measure set so that it is not of the main focus of our paper where we consider convergence rate from testing problems' perspective so that $\lambda^* \approx 0$, $\alpha^* \approx \alpha_0$, and $\beta^* \approx \beta_0$.
>
> For skew-normal distribution, it consists of three parameters: the location, scale, and skewness (shape) parameter. When the skewness parameter is 0, the skew-normal distribution becomes the normal distribution. This distribution is useful for modeling asymmetric data. The skew-normal distribution possesses more complex algebraic structures among the location, scale, and skewness parameters (via partial differential equations of the skew-normal distribution with respect to these parameters) than those in the location-scale Gaussian distribution (c.f. equations (2) and (3) in [1]). Therefore, the theoretical results in Section 3.2.2. will be richer and more complicated for skew-normal distribution than those for Gaussian distribution. We leave detailed theoretical analyses of both Gamma and skew-normal distributions for future work.
>
> **References**
>
> [1] N. Ho and L. Nguyen. Singularity structures and impacts on parameter estimation in finite mixtures of distributions. SIAM Journal on Mathematics of Data Science, 2019.

---

> > ### Comment · Reviewer_UU3o · 2023-08-16
> >
> > Thank you very much for the response. I will maintain my score.

---

> > > ### Author Response · Authors · 2023-08-16
> > > **Response to Reviewer UU3o**
> > >
> > > We thank Reviewer UU3o for your positive evaluation of our paper after the rebuttal and for maintaining your score (7).
> > >
> > > Best,
> > >
> > > The Authors

---

### Official Review · Reviewer_bb5i · 2023-07-06

**Soundness:** 2 fair
**Presentation:** 2 fair
**Contribution:** 3 good
**Rating:** 6
**Confidence:** 2

**Summary:**

## Summary

The authors study the minimax rate for parameter recovery in deviated multivariate models.
In this setting, we observe samples from a mixture (of *unknown* weight \lambda) of a "null" distribution h_0 and a distribution from a parametric family f( | \mu, \Sigma).
The goal is to recover from n samples the (\lambda, \mu, \Sigma).

## Contribution

The authors study the MLE performance in various regimes (depending on how "far" is h_0 from the parametric family of distributions.
Their analysis is very tight, leading to obtaining the minimax rates.


**Strengths:**

I like the result: it is mathematically clean and leads to tight results. It is always nice to see the minimax rates for new problems.

**Weaknesses:**

I am unfortunately failing to conclude the position of the paper in the literature.
Are the authors the first to obtain results in this setting?
If yes, please explain (much) more why studying the model is interesting/important.
If not, please compare thoroughly the results with the existing ones.

Also, are the techniques used new in any way? Or are the results simply an application of known techniques?

All in all, I think the paper needs a much more thorough literature review.

**Questions:**

See above.

**Limitations:**

See above.

---

> ### Author Rebuttal · Authors · 2023-08-09
>
> **Q1: Are the authors the first to obtain results in this setting? If yes, please explain why studying the model is important. If not, please compare thoroughly the results with the existing ones.**
>
> Thanks for your questions. We provide below a more thorough comparison with other existing results and hope that it helps to clarify the position of this paper in the literature.
>
> 1. This model gained some of its early interest from testing and detection problems in biology [1], where the authors consider the semi-parametric setting:
> $$p^*(\cdot) = \lambda^* h_0(\cdot) + (1-\lambda^*) f^*(\cdot - \mu_2^*),$$
> where $h_0$ is known, and $\lambda^*$, $\mu^*$, $f^*$ are to be estimated from the data. The density $f$ is further assumed to be symmetric but non-parametric. Then the identifiability, consistency and asymptotic normality of parameters are studied.
>
> 2. In [2], the authors consider the parametric location deviated model:
> $$p^*(\cdot) = (1-\lambda^*) N(\cdot | 0, 1) + \lambda^* N(\cdot | \mu^*, 1),$$
> where the density $N$ is standard Gaussian, the true parameter $\lambda^*$ and $\mu^*$ are of interest and would be recovered given data. They specifically consider $l^2$ estimation and provide the uniform bound for parameter estimation (i.e., holds for all $(\lambda^*, \mu^*)$, which is
> similar to our result). It is worth noting that the setting in our paper is more general in the sense that the known component $N(0, 1)$ can be assumed to be any density $h_0$ and the estimated component $N(\mu^*, 1)$ can be any vector-matrix family $f(\cdot | \mu^*, \Sigma^*)$.
>
> 3. In [3, 4, 5], a similar setting is considered:
> $$p^*(\cdot) = (1- \epsilon_n) N(\cdot | 0, 1) + \epsilon_n N(\cdot | \mu_n, 1),$$
> in which the main question is: How small $\epsilon_n$ can be compared to the sample size $n$ in order to reliably test (or
> detect) $\epsilon > 0$ against $\epsilon = 0$. In the so-called dense setting, where they assume $\epsilon_n \asymp n^{-\beta}$ for $\beta\in (0,1/2)$, it is possible to do so when $\epsilon_n \mu_n \gtrsim (\log(n)/n)^{1/2}$, which matches the result that we have in Theorem A.1. We go beyond this setting by also considering varying covariance of the second component $N(\cdot | \mu_n, \sigma_n^2)$. Theorem A.3 implicitly says that it is possible to estimate parameters when $\epsilon_n \mu_n^2 \gtrsim (\log n/n)^{1/2}$ and $\epsilon_n \sigma_n^2 \gtrsim (\log n/n)^{1/2}$ Additional experiments in Section F.2 and F.3 also support this finding.
>
> 4. Although this model is statistically interesting and worth studying in the classical sense, please note that its idea is still used in the Machine Learning community [9, 10]. Specifically, $h_0$ can be a pre-trained Large Model, and we wish to adapt it to some specific downstream task. A popular solution is to consider the deviated model with small deviated weight $\lambda^*$ and a simple model $f(\cdot | \mu^*, \Sigma^*)$, which costs less to train compared to $h_0$. By doing that, it does not require to re-train $h_0$ but still can borrow some knowledge learned from it. We hope the theoretical result that we build in our model can shed some light to the estimation problem for this Low-rank adaptation technique.
>
> **Q2: Are the techniques used new in any way? Or are the results simply an application of known techniques?**
>
> The core of the techniques in our paper is from [6, 7, 8], which allows us to develop lower bounds of density distance by parameter distance using the notion of "identifiability" or "distinguishability." A novel technique here is we look at the convergence rate of each parameter in detail instead of relying on the Wasserstein distance, which could not capture the specific rate for each parameter. Besides, the novelty that we wish to highlight is that our bounds are uniform in the parameter space $(
> \lambda^*, \mu^*, \Sigma^*)$. Because of that, we can detect whether $(\mu^*, \Sigma^*)$ is estimable when $\lambda^*\to 0$, which is a singular point of the model (i.e., when $\lambda^*=0$, every pair of $(\mu^*, \Sigma^*)$ give the same density). Both the minimax rate and the convergence rate of $(\mu^*, \Sigma^*)$ are then developed in this setting.
>
> **Q3: I think the paper needs a much more thorough literature review.**
>
> Thanks for your comment. We hope that the answer to your first question can partially help to clarify the current literature. Also, kindly refer to a more detailed review of our general answer. We will include them in the revision of the paper.
>
> [1] L. Bordes. Semiparametric estimation of a two-component mixture model where one component is known. Scandinavian journal of statistics, 2006.
>
> [2] S. Gadat. Parameter recovery in two-component contamination mixtures: The l2 strategy. In Annales de l’Institut Henri Poincaré, Probabilités et Statistiques, volume 56, pages 1391–1418. Institut Henri Poincaré, 2020.
>
> [3] T. Cai. Optimal detection of heterogeneous and heteroscedastic mixtures. Journal of the Royal Statistical Society: Series B,  2011.
>
> [4] T. Cai. Estimation and confidence sets for sparse normal mixtures. Annals of Statistics, 2007.
>
> [5] T. Cai. Optimal detection of sparse mixtures against a given null distribution. IEEE Transactions on Information Theory, 2014.
>
> [6] Chen, J. H. Optimal rate of convergence for finite mixture models. Annals of Statistics, 1995
>
> [7] N. Ho. Convergence rates of parameter estimation for some weakly identifiable finite mixtures. Annals of Statistic, 2016.
>
> [8] P. Heinrich and J. Kahn. Strong identifiability and optimal minimax rates for finite mixture estimation. Annals of Statistics, 2018.
>
> [9] Hu, Edward J. Lora: Low-rank adaptation of large language models. arXiv preprint, 2021.
>
> [10] Jiang, Ziyu. DnA: Improving Few-Shot Transfer Learning with Low-Rank Decomposition and Alignment. In European Conference on Computer Vision, 2022.

---

> > ### Comment · Reviewer_bb5i · 2023-08-14
> >
> > I thank the authors for their response which addressed my concerns. I am maintaining my score (6), but with low confidence, subject to the authors adding all the above literature comments in their paper.

---

> > > ### Author Response · Authors · 2023-08-14
> > >
> > > We thank Reviewer bb5i for your positive evaluation of our paper after the rebuttal and for maintaining your score of weak accept (6).

---

### Official Review · Reviewer_9DGR · 2023-07-06

**Soundness:** 3 good
**Presentation:** 3 good
**Contribution:** 3 good
**Rating:** 6
**Confidence:** 4

**Summary:**

The paper studies the problem of parameter recovery in the multivariate deviated model where the data is generated according to the following distribution:
$$
    (1 - \lambda) h_0 (x) + \lambda f(x | \mu, \sigma)
$$
where $f$ belongs to a mean-variance family and $h_0$ is known. One prominent example of such a family is the family of Gaussian distributions.

The paper studies the recovery problem under three settings. The first is the distinguishable setting where $h_0$ and the density $f$ are distinguishable (essentially $h_0$ cannot be written as a linear combination of two distributions from the family the family) where statistical recovery is guaranteed at a $\sqrt{n}$ rate. In the second setting of strong identifiability, $f$ is assumed to belong to the mean-variance family and here the convergence rates depend on the closeness of $\mu_0, \Sigma_0$, the parameters of $h_0$, to $\mu, \Sigma$, the parameters of the unknown mixture component. Finally, in the third setting where the mean-variance family is the Gaussian distribution where different convergence behavior is observed from the strongly identifiable setting where a second order PDE guarantees improved performance.

From a technical standpoint, the algorithm (MLE) is analyzed in the following way. First, they show that under some mild assumptions on the function class, the MLE solutions approximate the distribution in Hellinger-distance. Then, by noting the relationship between the Hellinger and TV distances, the paper then shows that the for this class of distributions, the distance between the parameters is upper bounded by a constant multiple of the TV distance. The first step follows by standard empirical process theory. The second, however, relies on some intricate recently developed machinery. Roughly speaking, one first shows that the TV and parameter-distances approximate each other locally where the limit of the neighborhood is taken to $0$. Subsequently, a short analytic argument leads to a global approximation guarantee. This technique, while inspired prior work, still takes significant care to execute.

Overall, the results in the paper are interesting and the technical contributions seem strong. The fact that the statistical performance in this setting may be distinguished from algorithms that operate on mixture models where both components are unknown is also intriguing. However, Theorem 3.6 is only proved for the univariate setting (Appendix C3) while the rest of the paper focuses on the multivariate setting.


**Strengths:**

See main review

**Weaknesses:**

See main review

**Questions:**

See main review

**Limitations:**

Yes

---

> ### Author Rebuttal · Authors · 2023-08-09
>
> **Q1: Theorem 3.6 is only proved for the univariate setting (Appendix C3) while the rest of the paper focuses on the multivariate setting.**
>
> Thanks for your comment. Although the proof of Theorem 3.6 is only proved for the univariate setting, it can be adapted to high-dimensional settings along with some according changes of notations to those settings. Since there are too many scenarios arising in that proof, we consider the univariate setting to avoid the unnecessary complexity of notations which might make the main arguments implicit and the proof untidy.

---

> > ### Comment · Reviewer_9DGR · 2023-08-18
> >
> > Thank you for your response. I will retain my current evaluation.

---

> > > ### Author Response · Authors · 2023-08-18
> > > **Thank you**
> > >
> > > We thank Reviewer 9DGR for your positive evaluation of our paper after the rebuttal and for maintaining your score of weak accept (6) with high confidence.

---

### Official Review · Reviewer_1EFM · 2023-07-07

**Soundness:** 2 fair
**Presentation:** 3 good
**Contribution:** 2 fair
**Rating:** 3
**Confidence:** 3

**Summary:**

This paper tackles the issue of parameter estimation in the deviated Gaussian mixture of experts problem using the Maximum Likelihood Estimation (MLE) method. The authors propose new distances and analyze the convergence of MLE under distinguishable and non-distinguishable conditions.

**Strengths:**

This paper is in relatively good shape. The results seem to be solid.

**Weaknesses:**

The major weakness is the novelty.

This paper basically considers a much simpler case than the paper https://huynm99.github.io/Deviated_MoE.pdf.

They consider multiple $k$ while this paper considers a single $k$.

The definitions, results,  organization, and even the notations are almost the same. For example, the hellinger distance and TV distance (quite strange to me but adopted by both papers interesting), although this paper changes the distance from $D$.






**Questions:**


1. what is the mini-max rate of this problem? how close is the current rate to the min-max rate?

2. What is the difference and technical novelty compared to https://huynm99.github.io/Deviated_MoE.pdf

---

> ### Author Rebuttal · Authors · 2023-08-09
>
> **Q1: The major weakness is the novelty.**
>
> Thanks for your comment. We would like to refer you to the General Response section for our elaboration on the novelty of this paper.
>
> **Q2: This paper basically considers a much simpler case than the paper [1]. In particular, the authors in that paper consider multiple $k$ while this paper considers a single $k$.**
>
> Thanks for your comment. However, we gracefully disagree with the viewpoint that the problem considered our paper is a simpler case than that in [1] for the following reasons:
>
> Based on the formulations of the models considered in two papers (ours and [1]), it might seem that our model is simpler as we test whether data are sampled from a distribution with known density $h_0$ (null hypothesis) or to another single distribution (alternative hypothesis), whereas the authors in [1] also test the previous null hypothesis but against another alternative hypothesis which says that data are generated from a mixture of experts.
>
> Nevertheless, it turns out to be untrue as the nature of two papers are different. In particular, in [1], the ground-truth parameters are not assumed to change with the sample size, therefore, the main goal is to derive the point-wise convergence rates for parameter estimation. By contrast, we do impose this assumption on our problem setup as mentioned in Question 1. Thus, the objective of our paper is to characterize the uniform convergence rates, which is demanding but more precise than the point-wise counterparts.
>
> Hence, our work is not a simpler case of [1], but it lays a foundation to capture uniform convergence rates for parameter estimation as well as minimax lower bounds in the deviated Gaussian mixture of experts. However, these directions are beyond the scope of our paper and we leave them for future work.
>
> **Q3: The definitions, results, organization, and even the notations of this paper and [1] are almost the same.**
>
> Thanks for your comment, but we gracefully disagree with this claim for the following reasons:
>
> 1) Regarding the definitions and notations: notions introduced in our paper such as identifiability and distinguishability or technical tools like the Hellinger distance and Total Variation distance are commonly used in the literature of mixture models to capture the convergence rates for parameter estimation, namely [2, 3]. Thus, we strongly believe that the usage of those ingredients should be considered normal rather than as a weakness.
>
> 2) Regarding the results: as we stated in our responses to Question 1 and Question 2, the results introduced in this paper are novel and totally different from those in [1].
>
> 3) Regarding the organization: our paper is organized in a different way compared to [1]. In particular, while we present the Total Variation lower bounds in Section 3, and then ntroduce uniform and minimax rates in Section 4, [1] respectively provides the point-wise rates under distinguishable and non-distinguishable settings in those sections. Moreover, our Section 5 is devoted for experiments, whereas [1] use this section for proof sketch.
>
>
> **Q4: What is the mini-max rate of this problem? how close is the current rate to the min-max rate?**
>
> Thanks for your questions. Actually, we already provided the result of minimax lower bounds in Section 4 and Appendix A of our paper. For example, under the distinguishable settings, the minimax lower bounds are given line 316 indicates that the minimax rate of $\widehat{\lambda}_n$ is of order $\mathcal{O}(n^{-1/2r})$ for $r<1$. As a consequence, the convergence rate of $\widehat{\lambda}_n$ to $\lambda^{\ast}$, which is of order $\mathcal{O}(n^{-1/2})$, is sharp.
>
> **Q5: What is the difference and technical novelty compared to [1]?**
>
> Thanks for your question. Firstly, we would like to refer the reviewer to Question 1 for the difference between our paper and [1]. Secondly, we will elaborate on the technical novelty in our paper as follows:
>
> **Uniform Convergence Rates:** under the distinguishable settings, although our work and [1] both point out that the estimation rate for $\lambda^{\ast}$ is of order $\mathcal{O}(n^{-1/2})$, we arrive at different rates for estimating $(\mu^{\ast},\Sigma^{\ast})$ from that in [1] because of the sample size dependence assumption. Specifically, while the authors of [1] show that this rate is of order $\mathcal{O}(n^{-1/2})$, we demonstrate that it should rather be slower since it is actually determined by the convergence rate of $\lambda^{\ast}$ to zero via the following bound:
>
> $\lambda^{\ast}||(\widehat{\mu}_{n}-\mu^{\ast},$
>
> $\widehat{\Sigma}_{n}-\Sigma^{\ast})||$
>
> $=\mathcal{O}(n^{-1/2})$.
>
> It is clear that these rates are sophisticated and able to highlight the implicit interactions between the convergence rates of various parameter estimations, which remains missing in [1]. To achieve the above rates, we have to face many challenging settings in our proofs. For instance, we first need to make sure that two sequence $G_{n}$ and $G_{\ast,n}$ converge to the same limit $\overline{G}$ under the proposed loss functions. Furthermore, there is still a possibility that the last two components of $G_{n}$ or $G_{\ast,n}$ may not converge to those of $\overline{G}$ under the $2$-norm. Thus, it takes us greater effort to consider all these possible scenarios than in [1] where the authors only need to control the convergence of $G_n$ to $G_{\ast}$.
>
> **References**
>
> [1] H. Nguyen, K. Nguyen, N. Ho. On Parameter Estimation in Deviated Gaussian Mixture of Experts.
>
> [2] S. Gadat, J. Kahn, C. Marteau, and C. Maugis-Rabusseau. Parameter recovery in two-component
> contamination mixtures: The l2 strategy. In Annales de l’Institut Henri Poincaré, Probabilités
> et Statistiques, volume 56, pages 1391–1418. Institut Henri Poincaré, 2020.
>
> [3] D. Do, L. Do, and X. Nguyen. Strong identifiability and parameter learning in regression with
> heterogeneous response. arXiv preprint arXiv:2212.04091, 2022.

---

> > ### Author Response · Authors · 2023-08-19
> >
> > Dear Reviewer 1EFM,
> >
> > We would like to thank you very much for your feedback, and we hope that our response addresses your previous concerns about our paper. However, as the discussion period is expected to conclude in the next few days, please feel free to let us know if you have any further comments on our work. We would be more than happy to address any additional concerns from you.
> >
> > Thank you again for spending time on the paper, we really appreciate that!
> >
> > Best regards,
> >
> > The Authors

---

> > > ### Comment · Reviewer_1EFM · 2023-08-20
> > > **Thank you**
> > >
> > > Dear authors,
> > >
> > > After reading your response, I think we all agree that the setting of your paper and the paper I listed are quite similar, and some of your discovered results are almost the same (as in your response the $O(n^{-1/2})$ convergence rate).
> > >
> > > I agree that there are some differences in the proof procedure, otherwise, they should be in one paper.
> > >
> > > Still, most of your derivations follow almost the same procedure as the paper I mentioned. The differences you mention are mostly driven by some mathematical manipulations. I am afraid the impact of this paper is limited.
> > >
> > > In summary, I would like to remain my previous ratings. Thank you.

---

> > > > ### Author Response · Authors · 2023-08-21
> > > > **Response to Reviewer 1EFM**
> > > >
> > > > Dear Reviewer 1EFM,
> > > >
> > > > Thank you for your response. However, we gracefully disagree with your comments due to the following reasons:
> > > >
> > > > **(1)** The objectives of our paper and [1] are totally different. While our paper focuses on establishing **uniform rates** for parameter estimation in multivariate deviated models, the paper [1] concentrates on deriving **point-wise rates** for parameter estimation in deviated Gaussian mixture of experts. In particular, we allow true parameters $G_{\ast}=(\lambda^{\ast},\mu^{\ast},\Sigma^{\ast})$ to vary with the sample size $n$. Meanwhile, ground-truth parameters in [1] remain unchanged with respect to $n$. Thus, our derived parameter estimation rates are uniform, but those in [1] are point-wise.
> > > >
> > > > Additionally, the convergence behavior of parameter estimation in our paper is not similar to that in [1]. For instance, under the distinguishable settings, [1] claims that the estimation rates for true parameters are of order $\mathcal{O}(n^{-1/2})$. By contrast, our paper points out that the rate for estimating $(\mu^{\ast},\Sigma^{\ast})$ should be lower than $\mathcal{O}(n^{-1/2})$ since it is determined by the convergence rate of $\lambda^{\ast}$ to zero via the following bound:
> > > >
> > > > $\lambda^{\ast}||(\widehat{\mu}_{n}-\mu^{\ast},$
> > > >
> > > > $\widehat{\Sigma}_{n}-\Sigma^{\ast})||=\mathcal{O}(n^{-1/2})$.
> > > >
> > > > It is clear that **our rates are sophisticated and able to highlight the implicit interactions between the convergence rates of various parameter estimations, which remains missing in [1]**. To achieve the above rates, we have to face several challenging settings in our proofs. For instance, we first need to make sure that two sequence $G_{n}$ and $G_{\ast,n}$ converge to the same limit $\overline{G}$ under the proposed loss functions. Furthermore, there is still a possibility that the last two components of $G_{n}$ or $G_{\ast,n}$ may not converge to those of $\overline{G}$ under the $2$-norm. Thus, it takes us greater effort to consider all these possible scenarios than in [1], where the authors only need to control the convergence of $G_n$ to $G_{\ast}$.
> > > >
> > > > Finally, we would like to emphasize that while we present the minimax lower bound results in Section 4 of our paper, such results remain missing in [1].
> > > >
> > > > **(2)** Let us briefly summarize the literature review for deviated models here, and we would like to refer the reviewer to our general response for further details. The general deviated model is given by:
> > > > $$p^{\ast}(x) = \lambda^{\ast} h_0(x) + (1-\lambda^{\ast}) f^{\ast}(x),$$
> > > > where $h_0$ is known and $(\lambda^{\ast}, f^{\ast})$ are to be estimated from data.
> > > >
> > > > (2.1) [2] considers this model specifically when $h_0 = N(0, 1)$ and $f^{\ast} = N(\mu^{\ast}, 1)$ are normal distributions. In the setting $\lambda^{\ast} = n^{-\beta}$ where $\beta \in (0, 1/2)$, they prove that no test can reliably detect $\lambda^{\ast} = 0$ against $\lambda^{\ast} \neq 0$ when  $\lambda^{\ast} \mu^{\ast} = o(n^{-1/2})$, while the Likelihood Ratio Test can consistently do it when $\lambda^{\ast} \mu^{\ast} \gtrsim n^{-1/2+\epsilon}$ for any $\epsilon > 0$. However, no guarantee for estimation of $\lambda^{\ast}$ and $\mu^{\ast}$ is provided.
> > > >
> > > > (2.2) The uniform convergence of estimating $\lambda^{\ast}$ and $\mu^{\ast}$ is then revisited in [3], in the same setting, where it provides minimax rate and uniform convergence rates for both $\lambda^{\ast}$ and $\mu^{\ast}$ under the $l^2$ estimation strategy. They prove the tight convergence rate for $\lambda^{\ast}$ and $\mu^{\ast}$ when $\lambda^{\ast} |\mu^{\ast}| \gtrsim n^{-1/2 + \epsilon}$ and $|\mu^{\ast}| \gtrsim n^{-1/4}$. However, their technique heavily relies on the properties of the location Gaussian family, which might be difficult to generalize to other settings of kernel densities.
> > > >
> > > > **(3)** Minor point: The paper [1] that the reviewer mentioned is actually a draft and it has not been published at any official venues (journals, conferences or even arXiv).
> > > >
> > > >
> > > > With the three reasons (1)-(3), we think that it is not fair to use [1] as the main reason to reject our paper. We hope that the reviewer will consider re-evaluating our paper.
> > > >
> > > > **References**
> > > >
> > > > [1] H. Nguyen, K. Nguyen, N. Ho. On Parameter Estimation in Deviated Gaussian Mixture of Experts.
> > > >
> > > > [2] T. Cai. Optimal detection of heterogeneous and heteroscedastic mixtures. Journal of the Royal Statistical Society: Series B (Statistical Methodology), 2011.
> > > >
> > > > [3] S. Gadat. Parameter recovery in two-component contamination mixtures: The l2 strategy. In Annales de l’Institut Henri Poincaré, Probabilitéset Statistiques. Institut Henri Poincaré, 2020.
> > > >
> > > > [4] Heinrich, Philippe, and Jonas Kahn. 2018. “Strong Identifiability and Optimal Minimax Rates for Finite Mixture Estimation.” The Annals of Statistics 46 (6A): 2844–70.

---

### Author Rebuttal · Authors · 2023-08-09

**General Response**

Dear AC and reviewers,

We would like to express our gratitude for your constructive reviews, which help us improve our work significantly. There are two common concerns about the literature on the deviated models and the novelty of our paper. Thus, we dedicate this general response to clarify these concerns and then include them in the revision of our paper.

**I. Literature of the deviated models**

Let us consider the general model:
$$p^{\ast}(x) = \lambda^{\ast} h_0(x) + (1-\lambda^{\ast}) f^{\ast}(x),$$
where $h_0$ is known and $(\lambda^{\ast}, f^{\ast})$ are to be estimated from data. Several attempts to provide theoretical guarantees and algorithms for this problem have been made in the literature. Starting from multiple-testing problems (microarray analysis,
neuroimaging) [1, 2], the p-values, obtained from the numerous (independent) hypotheses tests, are uniformly distributed on $[0,1]$, under hypothesis $H_0$. Hence $h_0$ is the uniform distribution, and $f^*$ is required to be estimated. When $f^*$ is assumed to be symmetric and non-parametric, [3] provides identifiability and consistency results of the parameter estimation. In the parametric setting of $f^*$, the related work is listed below.

1. [4] considers this model specifically when $h_0 = N(0, 1)$ and $f^* = N(\mu^*, 1)$ are normal distributions. In the setting $\lambda^* = n^{-\beta}$ where $\beta \in (0, 1/2)$, they prove that no test can reliably detect $\lambda^* = 0$ against $\lambda^* \neq 0$ when  $\lambda^* \mu^* = o(n^{-1/2})$, while the Likelihood Ratio Test can consistently do it when $\lambda^* \mu^* \gtrsim n^{-1/2+\epsilon}$ for any $\epsilon > 0$. However, no guarantee for estimation of $\lambda^*$ and $\mu^*$ is provided.

2. The uniform convergence of estimating $\lambda^*$ and $\mu^*$ is then revisited in [5], in the same setting, where it provides minimax rate and uniform convergence rates for both $\lambda^*$ and $\mu^*$ under the $l^2$ estimation strategy. They prove the tight convergence rate for $\lambda^*$ and $\mu^*$ when $\lambda^* |\mu^*| \gtrsim n^{-1/2 + \epsilon}$ and $|\mu^*| \gtrsim n^{-1/4}$. However, their technique heavily relies on the properties of the location Gaussian family, which might be difficult to generalize to other settings of kernel densities.

3. When $f^* = N(\mu^*, \Sigma^*)$, it is possible to derive the estimation rate for $(\lambda^*, \mu^*, \Sigma^*)$ from results for the general mixture of two components in the literature. However, those bounds are often less sharp compared to the bound that we developed in our paper due to the lack of information about the known component.
In particular, when estimating $\mu^*$ with $\Sigma^*$ being fixed, an application of the results with moment methods from [7] to the deviated models leads to $||\Delta \mu^{\ast}||^{3} |\lambda_{n}^{\text{moment}}-\lambda^{\ast}| = \mathcal{O}(n^{-1/2})$ and $\lambda^{\ast} ||\mu_{n}^{\text{moment}} - \mu^{\ast}||^3 = \mathcal{O}(n^{-1/2})$, which are much slower compared to the results for the MLE in the strongly identifiable and non-distinguishable settings in our work, where $(\lambda_{n}^{\text{moment}}, \mu_{n}^{\text{moment}})$ denote moment estimators of $\lambda^{\ast},\mu^{\ast}$. When we estimate both $\mu^*$ and $\Sigma^*$, an adaptation of the moment estimators from the seminal work [6] to the multivariate deviated models show $(||\Delta \mu^{\ast}||^{6} + ||\Delta \Sigma^{\ast}||^{3})  |\lambda_{n}^{\text{moment}}-\lambda^{\ast}| = \mathcal{O}(n^{-1/2})$, which is also slower than those of the MLE in weakly identifiable setting.

**II. The novelty of our paper**

**1. Novel settings:** We allow ground-truth parameters $G_*=(\lambda^{\ast},\mu^{\ast},\Sigma^{\ast})$ to change with the sample size $n$, which is closer to practical settings than assumptions in previous work [8]. However, this induces two main obstacles:
 (i) $h_0$ may belong to the distribution family of $f$, which leads to some interaction between these densities;
 (ii) The deviated proportion $\lambda^{\ast}$ can go to zero as the sample size is sufficiently large. Then, any pairs of $(\mu^{\ast},\Sigma^{\ast})$ will induce the same model $h_0$, which makes the parameter estimation more challenging.

**2. Uniform Convergence Rates:** Since true parameters may vary with the sample size, the convergence rates of parameter estimations in our work are uniform rather than point-wise, as in [8]. Additionally, these rates are able to capture the interaction between the convergences of parameter estimations.

**3. Minimax Lower Bounds:** Finally, we determine minimax lower bounds under both distinguishable and non-distinguishable settings. Based on these lower bounds, we deduce that our derived convergence rates are sharp.


[1] B. Efron. Empirical Bayes analysis of a microarray experiment. Journal of the American statistical association 96, no. 456 (2001): 1151-1160.

[2] S. Robin. "A semi-parametric approach for mixture models: Application to local false discovery rate estimation." Computational statistics \& data analysis 51, no. 12 (2007): 5483-5493.

[3] L. Bordes. Semiparametric estimation of a two-component mixture model where one component is known. Scandinavian journal of statistics, 33(4):733–
752, 2006.

[4] T. Cai. Optimal detection of heterogeneous and heteroscedastic mixtures.
Journal of the Royal Statistical Society: Series B (Statistical Methodology), 73(5):629–662,
2011.

[5] S. Gadat. Parameter recovery in two-component contamination mixtures: The l2 strategy. In Annales de l’Institut Henri Poincaré, Probabilitéset Statistiques, volume 56, pages 1391–1418. Institut Henri Poincaré, 2020.

[6] M. Hardt. Tight bounds for learning a mixture of two gaussians. In STOC, 2015.

[7] Y. Wu. Optimal estimation of Gaussian mixtures via denoised method of moments.
The Annals of Statistics, 48:1987–2007, 2020.

[8] H. Nguyen. On Parameter Estimation in Deviated Gaussian Mixture of Experts.

---

### Decision · Program_Chairs · 2023-09-21

**Decision:**

Accept (poster)

**Comment:**

The reviewers have a mixed opinion on this work. One reviewer maintains in their review and throughout the discussion phase that the submission is too similar to another paper. Other reviews support acceptance of the paper. I believe the authors have very clearly explained why their submission differs from the other paper in important ways; therefore, I do not think the other paper is relevant for the decision-making process on the submission and will focus on the other reviews.

I will briefly summarize the positives:
- The technical contributions are strong; the proofs have a high level of sophistication.
- The case when distinguishability does not hold seems very challenging, and the authors' results here are impressive.
- The bounds holding uniformly over the parameter space and so also give results when $\lambda^*$ goes to zero, which is quite interesting.
- There are potential applications in domain adaptation.

One reviewer wondered if NeurIPS is a good venue for this work, as compared to a stats journal. I believe results like this do have a home in NeurIPS.

The main downside is (as mentioned by several reviewers) the inadequacy of the related work section. The authors have said they will add a detailed literature review in the revised version.

This looks like a solid paper and is a welcome addition to the conference proceedings. As promised, please be sure to add the detailed literature review; it is vital for contextualizing the results.